# Defense Against Reward Poisoning Attacks in Reinforcement Learning

**Kiarash Banihashem**\*                                              *kiarash@umd.edu*
*University of Maryland*

**Adish Singla**                                                   *adishs@mpi-sws.org*
*Max Planck Institute for Software Systems*

**Goran Radanovic**                                          *gradanovic@mpi-sws.org*
*Max Planck Institute for Software Systems*

**Reviewed on OpenReview:** *https://openreview.net/forum?id=goPsLn3RVo*

## Abstract

We study defense strategies against reward poisoning attacks in reinforcement learning. As a threat model, we consider cost-effective targeted attacks—these attacks minimally alter rewards to make the attacker's target policy uniquely optimal under the poisoned rewards, with the optimality gap specified by an attack parameter. Our goal is to design agents that are robust against such attacks in terms of the worst-case utility w.r.t. the true, unpoisoned, rewards while computing their policies under the poisoned rewards. We propose an optimization framework for deriving optimal defense policies, both when the attack parameter is known and unknown. For this optimization framework, we first provide characterization results for generic attack cost functions. These results show that the functional form of the attack cost function and the agent's knowledge about it are critical for establishing lower bounds on the agent's performance, as well as for the computational tractability of the defense problem. We then focus on a cost function based on $\ell_2$ norm, for which we show that the defense problem can be efficiently solved and yields defense policies whose expected returns under the true rewards are lower bounded by their expected returns under the poison rewards. Using simulation-based experiments, we demonstrate the effectiveness and robustness of our defense approach.

## 1 Introduction

One of the key challenges in designing trustworthy AI systems is ensuring that they are technically robust and resilient to security threats European Commission (2019). Amongst many requirements that an AI system ought to satisfy in order to be deemed trustworthy is robustness to adversarial attacks Hamon et al. (2020). Standard approaches to reinforcement learning (RL) Sutton & Barto (2018) have shown to be susceptible to adversarial attacks which manipulate the feedback that an agent receives from its environment. These attacks broadly fall under two categories: a) *test-time* attacks, which manipulate an agent's input data at test-time without changing its policy Huang et al. (2017); Lin et al. (2017); Tretschk et al. (2018); Kumar et al. (2021), and b) *training-time* attacks that manipulate an agent's input data at training-time, influencing the agent's learned policy Zhang & Parkes (2008); Ma et al. (2019); Huang & Zhu (2019); Rakhsha et al. (2020; 2021); Zhang et al. (2020b); Sun et al. (2020); Liu & Lai (2021); Rangi et al. (2022). In this paper, we focus on training-time attacks, and more specifically, on *targeted reward poisoning* attacks that modify (i.e., *poison*) rewards to force an agent into adopting a *target* policy Ma et al. (2019); Rakhsha et al. (2021); Rangi et al. (2022). Prior work on reward poisoning attacks on RL primarily focuses on designing optimal attacks. In this paper,

---

\*This work was done as a part of an internship project at Max Planck Institute for Software Systems.

we take a different perspective on *targeted* reward poisoning attacks, and focus on designing *defense strategies* against such attacks. This is challenging, given that the attacker is typically unconstrained in poisoning the rewards to force the target policy, while the agent's performance is measured under the true reward function, which is unknown. The key idea that we exploit in our work is that the poisoning attacks have an underlying *structure* arising from the attacker's objective to minimize the cost of the attack needed to force the target policy. We therefore ask the following question:

*Can we design an effective defense strategy against targeted reward poisoning attacks by exploiting the cost-effective nature of these attacks?*

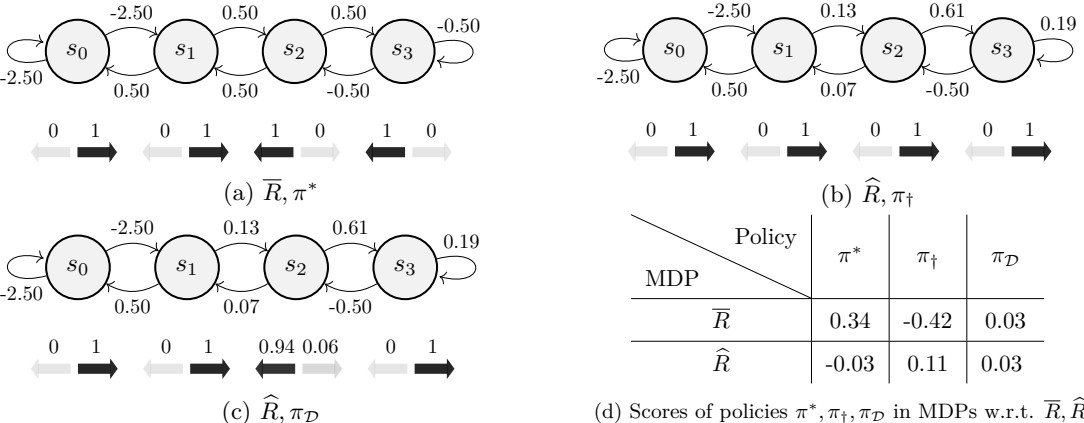

(a) $\overline{R}, \pi^*$

(b) $\widehat{R}, \pi_\dagger$

(c) $\widehat{R}, \pi_\mathcal{D}$

(d) Scores of policies $\pi^*, \pi_\dagger, \pi_\mathcal{D}$ in MDPs w.r.t. $\overline{R}, \widehat{R}$

| MDP \ Policy | $\pi^*$ | $\pi_\dagger$ | $\pi_\mathcal{D}$ |
|---|---|---|---|
| $\overline{R}$ | 0.34 | -0.42 | 0.03 |
| $\widehat{R}$ | -0.03 | 0.11 | 0.03 |

Figure 1: A simple chain environment with 4 states and two possible actions: *left* and *right*. $s_0$ is the initial state. The agent goes in the direction of its action with probability 90%, and otherwise the next state is selected uniformly at random from the other 3 states. Weights on edges indicate rewards for the action taken. For example in Fig. 1a, if the agent takes *left* in state $s_1$, it receives 0.5. We denote the true rewards by $\overline{R}$, the poisoned rewards by $\widehat{R}$, the optimal policy under $\overline{R}$ by $\pi^*$, the target policy (which is uniquely optimal under $\widehat{R}$) by $\pi_\dagger$, and the defense policy (which is derived from our framework) by $\pi_\mathcal{D}$. **(a)** shows $\overline{R}$ and $\pi^*$. In particular, the numbers above the arrows and the different shades of gray show the probabilities of taking actions *left* and *right* under $\pi^*$. **(b)** shows $\widehat{R}$ and $\pi_\dagger$. **(c)** shows $\pi_\mathcal{D}$ that our optimization framework derived from $\widehat{R}$, and by reasoning about the goal of the attack ($\pi_\dagger$). In particular, our optimization framework maximizes the worst-case performance under $\overline{R}$: while the optimization procedure does not know $\overline{R}$, it can constrain the set of plausible candidates for $\overline{R}$ using $\widehat{R}$. **(d)** Table. 1d): Each entry in the table indicates the score of a (policy, reward function) pair, where the score is a scaled version of the total discounted return (see Section 3). For example, the score of policy $\pi_\dagger$ equals $-0.42$ and $0.11$ under $\widehat{R}$ and $\overline{R}$ respectively. Our defense policy significantly improves upon this and achieves a score of 0.03. For comparison, the score of $\pi^*$ equals 0.34. Moreover, unlike for the target policy $\pi_\dagger$, the score of our defense policy $\pi_\mathcal{D}$ under $\overline{R}$ is always at least as high as its score under $\widehat{R}$, as predicted by our results (see Theorem 5.1). The results are obtained with parameters $\epsilon_\dagger = 0.1$, $\epsilon_\mathcal{D} = 0.2$ and $\gamma = 0.99$ (see Section 3).

In this paper, we study this question in depth and under different assumption on the agent's knowledge about the attack cost function. Perhaps surprisingly, the answer to this question is sometimes affirmative. While an agent only has access to the poisoned rewards, it may still be able infer some information about the true reward function, using the fact that the attack is cost-effective. By maximizing the worst-case utility over the set of plausible candidates for the true reward function, the agent can substantially limit the influence of the attack. The approach we study can be understood from Figure 1 which uses the chain environment from Rakhsha et al. (2021) to demonstrate the main ideas.

**Contributions.** We formalize this reasoning, and characterize the utility of our novel framework for designing defense policies. In summary, the key contributions include:

- We formalize the problem of designing defense policies against targeted and cost-effective reward poisoning attacks, which minimally modify the original reward function to achieve their goal (force a target policy).

- We introduce a novel optimization framework for finding *optimally robust* defense policies—this framework focuses on optimizing the agent's worst-case utility among the set of reward functions that are plausible candidates of the true reward function.
- We provide characterization results that establish feasibility and computational complexity of finding optimally robust defense policies for different classes of attack cost functions. These results show that the functional form of the attack cost function and the agent's knowledge about it play a critical role in deriving optimally robust defense policies with provable performance guarantees.
- Focusing on a cost function based on $\ell_2$ norm, we show that optimally robust defense policies can be efficiently computed. We further establish lower bounds on the true performance of defense policies derived from our framework and computable from the poisoned rewards.
- We empirically demonstrate the effectiveness and robustness of our approach using numerical simulations.

To our knowledge, this is the first framework for studying this type of defenses against reward poisoning attacks that try to force a target policy at a minimal cost.

## 2 Related Work

While this paper is broadly related to the literature on adversarial machine learning (e.g., Huang et al. (2011)), we recognize four themes in supervised learning (SL) and reinforcement learning (RL) that closely connect to our work.

**Poisoning attacks in SL and RL.** This paper is closely related to data poisoning attacks, first introduced and extensively studied supervised learning Biggio et al. (2012); Xiao et al. (2012); Mei & Zhu (2015); Xiao et al. (2015); Li et al. (2016); Koh & Liang (2017); Biggio & Roli (2018). These attacks are also called *training-time attacks*, and unlike *test time attacks* Szegedy et al. (2014); Pinto et al. (2017); Behzadan & Munir (2017); Zhang et al. (2020a); Moosavi-Dezfooli et al. (2016); Nguyen et al. (2015); Madry et al. (2018), which attack an already trained agent, they change data points during the training phase, which in turn affects the parameters of the learned model. Data poisoning attacks have also been studied in the bandits literature Jun et al. (2018); Ma et al. (2018); Liu & Shroff (2019) and in RL (see Section 1).

**Defenses against poisoning attacks in SL.** In supervised learning, defenses against data poisoning attacks are often based on data sanitization that removes outliers from the training set Cretu et al. (2008); Paudice et al. (2018), trusted data points that support robust learning Nelson et al. (2008); Zhang et al. (2018), or robust estimation Charikar et al. (2017); Diakonikolas et al. (2019). Recently, Wu et al. (2022) have considered aggregation based defenses that can certify an RL agent's policy against a limited number of changes in the training dataset. While such defenses can mitigate some attack strategies, they are in general susceptible to data poisoning attacks Steinhardt et al. (2017); Koh et al. (2018).

**Robustness to model uncertainty.** There is a rich literature that studies robustness to uncertainties in reward functions McMahan et al. (2003); Regan & Boutilier (2010), and transition models Nilim & El Ghaoui (2005); Iyengar (2005); Bagnell et al. (2001) for MDP models. Typically, these works consider settings in which instead of knowing the exact parameters of the MDP, the agent has access to a set of possible parameters (uncertainty set). These works design policies that perform well in the worst case. More recent works have proposed ways to scale up these approaches via function approximation Tamar et al. (2014), as well as utilize them in online settings Lim et al. (2013). While our work uses the same principles of robust optimization, we do not assume that the uncertainty set, i.e., the set of all possible rewards, is directly given. Instead, we show how to derive it from the poisoned reward function.

**Robustness to corrupted episodes.** Another important line of work is the literature on robust learners that receive corrupted input during their training phase. Such learners have recently been designed for bandits and experts settings Lykouris et al. (2018); Gupta et al. (2019); Bogunovic et al. (2020); Amir et al. (2020), and episodic reinforcement learning Lykouris et al. (2019); Zhang et al. (2021). Typically, these works consider an attack model in which the adversary can arbitrarily corrupt a limited number of episodes. As we operate in the non-episodic setting and do not assume a limit in the attacker's poisoning budget, these works are orthogonal to the aspects we study in this paper. Instead, we utilize the *structure* of the attack in order to design a defense algorithm.

## 3 Formal Setting

In this section, we describe our formal setting, and identify relevant background details on reward poisoning attacks, as well as our problem statement. The problem formulation specifies our objectives that we establish and formally analyze in the next sections.

### 3.1 Preliminaries

We consider a standard reinforcement learning setting in which the environment is described by a discrete-time discounted Markov Decision Processes (MDP) Puterman (1994), defined as $M = (S, A, R, P, \gamma, \sigma)$, where: $S$ is the state space, $A$ is the action space, $R : S \times A \to \mathbb{R}$ is the reward function, $P : S \times A \times S \to [0, 1]$ is the transition model with $P(s, a, s')$ defining the probability of transitioning to state $s'$ by taking action $a$ in state $s$, $\gamma \in [0, 1)$ is the discount factor, and $\sigma$ is the initial state distribution. We consider state and action spaces, i.e., $S$ and $A$, that are finite and discrete, and due to this we can adopt a vector notation for quantities dependent on states or state-action pairs. W.l.o.g., we assume that $|A| \geq 2$.

A generic (stochastic) policy is denoted by $\pi$, and it is a mapping $\pi : S \to \mathcal{P}(A)$, where $\mathcal{P}(A)$ is the probability simplex over action space $A$. We use $\pi(a|s)$ to denote the probability of taking action $a$ in state $s$. While deterministic policies are a special case of stochastic policies, when explicitly stating that a policy $\pi$ is deterministic, we assume that it is a mapping from states to actions, i.e., $\pi : S \to A$. We denote the set of all policies by $\Pi$ and the set of all deterministic policies by $\Pi^{\text{det}}$. For policy $\pi$, we define its *score*, $\rho^\pi$, as $\mathbb{E}\left[(1 - \gamma) \sum_{t=1}^\infty \gamma^{t-1} R(s_t, a_t) | \pi, \sigma\right]$, where state $s_1$ is sampled from the initial state distribution $\sigma$, and then subsequent states $s_t$ are obtained by executing policy $\pi$ in the MDP. The score of a policy is therefore its total expected return scaled by a factor of $1 - \gamma$.

Finally, we consider occupancy measures. We denote the state-action occupancy measure in the Markov chain induced by policy $\pi$ by $\psi^\pi(s, a) = \mathbb{E}\left[(1 - \gamma) \sum_{t=1}^\infty \gamma^{t-1} \mathbb{1}\left[s_t = s, a_t = a\right] | \pi, \sigma\right]$. Given the MDP $M$, the set of realizable state-action occupancy measures under any (stochastic) policy $\pi \in \Pi$ is denoted by $\Psi$. Score $\rho^\pi$ and $\psi^\pi$ satisfy $\rho^\pi = \langle \psi^\pi, R \rangle$, where $\langle ., . \rangle$ computes the dot product between two vectors of sizes $|S| \cdot |A|$. We denote by $\mu^\pi(s) = \mathbb{E}\left[(1 - \gamma) \sum_{t=1}^\infty \gamma^{t-1} \mathbb{1}\left[s_t = s\right] | \pi, \sigma\right]$ the state occupancy measure in the Markov chain induced by policy $\pi \in \Pi$. State-action occupancy measure $\psi^\pi(s, a)$ and state occupancy measure $\mu^\pi(s)$ satisfy $\psi^\pi(s, a) = \mu^\pi(s) \cdot \pi(a|s)$. We focus on *ergodic* MDPs, which in turn implies that $\mu^\pi(s) > 0$ for all $\pi$ and $s$ Puterman (1994). This is a standard assumption in this line of work (e.g, see Rakhsha et al. (2021)) and is used to ensure the feasibility of the attacker's optimization problem.

### 3.2 Reward Poisoning Attacks

We consider reward poisoning attacks on an offline learning agent that optimally change the original reward function with the goal of deceiving the agent to adopt a deterministic policy $\pi_\dagger \in \Pi^{\text{det}}$, called *target policy*. This type of attack has been extensively studied in the literature, and here we utilize the attack formulation based on the works of Ma et al. (2019); Rakhsha et al. (2020; 2021); Zhang et al. (2020b). In the following, we introduce the necessary notation, the attacker's model, and the agent's model (without defense).

**Notation.** We use $\overline{M}$ to denote the *true* or *original* MDP with true, unpoisoned, reward function $\overline{R}$, i.e., $\overline{M} = (S, A, \overline{R}, P, \gamma, \sigma)$. We use $\widehat{M}$ to denote the *modified* or *poisoned* MDP with poisoned reward function $\widehat{R}$, i.e., $\widehat{M} = (S, A, \widehat{R}, P, \gamma, \sigma)$. Note that only the reward function $R$ changes across these MDPs. Quantities that depend on reward functions have analogous notation. For example, the score of policy $\pi$ under $\overline{R}$ is denoted by $\overline{\rho}^\pi$, whereas its score under $\widehat{R}$ is denoted by $\widehat{\rho}^\pi$. We denote an optimal policy under $\overline{R}$ by $\pi^*$, i.e., $\pi^* \in \arg\max_{\pi \in \Pi} \overline{\rho}^\pi$.

**Attack model.** The attacker we consider in this paper has full knowledge of $\overline{M}$. It can be modeled by a function $\mathcal{A}(c, R', \pi_\dagger, \epsilon_\dagger)$ that returns a set of poisoned rewards functions for a given attack cost function $c$, reward function $R'$, target policy $\pi_\dagger$, and a desired attack parameter $\epsilon_\dagger$.[1] In particular, the attack problem is defined by the following optimization problem:

$$\min_R \quad c(R, R') \quad \text{s.t.} \quad \rho^{\pi_\dagger} \geq \rho^\pi + \epsilon_\dagger \quad \forall \pi \in \Pi^{\text{det}} \backslash \{\pi^\dagger\}. \tag{P1}$$

---

[1] For cost functions $c_p$ defined by (1) with finite $p > 1$, this set has a single element.

A common class of cost functions are $\ell_p$-norms of manipulations Ma et al. (2019); Rakhsha et al. (2020; 2021), i.e.,

$$c(R, R') = c_p(R, R') = \|R - R'\|_p,  \tag{1}$$

with $p \geq 1$. As shown by Rakhsha et al. (2021), the attack problem (P1) is feasible for this class of cost functions and ergodic MDPs. Furthermore, instead of considering all deterministic policies, it is sufficient to consider policies that differ from $\pi_\dagger$ in a single action. Using $\pi_\dagger\{s; a\}$ to denote a policy that follows $a \neq \pi_\dagger(s)$ in state $s$ and $\pi_\dagger(\tilde{s})$ in states $\tilde{s} \neq s$, (P1) can be rewritten as follows:

$$\min_R \quad c(R, R') \quad \text{s.t.} \quad \rho^{\pi_\dagger} \geq \rho^{\pi_\dagger\{s;a\}} + \epsilon_\dagger \quad \forall s, a \neq \pi_\dagger(s).  \tag{P1'}$$

The equivalence between (P1) and (P1') is shown by Rakhsha et al. (2021), but we also provide further details in Appendix. Intuitively, (P1) and (P1') are equivalent because in order for a policy to be optimal it is sufficient (and necessary) that the policy is better than any of its neighbor policies. This fact allows us to reduce the number of constraints. Whereas the number of constraints in (P1) is exponential in $|S|$ and $|A|$, the number of constraint in (P1') is polynomial in $|S|$ and $|A|$, which in turn implies that (P1') is tractable (for a fixed $p$).

To better understand the optimization problem (P1'), we can consider $\ell_2$ attack cost (i.e., $c_2$) defined as the Euclidean distance between $R'$ and $R$. By solving this problem, i.e., setting $\widehat{R} \in \mathcal{A}(c_2, \overline{R}, \pi_\dagger, \epsilon_\dagger)$, the attacker finds the closest reward function to $\overline{R}$ for which $\pi_\dagger$ is a uniquely optimal policy (with attack parameter $\epsilon_\dagger$).

*Remark* 3.1. Note that the optimization problem (P1) may not be feasible if we lift the assumption that underlying MDP is ergodic. This can be seen from the constraints of the optimization problem (P1'): if target policy $\pi_\dagger$ does not visit a certain state $s$, then it has the same score as its neighbor policies $\pi_\dagger\{s; a\}$. Hence, the primary reason for assuming ergodicity is to make the attack problem feasible.

**Agent without defense.** The agent receives the poisoned MDP $\widehat{M} := (S, A, \widehat{R}, P, \gamma, \sigma)$ where the underlying true reward function $\overline{R}$ (unknown to the agent) has been poisoned to $\widehat{R}$. In the existing works on reward poisoning attacks, an agent naively optimizes score $\widehat{\rho}$ (score w.r.t. $\widehat{R}$). Because of this, the agent ends up adopting policy $\pi_\dagger$.

### 3.3 Problem Statement

Perhaps unsurprisingly, the agent without defense, could perform arbitrarily badly under the true reward function $\overline{R}$. Our goal is to design a robust agent that derives its policy using the poisoned MDP $\widehat{M} := (S, A, \widehat{R}, P, \gamma, \sigma)$, but has provable worst-case guarantees w.r.t. $\overline{R}$. This agent has access to the poisoned reward vector $\widehat{R} \in \mathcal{A}(c, \overline{R}, \pi_\dagger, \epsilon_\dagger)$, but $\overline{R}$, $\pi_\dagger$, and $\epsilon_\dagger$ are not given to the agent. Figure 2 illustrates a generic problem setting studied in this paper. In general, the agent does not know $c$, but is given a class of cost functions $\mathcal{C}$ that contains $c$, i.e., $c \in \mathcal{C}$. $\mathcal{C}$ represents' the agent's knowledge about the attack cost function; in a special case when $\mathcal{C}$ contains only one element, the agent knows the cost function. Notice that $\pi_\dagger$ is obtainable by solving the optimization problem $\arg\max_\pi \widehat{\rho}^\pi$ as $\pi_\dagger$ is uniquely optimal in $\widehat{M}$. On the other hand, $\overline{R}$ is unknown to the agent. In terms of $\epsilon_\dagger$, we will focus on two cases, the case when $\epsilon_\dagger$ is known to the agent, and the case when it is not.

In the first case, we can formulate the following optimization problem of maximizing the worst case performance of the agent, given that $\overline{R}$ is unknown:

$$\max_\pi \min_{R, c \in \mathcal{C}} \rho^\pi \quad \text{s.t.} \quad \widehat{R} \in \mathcal{A}(c, R, \pi_\dagger, \epsilon_\dagger).  \tag{P2a}$$

In other words, we calculate the set of all possible reward functions $R$ such that an attack on $R$ *could* lead to the solution $\widehat{R}$, i.e., $\widehat{R} \in \mathcal{A}(c, R, \pi_\dagger, \epsilon_\dagger)$. We then find a policy $\pi$ for which the worst-case score, i.e., $\min_{R,c} \rho^\pi$ is maximized, where the minimum is over all reward functions $R$ calculated previously and all cost functions $c \in \mathcal{C}$.

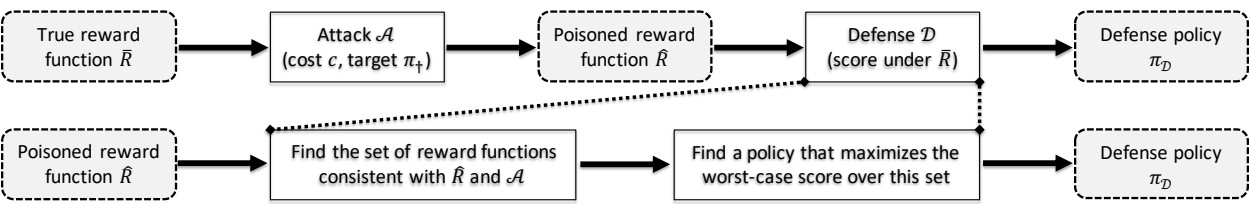

Figure 2: The problem setting studied in this paper. The attack $\mathcal{A}$ modifies the original (true) reward function $\overline{R}$ to force $\pi_\dagger$ while minimizing its cost. The defense $\mathcal{D}$ aims to optimize the agent's score under $\overline{R}$, but it only sees the poisoned reward $\widehat{R}$. Nevertheless, it can (in principle) find the set of all reward functions consistent with $\widehat{R}$ and $\mathcal{A}$, and search for a policy that maximizes the worst-case score of the agent over this consistent set.

| Knowledge about attack cost | Guarantees on the value | Complexity |
|---|---|---|
| General $\mathcal{C}$ (e.g., s.t. $c_{\mathrm{const}} \in \mathcal{C}$) | **No** for any $\widehat{R}$, Proposition 4.2 | — |
| $\mathcal{C} = \{\|R - R'\|_p \text{ s.t. } p \in [1, \infty)\}$ | **Yes**, Theorem 4.4 | **NP-hard**, Theorem 4.5 |
| $\mathcal{C} = \{\|R - R'\|_\infty\}$ | **No** for some $\widehat{R}$, Theorem 4.3 | **NP-hard**, Appendix |
| $\mathcal{C} = \{\|R - R'\|_2\}$ | **Yes**, Section 5 | **Convex**, Section 5 |
| $\mathcal{C} = \{\|R - R'\|_1\}$ | **Yes**, Appendix | **Convex**, Appendix |

Table 1: Characterization results for different levels of the agent's knowledge about the attack cost function, expressed through $\mathcal{C}$. In general, if $\mathcal{C}$ can be arbitrary, the optimization problem (P2a) may be unbounded from below regardless of $\widehat{R}$. For some classes $\mathcal{C}$, e.g., that contain $\ell_p$-norm attack costs ($p \neq \infty$), (P2a) has the optimal solution, but this solution may be computationally hard. When the attack cost function is known, properties of problem (P2a) depend on the functional form of the attack cost function, as indicated by the $\ell_1$-norm, $\ell_2$-norm, and $\ell_\infty$-norm attack costs.

For the case when the agent does not know $\epsilon_\dagger$, we use the following optimization problem:

$$\max_\pi \min_{R, \epsilon, c \in \mathcal{C}} \rho^\pi \quad \text{s.t.} \quad \widehat{R} \in \mathcal{A}(c, R, \pi_\dagger, \epsilon) \text{ and } 0 < \epsilon \leq \epsilon_\mathcal{D}. \tag{P2b}$$

where the agent uses $\epsilon_\mathcal{D}$ as an upper bound on $\epsilon_\dagger$. We denote solutions to the optimization problems (P2a) and (P2b) by $\pi_\mathcal{D}$, and it will be clear from the context which optimization problem we are referring to with $\pi_\mathcal{D}$.

*Remark* 3.2. We note that some structural assumptions on the attack model are needed to guarantee robustness. For example, prior work typically considers untargeted attacks with budget constraints, e.g., that put an upper limit on the cost of the attack or the number of episodes in which the attacker can attack Lykouris et al. (2019); Zhang et al. (2021). This paper focuses on targeted attacks that minimize the cost of the attack. Given the strategic nature of the attacker, the structural assumptions on the attack model that (P2a) and (P2b) rely on are fairly natural.

## 4 Characterization Results for Generic Attack Cost

In this section, we provide characterization results showing the importance of the agent's knowledge about the attack cost function. The overview of the characterization results is shown in Table 1. To corresponding proofs can be found in Appendix.

*Remark* 4.1. The results presented in this section are stated for the optimization problem (P2a). However, the same results also hold for the optimization problem (P2b).

### 4.1 General Attack Cost

We start by stating what is perhaps an expected result: if the attack cost function can be arbitrary, then no defense can achieve any provable guarantee. It is relatively easy to see why this claim should hold. If the

agent believes that the attack cost function can be constant, then from the agent's perspective, $\overline{R}$ can be any reward function. Since rewards are not bounded, this in turn implies that no matter which policy the agent selects, no worst-case guarantees are possible. More formally, we obtain the following claim.

**Proposition 4.2.** *Let $c_{const}(R, R')$ be a constant cost function, and assume that $c_{const} \in \mathcal{C}$. Then the optimization problem* (P2a) *is unbounded from below.*

Given this result, it is clear that for provable defenses: a) the attack cost function cannot not be arbitrary in that $\widehat{R}$ has to be informative about $\overline{R}$; b) the agent should have some knowledge about the attack cost function. In the next subsection, we consider cost functions based on $\ell_p$ norms, i.e., $c_p$, commonly adopted by prior work on reward poisoning attacks Ma et al. (2019); Rakhsha et al. (2020; 2021).

## 4.2 $\ell_p$ Attack Cost

In contrast to constant cost functions, the strategic nature of the attacker is more apparent when it optimizes $c_p$, so the agent may infer some information about the true reward function $\overline{R}$ from the poisoned rewards $\widehat{R}$ which it can access. Our first result shows that for $p = \infty$ the success of such an inference procedure depends on $\widehat{R}$. This is formally captured by the following theorem.

**Theorem 4.3.** *There exists an instance of the problem setting, i.e., MDP $M = (S, A, \widehat{R}, P, \gamma, \sigma)$ for which the optimization problem* (P2a) *is unbounded from below when $c_\infty \in \mathcal{C}$.*

This theorem paints a relatively bleak picture for the possibility of achieving provable guarantees. Note two important observations. First, the impossibility result is a weaker variant of the result stated in Proposition 4.2 as it holds only for some MDPs. Second, $c_\infty$ is measuring the maximum modification of reward function $\overline{R}$, so critical information about $\overline{R}$ may be lost—this also provides intuition behind the impossibility results. In contrast, when $p \neq \infty$, $c_p$ is affected by all the modifications of $\overline{R}$. In fact, when $p$ is restricted to take values in $[1, \infty)$, a lower bound on the optimal value of (P2a) can always be derived from $\widehat{R}$.

**Theorem 4.4.** *Consider any policy $\pi_{\not{\dagger}}$ s.t. $\pi_{\not{\dagger}}(\pi_\dagger(s)|s) = 0$. For $\mathcal{C} = \{c_p \text{ s.t. } p \in [1, \infty)\}$, the optimal value of the optimization problem* (P2a) *is bounded from below by $\widehat{\rho}^{\pi_{\not{\dagger}}}$.*

A direct consequence of Theorem 4.4 is that the optimal solution to (P2a) always exists, and its worst case performance under $\overline{R}$ is at least $\widehat{\rho}^{\pi_{\not{\dagger}}}$. Note that we can easily find a (deterministic) policy $\pi_{\not{\dagger}}$ that maximizes the lower bound by solving $\max_{\pi \text{ s.t. } \pi(s) \neq \pi_\dagger(s)} \widehat{\rho}^\pi$.[2] Furthermore, for any policy $\pi_{\not{\dagger}}$ s.t. $\pi_{\not{\dagger}}(\pi_\dagger(s)|s) = 0$ we have that $\overline{\rho}^{\not{\dagger}} \geq \widehat{\rho}^{\not{\dagger}}$. This means that we can efficiently find a defense policy with provable performance guarantees. However, such a defense policy may not be *optimally robust* in that its performance lower bound would not match the optimal one. We now turn to computational complexity challenges in deriving optimally robust defense policies: the next theorem provides a hardness result for $\mathcal{C} = \{c_p \text{ s.t. } p \in [1, \infty)\}$, which admits guarantees on the optimal solution to (P2a).

**Theorem 4.5.** *For $\mathcal{C} = \{c_p \text{ s.t. } p \in [1, \infty)\}$, it is NP-hard to determine whether the optimal value of the optimization problem* (P2a) *is greater than or equal to $\widehat{\rho}^{\pi_\dagger}$.*

Given that even for this natural choice of cost functions, $\mathcal{C} = \{c_p \text{ s.t. } p \in [1, \infty)\}$, the problem of finding optimally robust policies is computationally hard, in the next section we focus on $\ell_2$ attack cost that is known to the agent, i.e., $\mathcal{C} = \{c_2\}$. In Appendix, we provide similar analysis for $\ell_1$ attack cost, i.e., $\mathcal{C} = \{c_1\}$.

## 5 Characterization Results for $\ell_2$ Attack Cost

In this section, we consider the attack cost function $c_2$, and assume that it is known to the agent ($\mathcal{C} = \{c_2\}$). In the first part, we focus on characterization results for the case when the attack parameter $\epsilon_\dagger$ is known to the agent, i.e., the optimization problem (P2a). In the second part, we focus on the optimization problem (P2b), and generalize the results from the first part to the unknown attack parameter setting. The proofs of our formal results can be found in Appendix. Figure 3 provides intuition behind the results of this section.

---

[2]Stochastic policies are not necessary in this case since we can think of this problem as searching for an optimal policy over a truncated actions space (because actions $\pi_\dagger(s)$ are not admissible), so an optimal deterministic policy always exists.

In particular, it illustrates attack and defense strategies for a single-state MDP with action set $\{a_1, ... a_7\}$ and the $\ell_2$ cost function. We refer the reader to Appendix for more details.

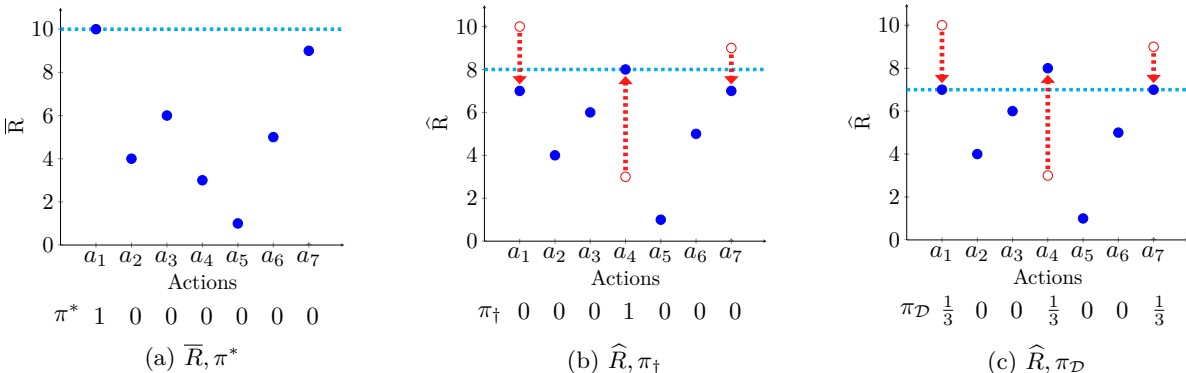

Figure 3: A single-state environment environment with 7 actions and state $s$. In each figure, the denoted policy is uniform over actions on or above the dashed line. **(a)** shows $\overline{R}$ and $\pi^*$. Here, the optimal policy selects action $a_1$. **(b)** shows $\widehat{R}$ and target policy $\pi_\dagger$ with $\epsilon_\dagger = 1$. Here, the target policy selects action $a_4$. $\widehat{R}$ is obtained by solving (P1) (or (P1')) with the $\ell_2$ cost function and $R' = \overline{R}$. In this case, the attack only modifies the rewards of three actions, $a_1$, $a_4$, and $a_7$. **(c)** shows $\widehat{R}$ and $\pi_\mathcal{D}$ with $\epsilon_\mathcal{D} = 2$. The defense strategy only sees poisoned rewards $\widehat{R}$. It first calculates the optimal action and the set of all second-best actions under $\widehat{R}$, in this case $\{a_1, a_7\}$. If the reward of the second-best actions are no worse than $\epsilon_\mathcal{D}$, they form the set $\Theta^\epsilon = \{(s, a_1), (s, a_7)\}$ or simply $\Theta^\epsilon_s = \{a_1, a_7\}$. We show in Appendix that the defense strategy, i.e., the solution to (P2b), selects an action uniformly at random from the set $\{\pi_\dagger(s)\} \cup \Theta^\epsilon_s = \{a_1, a_4, a_7\}$. As we show in this section, the expected reward of the defense policy under $\overline{R}$ is at least as much as its expected reward under $\widehat{R}$, which can be easily verified from the figures: in this case, both are equal to $22/3$.

## 5.1 Known Parameter Setting

We begin by analyzing the optimization problem (P2a). Denote by $\Theta^\epsilon$ state-action pairs $(s, a)$ for which the difference between $\widehat{\rho}^{\pi_\dagger}$ and $\widehat{\rho}^{\pi_\dagger\{s;a\}}$ is equal to $\epsilon$, i.e., $\Theta^\epsilon = \left\{ (s, a) : \widehat{\rho}^{\pi_\dagger\{s;a\}} - \widehat{\rho}^{\pi_\dagger} = -\epsilon \right\}$.[3] For the results of this section, $\Theta^\epsilon$ with $\epsilon = \epsilon_\dagger$ plays a critical role—as we show in our formal analysis, it characterizes the feasible set of the optimization problem (P2a). In particular, in our analysis we show that $\widehat{R}$ is the solution to the attack problem for an underlying reward function $R$, i.e., $\widehat{R} = \mathcal{A}(R, \pi_\dagger, \epsilon_\dagger)$, if and only if $R$ can be expressed as

$$R = \widehat{R} + \sum_{(s,a) \in \Theta^{\epsilon_\dagger}} \alpha_{s,a} \cdot \left( \psi^{\pi_\dagger\{s;a\}} - \psi^{\pi_\dagger} \right),$$

with $\alpha_{s,a} \geq 0$. To see the importance of this result, let us instantiate $R = \overline{R}$ and calculate $\overline{\rho}^\pi$:

$$\overline{\rho}^\pi = \left\langle \psi^\pi, \overline{R} \right\rangle = \left\langle \psi^\pi, \widehat{R} \right\rangle + \sum_{(s,a) \in \Theta^{\epsilon_\dagger}} \alpha_{s,a} \cdot \left\langle \psi^{\pi_\dagger\{s;a\}} - \psi^{\pi_\dagger}, \psi^\pi \right\rangle.$$

When the occupancy measure of $\pi_\mathcal{D}$ is positively aligned with vectors $\psi^{\pi_\dagger\{s;a\}} - \psi^{\pi_\dagger}$, the performance of $\pi_\mathcal{D}$ under the original reward function $\overline{R}$ is at least $\widehat{\rho}^{\pi_\mathcal{D}} = \left\langle \psi^{\pi_\mathcal{D}}, \widehat{R} \right\rangle$. Hence, constraining $\pi_\mathcal{D}$ to satisfy $\left\langle \psi^{\pi_\dagger\{s;a\}} - \psi^{\pi_\dagger}, \psi^{\pi_\mathcal{D}} \right\rangle \geq 0$ for all $s, a \in \Theta^{\epsilon_\dagger}$ yields a guarantee on the score $\overline{\rho}^{\pi_\mathcal{D}}$, i.e., $\overline{\rho}^{\pi_\mathcal{D}} \geq \widehat{\rho}^{\pi_\mathcal{D}}$. These insights are formalized by Theorem 5.1, which also describes a procedure for solving (P2a). In Appendix, we provide intuition behind our analysis using a special case of our setting.

---

[3]In practice, $\Theta^\epsilon$ should be calculated with some tolerance due to numerical imprecision (See Section 6).

**Theorem 5.1.** *Consider the following optimization problem parameterized by $\epsilon$:*

$$\max_{\psi \in \Psi} \left\langle \psi, \widehat{R} \right\rangle \quad \text{s.t.} \quad \left\langle \psi^{\pi_\dagger\{s;a\}} - \psi^{\pi_\dagger}, \psi \right\rangle \geq 0 \quad \forall s, a \in \Theta^\epsilon. \tag{P3}$$

*For $\epsilon = \epsilon_\dagger$, this optimization problem is always feasible, and its optimal solution $\psi_{\max}$ specifies an optimal solution to the optimization problem* (P2a) *for $\mathcal{C} = \{c_2\}$ with*

$$\pi_{\mathcal{D}}(a|s) = \frac{\psi_{\max}(s, a)}{\sum_{a'} \psi_{\max}(s, a')}. \tag{2}$$

*The score of $\pi_{\mathcal{D}}(a|s)$ is lower bounded by $\overline{\rho}^{\pi_{\mathcal{D}}} \geq \widehat{\rho}^{\pi_{\mathcal{D}}}$.*

In addition to providing a characterization of the solution to (P2a), the above theorem provides an efficient algorithm for finding this solution using linear programming. As we discuss in Appendix D.3, the set of vectors $\psi^{\pi_\dagger}$ and $\psi^{\pi_\dagger\{s;a\}}$ can be precomputed since the agent is given a poisoned model $\widehat{M}$ (and hence, knows the transition probabilities); for any policy $\pi$, the state-action occupancy measure $\psi^\pi$ satisfies $\psi^\pi(s, a) = \mu^\pi(s) \cdot \pi(a|s)$ where the state occupancy measure $\mu^\pi$ is the unique solution to the following Bellman identity

$$\mu^\pi(s) = (1 - \gamma)\sigma(s) + \gamma \sum_{\tilde{s}, \tilde{a}} \mu^\pi(\tilde{s})\pi(\tilde{a}|\tilde{s})P(\tilde{s}, \tilde{a}, s).$$

In addition, the set of valid occupancy measures $\Psi$ is the set of all vectors $\psi \in \mathbb{R}^{|S| \cdot |A|}$ satisfying the following Bellman flow constraints

$$\forall s : \sum_a \psi(s, a) = (1 - \gamma)\sigma(s) + \sum_{\tilde{s}, \tilde{a}} \gamma \cdot P(\tilde{s}, \tilde{a}, s) \cdot \psi(\tilde{s}, \tilde{a}),$$

$$\forall (s, a) : \psi(s, a) \geq 0.$$

Therefore, since occupancy measures $\psi^{\pi_\dagger\{s;a\}}$ and $\psi^{\pi_\dagger}$ can be precomputed, the optimization problem (P3) can be efficiently solved using linear programming. In other words, computing *optimally robust* defense policy is computationally tractable. In order to calculate each $\mu^\pi$, we need to solve a $|S| \times |S|$ system of linear equations, which requires $|S|^3$ operations in the worst case. Therefore, finding each $\psi^\pi$ for $\pi \in \{\pi_\dagger\} \cup \{\pi_\dagger\{s; a\} : a \neq \pi_\dagger(s)\}$ requires at most $|S|^4 \cdot |A|$ operations. The linear program (P3) takes $O(|S|^4 \cdot |A|^4)$ time in the worst case as it has $|S| \cdot |A|$ constrains and variables. Therefore, the overall complexity of the approach is $O(|S|^4 \cdot |A|^4)$ in the worst case.

Theorem 5.1 also provides a performance guarantee of the defense policy w.r.t. the true reward function, i.e., $\overline{\rho}^{\pi_{\mathcal{D}}} \geq \widehat{\rho}^{\pi_{\mathcal{D}}}$. Such a bound is important in practice since it provides a *certificate* of the worst-case performance under the true reward function $\overline{R}$, even though the agent can only optimize over $\widehat{R}$. In contrast to the lower bound in Theorem 4.4, the lower bound in Theorem 5.1 is optimal.

## 5.2 Unknown Parameter Setting

In this subsection, we focus on the optimization problem (P2b). First, note the structural difference between (P2a) and (P2b). In the former case, $\epsilon_\dagger$ is given, and hence, the defense can infer possible values of $\overline{R}$ by solving an inverse problem to the attack problem (P1). In particular, we know that the original reward function $\overline{R}$ has to be in the set $\{R : \widehat{R} \in \mathcal{A}(R, \pi_\dagger, \epsilon_\dagger)\}$. In the latter case, $\epsilon_\dagger$ is not known, and instead we use parameter $\epsilon_{\mathcal{D}}$ as an upper bound on $\epsilon_\dagger$. We distinguish two cases:

- *Overestimating Attack Parameter*: If $\epsilon_\dagger \leq \epsilon_{\mathcal{D}}$, then we know that $\overline{R}$ is in the set $\{R : \widehat{R} \in \mathcal{A}(R, \pi_\dagger, \epsilon)$ s.t. $0 < \epsilon \leq \epsilon_{\mathcal{D}}\}$. Note that this set is a super-set of $\{R : \widehat{R} \in \mathcal{A}(R, \pi_\dagger, \epsilon_\dagger)\}$, which means that it is less informative about $\overline{R}$.
- *Underestimating Attack Parameter*: If $\epsilon_\dagger > \epsilon_{\mathcal{D}}$, then the set $\{R : \widehat{R} \in \mathcal{A}(R, \pi_\dagger, \epsilon)$ s.t. $0 < \epsilon \leq \epsilon_{\mathcal{D}}\}$ will have only a single element, i.e., $\widehat{R}$. In other words, this set typically contains no information about $\overline{R}$.

We analyze these two cases separately, first focusing on the former one.

### 5.2.1 Overestimating Attack Parameter

When $\epsilon_{\mathcal{D}} \geq \epsilon_\dagger$, our formal analysis builds on the one presented in Section 5.1, and we highlight the main differences. Given that $\epsilon_\dagger$ is not exactly known, we cannot directly operate on the set $\Theta^{\epsilon_\dagger}$. However, since $\epsilon_{\mathcal{D}}$ upper bounds $\epsilon_\dagger$, the defense can utilize the procedure from the previous section (Theorem 5.1) with appropriately chosen $\epsilon$ to solve (P2b) as we show in the following theorem.

**Theorem 5.2.** *Assume that $\epsilon_{\mathcal{D}} \geq \epsilon_\dagger$, and define $\widehat{\epsilon} = \min_{s,a \neq \pi_\dagger(s)} \left[\widehat{\rho}^{\pi_\dagger} - \widehat{\rho}^{\pi_\dagger\{s;a\}}\right]$. Then, the optimization problem* (P3) *with $\epsilon = \min\{\epsilon_{\mathcal{D}}, \widehat{\epsilon}\}$ is feasible and its optimal solution $\psi_{\max}$ identifies an optimal policy $\pi_{\mathcal{D}}$ for the optimization problem* (P2b) *with $\mathcal{C} = \{c_2\}$ via Equation (2). This policy $\pi_{\mathcal{D}}$ satisfies $\overline{\rho}^{\pi_{\mathcal{D}}} \geq \widehat{\rho}^{\pi_{\mathcal{D}}}$.*

To interpret the bounds, let us consider three cases:

- $\overline{R} \neq \widehat{R}$: If the attack indeed poisoned $\overline{R}$, then the smallest $\epsilon' \in (0, \epsilon_{\mathcal{D}}]$ such that $\Theta^{\epsilon'} \neq \emptyset$ corresponds to $\epsilon_\dagger$. In this case, it turns out that $\epsilon_\dagger = \widehat{\epsilon}$, and somewhat surprisingly, the defense policies of (P2a) and (P2b) coincide. (Note that this analysis assumes that $\epsilon_{\mathcal{D}} \geq \epsilon_\dagger$.)
- $\overline{R} = \widehat{R}$ and $\epsilon_{\mathcal{D}} < \widehat{\epsilon}$: This corresponds to the case when the attack did not poison $\overline{R}$ and there is no $\epsilon' \in (0, \epsilon_{\mathcal{D}}]$ such that $\Theta^{\epsilon'} \neq \emptyset$. In this case, it turns out that the optimal solution to the optimization problem (P2b) is $\pi_{\mathcal{D}} = \pi_\dagger$ (indeed $\pi_\dagger$ is uniquely optimal under $\overline{R}$).
- $\overline{R} = \widehat{R}$ and $\epsilon_{\mathcal{D}} \geq \widehat{\epsilon}$: This corresponds to the case when the attack did not poison $\overline{R}$ and there is $\epsilon' \in (0, \epsilon_{\mathcal{D}}]$ such that $\Theta^{\epsilon'} \neq \emptyset$. In fact, $\widehat{\epsilon}$ is the smallest such $\epsilon'$. In this case, it turns out that, in general, the optimal solution to the optimization problem (P2b) is $\pi_{\mathcal{D}} \neq \pi_\dagger$, even though $\pi_\dagger$ is uniquely optimal under $\overline{R}$.

These three cases also showcase the importance of choosing $\epsilon_{\mathcal{D}}$ that is a good upper bound on $\epsilon_\dagger$. When $\overline{R} = \widehat{R}$, the agent should select $\epsilon_{\mathcal{D}}$ that is strictly smaller than $\widehat{\epsilon}$. On the other hand, when $\overline{R} \neq \widehat{R}$, the agent should select $\epsilon_{\mathcal{D}} \geq \widehat{\epsilon}$, as it will be apparent from the result of the next subsection, in particular Theorem 5.3. While the agent knows $\widehat{\epsilon}$, it does not know if $\overline{R} = \widehat{R}$ or $\overline{R} \neq \widehat{R}$.

### 5.2.2 Underestimating Attack Parameter

In this subsection, we analyze the case when $\epsilon_{\mathcal{D}} < \epsilon_\dagger$. We first state our result, and then discuss its implications.

**Theorem 5.3.** *If $\epsilon_\dagger > \epsilon_{\mathcal{D}}$, then $\pi_{\mathcal{D}} = \pi_\dagger$ is the unique solution of the optimization problem* (P2b) *with $\mathcal{C} = \{c_2\}$.*

Therefore, together with Theorem 5.2, Theorem 5.3 is showing the importance of having a good prior knowledge about the attack parameter $\epsilon_\dagger$. In particular:

- When the attack did not poison the reward function (i.e., $\widehat{R} = \overline{R}$), overestimating $\epsilon_\dagger$ implies that $\pi_{\mathcal{D}}$ might not be equal to $\pi_\dagger$ for larger values of $\epsilon_{\mathcal{D}}$, even though $\pi_\dagger$ is uniquely optimal under $\overline{R}$. This can have a detrimental effect since in this case $\overline{\rho}^{\pi_{\mathcal{D}}} < \overline{\rho}^{\pi_\dagger} = \overline{\rho}^{\pi^*}$.
- When the attack did poison the reward function $\overline{R}$ (i.e., $\widehat{R} \neq \overline{R}$), underestimating $\epsilon_\dagger$ implies $\pi_{\mathcal{D}} = \pi_\dagger$, but $\pi_\dagger$ might be suboptimal. In this case, the defense policy does not limit the negative influence of the attack at all, i.e., $\overline{\rho}^{\pi_{\mathcal{D}}} = \overline{\rho}^{\pi_\dagger} \leq \overline{\rho}^{\pi^*}$.

We further discuss nuances to selecting $\epsilon_{\mathcal{D}}$ in Section 7.

## 6 Experimental Evaluation

In this section we evaluate our defense strategy in an experimental setting in order to better understand its efficacy and robustness. We focus on the setting from Section 5.2: $c_2$ attack cost functions, which is known to the agent, i.e., $\mathcal{C} = \{c_2\}$, with an unknown attack parameter $\epsilon_\dagger$.

Given the results in Section 5 (Theorems 5.2 and 5.3), we use the linear programming formulation (P3) together with the CVXPY solver Diamond & Boyd (2016); Agrawal et al. (2018) for calculating the solution to the defense optimization problem (P2b). In the experiments, due to limited numerical precision, $\Theta^\epsilon$ is calculated

with a tolerance parameter, set to $10^{-4}$ by default.[4]. In other words, $\Theta^\epsilon = \{(s,a) : |\widehat{\rho}^{\pi_\dagger} - \widehat{\rho}^{\pi_\dagger\{s;a\}} - \epsilon| \leq 10^{-4}\}$.

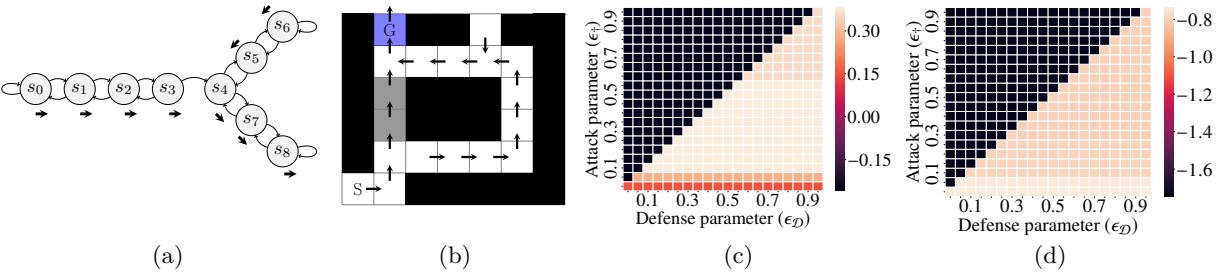

Figure 4: Experimental environments: Figures **(a)** and **(b)** show the Navigation and Grid world environment respectively while figrues **(c)** and **(d)** show $\overline{\rho}^{\pi_\mathcal{D}}$ in these environments. For comparison, in the navigation environment, $\overline{\rho}^{\pi_\dagger} = -0.26$ and $\overline{\rho}^{\pi^*} = 0.45$ while in the grid world environment, $\overline{\rho}^{\pi_\dagger} = -1.75$ and $\overline{\rho}^{\pi^*} = -0.70$.

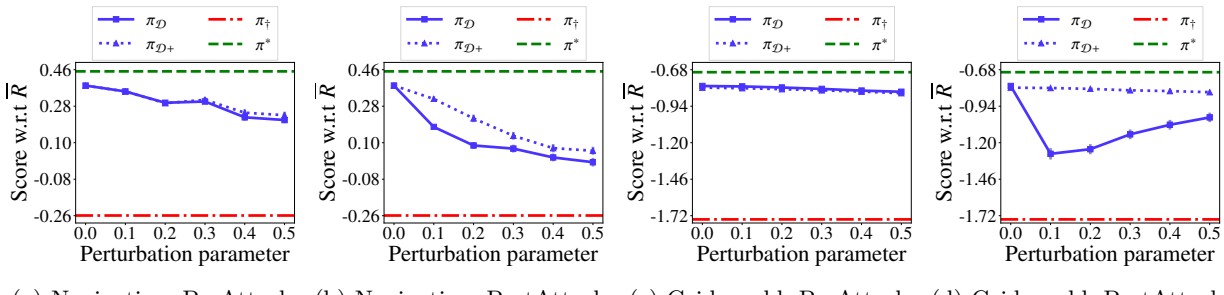

(a) Navigation, PreAttack  (b) Navigation, PostAttack  (c) Grid world, PreAttack  (d) Grid world, PostAttack

Figure 5: Robustness of the defense policy against random perturbation. Results are based on average of 100 runs for each data point. Error bars around the data points indicate standard error.

**Navigation environment**. Our first environment, shown in Figure 4a is the Navigation environment taken from Rakhsha et al. (2021). The environment has 9 states and 2 possible actions. The reward function is action independent and has the following values: $\overline{R}(s_0,.) = \overline{R}(s_1,.) = \overline{R}(s_2,.) = \overline{R}(s_3,.) = -2.5$, $\overline{R}(s_4,.) = \overline{R}(s_5,.) = 1$ and $\overline{R}(s_6,.) = \overline{R}(s_7,.) = \overline{R}(s_8,.) = 0$. When the agent takes an action, it will successfully navigate in the direction shown by the arrows with probability 0.9; otherwise, the next state will be sampled uniformly at random. The bold arrows in the figure indicate the attacker's target policy. The initial state is $s_0$ and the discounting factor $\gamma$ equals 0.99.

**Grid world environment.** For our second environment, shown in Figure 4b, we use the grid world environment from Ma et al. (2019) with slight modifications in order to ensure ergodicity — we add a 10% failure probability to each action, sampling the next state randomly in case of failure. The environment has 18 states and 4 actions: *up*, *down*, *right* and *left*. The white, gray and blue cells in the figure represent the states and the black cells represent walls. In the white and gray states, the agent will attempt to go in the direction specified by its action if there is a neighboring state in that direction. If there is no such state, the agent will attempt to stay in its own place. In the blue state $G$, the agent will attempt to stay in its own place regardless of the action taken. In all states, each attempt will succeed with probability 0.9; with probability 0.1, the next state will be sampled uniformly at random. In the gray and white states, the agent's reward is a function of the state it is attempting to visit. Attempting to visit a gray, white and blue state will yield a reward of $-10$, $-1$ and 2 respectively. If the agent is in a blue state, it will always receive a reward of 0. The bold arrows in the figure specify the attacker's target policy. The initial state is $S$ and the discounting factor $\gamma$ equals 0.9.

**Policy score for different values of parameters.** We first analyze the score of our defense policy in

---

[4]The value was chosen because the CVXPY solver uses a precision of $10^{-5}$.

both environments with different values of $\epsilon_\dagger$ and $\epsilon_\mathcal{D}$. For comparison, we also report the scores of the target policy $(\pi_\dagger)$ and the optimal policy $(\pi^*)$. The results are shown in Figures 4c and 4d. As seen in the figures, as long as $\epsilon_\mathcal{D} \geq \epsilon_\dagger$, our defense policy significantly improves the agent's score compared to $\pi_\dagger$.

**Robustness to perturbations.** We now analyze our algorithm's robustness towards uncertainties in the reward functions used by the attacker and the defender. For our first experiment, which we call PreAttack, we randomly perturb the attacker's input. In particular, the input to the defender's optimization problem $\widehat{R}$ is sampled from $\mathcal{A}(c_2, \overline{R} + \mathcal{N}(0, \sigma^2 I), \pi_\dagger, \epsilon_\dagger)$ where $I$ is the identity matrix, $\mathcal{N}$ denotes the multivariate normal distribution and $\sigma$ is the perturbation parameter varied in the experiment. For our second experiment, called PostAttack, we randomly perturb the reward vector after the attack, sampling the defender's input from $\mathcal{A}(c_2, \overline{R}, \pi_\dagger, \epsilon_\dagger) + \mathcal{N}(0, \sigma^2 I)$. In both experiments we use $\epsilon_\dagger = 0.1$ and $\epsilon_\mathcal{D} = \infty$. As explained below, when calculating $\Theta^\epsilon$, we also experiment with a larger tolerance parameter of $10^{-1}$, denoting the defense policy in this case with $\pi_{\mathcal{D}+}$.

The results can be seen in Figure 5. As seen in the figures, our defense policy $\pi_\mathcal{D}$ consistently improves on the baseline obtained with no defense (i.e, $\pi_\dagger$). It is also clear that the PostAttack perturbations have a greater negative impact on our defense strategy's score. Results for $\pi_{\mathcal{D}+}$ indicate that this is due to random perturbations prohibiting our algorithm from identifying all of the elements in $\Theta^\epsilon$. While having a higher tolerance parameter helps with robustness, it can also lead to a lower performance when there is no noise because $\Theta^\epsilon$ would falsely include additional elements. We leave choosing the tolerance parameter in a more systematic way for future work.

# 7   Concluding Discussions

In this paper, we introduced an optimization framework for designing defense strategies against reward poisoning attacks that change an agent's reward structure in order to steer the agent to adopt a target policy. We analyzed the utility of using such defense strategies, providing characterization results and provable guarantees on their performance. Moving forward we see several interesting future research directions.

**Beyond the worst-case utility.** In this paper, we defined the defense objective as the maximization of the agent's worst-case utility. While this is a sensible objective, there are other objectives that one could analyze. For example, instead of focusing on the absolute performance, one can try to optimize performance relative to the target policy. Notice that this is a somewhat different, and possibly a weaker goal, given that the target policy can have arbitrarily bad utility under $\overline{R}$. Additionally, one could study the agent's *sub-optimality gap*, i.e., the difference between its score and the score of the optimal policy $\pi^*$, and compare it to the sub-optimality gap of the target policy. Going back to the example in Figure 3, we can see that the defense strategy can significantly reduce the suboptimality gap relative to not having a defense (in this case, from $10 - 3 = 7$ to $10 - 22/3 \approx 2.7$). However, when the original reward function already forces the target policy, i.e., when $\overline{R} = \widehat{R}$, the suboptimality gap of the target policy is equal to 0. In Figure 3, this happens if the target policy is action $a_1$. Nevertheless, in this case, the suboptimality gap of the defense strategy would also be low, equal to $1/2$ (since the defense would randomly select $a_1$ or $a_7$), and generally upper bounded by $\epsilon_\mathcal{D} = 2$ if the rewards of sub-optimal actions $\{a_2, ..., a_7\}$ were different. Providing a full theoretical treatment for the general setting is an interesting research direction.

**Informed prior.** We did not model prior knowledge that an agent might have about the attacker or the underlying reward function. In practice, the agent may have some information about the underlying true reward function. Incorporating such considerations calls for a Bayesian approach that could improve the agent's defense by, e.g., ruling out implausible candidates for $\overline{R}$ in the agent's inference of $\overline{R}$ given $\widehat{R}$.

**Selecting $\epsilon_\mathcal{D}$ and non-oblivious attacks.** The results in Section 5.2 indicate that choosing good $\epsilon_\mathcal{D}$ is important for having a functional defense. In practice, a selection procedure for $\epsilon_\mathcal{D}$ should take into account the cost that the attacker has for different choices of $\epsilon_\dagger$, as well as game-theoretic considerations: attacks might not be *oblivious* in that the strategy for selecting $\epsilon_\dagger$ might depend on the strategy for selecting $\epsilon_\mathcal{D}$. Namely, a direct consequence of Theorem 5.2 is that the attack optimization problem (P1) can successfully achieve its goal if it chooses a large value of $\epsilon_\dagger$ to large enough values. However, the cost of the attack also grows with $\epsilon_\dagger$, so the attack (if strategic) also needs to reason about $\epsilon_\mathcal{D}$ when selecting $\epsilon_\dagger$. We leave the full game-theoretic characterization of the parameter selection problem for the future work.

**Unknown-model and scalability**. Following prior work, we focused on attacks and defenses that have

access to an accurate transition model and operate in tabular settings. Given Theorems 5.1, 5.2 and 5.3, the defense problem can be solved using the linear program (P3), which is similar in size to the attack optimization problem (P1'). As such, we expect our method to be computationally scalable as long as the attack optimization problem can be solved.

For many RL problems however, the transition and reward functions are not known and the MDP is too large to be modeled in tabular settings. RL solutions for such problems typically rely on function approximation. For future work, it would be interesting to study the problem, for both attack and defense, in this more realistic scenario. An immediate question is how the attack problem would need to change in order to generalise to this setting. In terms of defense, while the new setting would likely pose new challenges, we expect the general min-max formulation in Section 3, as well as the techniques used for solving the optimization problems (P2a) and (P2b), to remain useful.

## Acknowledgments

The authors would like to thank the anonymous reviewers for their valuable comments and suggestions. This research was, in part, funded by the Deutsche Forschungsgemeinschaft (DFG, German Research Foundation) – project number 467367360.

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

## A   List of Appendices

In this section we provide a brief description of the content provided in the appendices of the paper.

- Appendix B provides an intuition of our results for the $\ell_2$ attack cost using special MDPs in which the agent's actions do not affect the transition dynamics. The proofs of the results presented in this Appendix can be found in Appendix I.
- Appendix C provides additional details regarding the experiments.
- Appendix D contains some background on reward poisoning attacks, and a brief overview of the MDP properties that are important for proving our formal results.
- Appendix E contains characterization results for the attack optimization problem (P1).
- Appendix F contains proofs of the formal results in Section 5.
    - The proof of Theorem 5.1 is in Section F.1.
    - The proof of Theorem 5.2 is in Section F.2.
    - The proof of Theorem 5.3 is in Section F.3.

- Appendix G contains the proofs for the results in Section 4 relating to Guarantees on values. The appendix also includes an optimization framework for solving the defense optimization problem (P2a) for the $\ell_1$ norm as well as additional characterization results for the defense optimization problem for more general cost functions which are used for proving the complexity results in Section 4.
  - The characterization result for the defense optimization problem with for the $\ell_1$ norm is provided in Section G.1.
  - Proof of Theorem 4.4 is provided in Section G.2.
  - Additional characterization results for the defense optimization problem are provided in Section G.3.
  - Proof of Proposition 4.2 is provided in Section G.4
  - Proof of Theorem 4.3 is provided in Section G.5.
- Appendix H contains the proofs of the formal results in Section 4 relating to the computational complexity of the defense optimization problem as well as the computational complexity result for the $\ell_\infty$ attack cost.
  - The hardness result for the $\ell_\infty$ norm is provided in Section H.1.
  - Proof of Theorem 4.5 is provided in Section H.2
- Appendix I contains a formal treatment of the results presented in Appendix B

## B    Intuition of Results using Special MDPs

In this Appendix, we describe characterization results on the $\ell_2$ attack cost for special MDPs, in which the agent's actions do not affect the transitions, that is, we assume that

$$P(s, a, s') = P(s, a', s') \quad \forall s, a, a', s'. \tag{3}$$

Variants of the above condition have been studied in the literature (e.g., Szepesvári (1997); Dimitrakakis et al. (2017); Sutton & Barto (2018); Radanovic et al. (2019); Ghosh et al. (2020)). Note that this assumption implies that any two policies $\pi$ and $\pi'$ have equal state occupancy measures, so we simplify the notation by denoting $\mu = \mu^\pi = \mu^{\pi'}$.

While the results from the previous sections incorporate this special case, we study this setting because: i) the optimal solutions to the defense problem have a simple form, enabling us to provide intuitive explanations of our main results from the previous sections, ii) using this setting, we show a tightness result for Theorem 5.2.

A more formal exposition of our results for this setting inlcuding the proofs can be found in Appendix I.

### B.1    Optimal Defense Policy

In this subsection, we provide the intuition behind defense policies for the unknown parameter setting with $\epsilon_\mathcal{D} \geq \epsilon_\dagger$ (Section 5.2.1). The key point about the assumption in Equation (3) is that it allows us to consider each state separately in the defense optimization problems. In particular, it can be shown that the optimization problem (P3) is equivalent to solving $|S|$ optimization problems of the form

$$\max_{\pi(\cdot|s) \,\in\, \mathcal{P}(A)} \left\langle \pi(\cdot|s), \widehat{R}(s, \cdot) \right\rangle \tag{P3b}$$

$$\pi(a|s) \geq \pi\big(\pi_\dagger(s) \mid s\big) \quad \forall a \in \Theta_s^\epsilon,$$

where $\Theta_s^\epsilon = \{a : \widehat{R}(s, a) - \widehat{R}(s, \pi_\dagger(s)) = -\frac{\epsilon}{\mu(s)}\}$.[5] If we instantiate Theorem 5.2 for special MDPs by putting $\epsilon = \min\{\epsilon_\mathcal{D}, \widehat{\epsilon}\}$, the set $\Theta_s^\epsilon$ has an intuitive description: it is the set of all "second-best" actions (w.r.t $\widehat{R}$) in state $s$ such that their poisoned reward is greater than or equal to $\widehat{R}(\pi_\dagger(s)) - \frac{\epsilon}{\mu(s)}$. It turns out that the defense policy for state $s$ selects an action uniformly at random from the set $\Theta_s^\epsilon \cup \{\pi_\dagger(s)\}$. In other words, the defense policy $\pi_\mathcal{D}$ is given by:

$$\pi_\mathcal{D}(a|s) = \begin{cases} \frac{1}{|\Theta_s^\epsilon|+1} & \text{if } a \in \Theta_s^\epsilon \cup \{\pi_\dagger(s)\} \\ 0 & \text{otherwise} \end{cases} .$$

---

[5]See Lemma 8 in Banihashem et al. (2022).

To see why, note that the objective in (P3b) only improves as we put more probability on selecting $\pi_\dagger(s)$ (since $\pi_\dagger(s)$ is optimal under $\widehat{R}$). However, the constraints in (P3b) require that the selection probability of any action in $\Theta_s^\epsilon$ has to be at least as high as the selection probability of $\pi_\dagger(s)$, which in turn give us the uniform at random selection rule. Figure 3 illustrates attack and defense policies for special MDPs using a single-state MDP with action set $\{a_1, ... a_7\}$. To obtain defense policy $\pi_\mathcal{D}$, we can solve the optimization problem (P3b), which implies that $\pi_\mathcal{D}$ should select an action uniformly at random from the set $\{\pi_\dagger(s)\} \cup \Theta_s^\epsilon = \{a_1, a_4, a_7\}$

## C    Additional Details Regarding Experiments

In this section we provide additional details regarding the experiments. The source code for our experiments, as well as instructions for replicating our results can be found in the Supplementary Material.

### C.1    Implementation details

Both the attacker optimization problem (P1), and the defense optimization problems (P2a) and (P2b) are convex since, by Theorems 5.1 and 5.2, the defense optimization problem reduces to the linear program (P3). We use CVXPY to calculate their solutions. The code for solving these optimization problems can be found in the file `MDP.py`. The specific functions used for solving these problems are as follows:

- The function `attack` implements the attacker's optimization problem (P1). As explained in the main text, this is done by solving (P1') since (P1') is equivalent to (P1).
- The function `defend_known` implements the optimization problem (P2a). As explained in Section 6, the tolerance parameter is set to $10^{-4}$ (default value).
- The function `defend_unknown` implements the optimization problem (P2b).

### C.2    Running time

Following prior work Rakhsha et al. (2021), to test the running times, we use the chain environment from Rakhsha et al. (2021), but with different number of states (additional states are added between $s_2$ and $s_3$, and the corresponding transitions and rewards are defined analogously to those for $s_2$). The attack and defense parameters are set to $\epsilon_\dagger = 0.1$ and $\epsilon_\mathcal{D} = 0.2$. Table 2 shows the average running times (across 10 runs) of the attack optimization problem (P1') and the defense optimization problem (P2b) for different sizes of the chain environment.

It should be noted that the attack and defense optimization problems are similar in size, both solve a problem with at most $|S| \cdot (|A| - 1)$ constraints on $\mathbb{R}^{|S| \cdot |A|}$. However, solving the defense problem takes more time, partly because $\pi_\dagger$, $\widehat{\epsilon}$ and $\Theta^\epsilon$ need to be identified before (P3) can be solved.

The machine used for obtaining these results is a Macbook Pro personal computer with 4 Gigabytes of memory and a 2.4 GHz Intel Core i5 processor.

## D    Background and Additional MDP Properties

In this section we briefly outline the background and MDP properties that we utilize in our proofs.

### D.1    Reward Poisoning Attacks

In this section, we provide some background on the cost-efficient reward poisoning attacks, focusing on the results from Rakhsha et al. (2021).

The setting studied in Rakhsha et al. (2021) incorporates both the average and the discounted reward optimality criteria in a discrete-time Markov Decision Process (MDP), with finite state and action spaces. Our MDP setting is equivalent to their MDP setting under the discounted reward optimality criteria. This criteria can be specified by score $\rho$. As defined in the main text, *score* $\rho^\pi$ of policy $\pi$ is the total expected

| |S| Problem | Attack | Defense |
|---|---|---|
| 4 | $0.01\text{s} \pm 0.5\text{ms}$ | $0.05\text{s} \pm 1.6\text{ms}$ |
| 10 | $0.01\text{s} \pm 0.2\text{ms}$ | $0.09\text{s} \pm 1.5\text{ms}$ |
| 20 | $0.01\text{s} \pm 0.1\text{ms}$ | $0.17\text{s} \pm 4.8\text{ms}$ |
| 30 | $0.02\text{s} \pm 2.0\text{ms}$ | $0.27\text{s} \pm 9.7\text{ms}$ |
| 50 | $0.04\text{s} \pm 6.8\text{ms}$ | $0.56\text{s} \pm 34.6\text{ms}$ |
| 70 | $0.07\text{s} \pm 3.0\text{ms}$ | $1.02\text{s} \pm 69.7\text{ms}$ |
| 100 | $0.13\text{s} \pm 5.4\text{ms}$ | $1.83\text{s} \pm 91.2\text{ms}$ |

Table 2: Run time of the attack and defense optimization problems for the chain environment with varied number of states $|S|$. Reported numbers are average of 10 runs; standard error is shown with $\pm$.

return scaled by factor $1 - \gamma$:

$$\rho^\pi = \mathbb{E}\left[(1 - \gamma)\sum_{t=1}^\infty \gamma^{t-1}R(s_t, a_t)|\pi, \sigma\right],$$

where the state $s_1$ is sampled from the initial state distribution $\sigma$, and subsequent states $s_t$ are obtained by executing policy $\pi$ in the MDP. Actions $a_t$ are sampled from policy $\pi$.

As explained in the main text, the following result is important for our analysis, since it allows us to simplify the optimization problem (P1) into the optimization problem (P1').

**Lemma D.1.** *(Lemma 1 in Rakhsha et al. (2021)) The score of a policy $\pi_\dagger$ is at least $\epsilon_\dagger$ greater than all other deterministic policies if and only if its score is at least $\epsilon_\dagger$ greater than the score of any policy $\pi_\dagger\{s; a\}$. In other words,*

$$\left(\forall \pi \in \Pi^{det}\backslash\{\pi^\dagger\} : \rho^{\pi_\dagger} \geq \rho^\pi + \epsilon_\dagger\right) \iff \left(\forall s, a \neq \pi_\dagger(s) : \rho^{\pi_\dagger} \geq \rho^{\pi_\dagger\{s;a\}} + \epsilon_\dagger\right).$$

*Remark* D.2. As explained in Rakhsha et al. (2021), this lemma implies that the optimization problem (P1) is equivalent to (P1'). Furthermore, the optimization problem is always feasible since any policy can be made optimal with sufficient perturbation of the reward function as formally shown by Rakhsha et al. (2021) and Ma et al. (2019).

### D.2 Overview of Important Quantities

Next, we provide an overview of standard MDP quantities and the quantities introduced in the main text that are important for our analysis.

In addition to score $\rho$, we consider state-action value function, or $Q$-value function, defined as

$$Q^\pi(s, a) = \mathbb{E}\left[\sum_{t=1}^\infty \gamma^{t-1}R(s_t, a_t)|s_1 = s, a_1 = a, \pi\right].$$

In other words, $Q^\pi(s, a)$ is the total expected return when the first state is $s$, the first action is $a$, while subsequent states $s_t$ and actions $a_t$ are obtained by executing policy $\pi$ in the MDP.

We consider two occupancy measures. By $\psi^\pi$ we denote the state-action occupancy measure in the Markov chain induced by policy $\pi$:

$$\psi^\pi(s, a) = \mathbb{E}\left[(1 - \gamma)\sum_{t=1}^\infty \gamma^{t-1}\mathbb{1}\left[s_t = s, a_t = a\right]|\pi, \sigma\right].$$

Given MDP $M$, the set of realizable occupancy measures under any (stochastic) policy $\pi \in \Pi$ is denoted by $\Psi$. Note that the following holds:

$$\rho^\pi = \langle \psi^\pi, R \rangle, \tag{4}$$

where $\langle .,. \rangle$ in the above equation computes a dot product between two vectors of size $|S| \cdot |A|$ (i.e., two vectors in $\mathbb{R}^{|S| \cdot |A|}$). We also denote by $\mu^\pi$ the state occupancy measure in the Markov chain induced policy $\pi \in \Pi$, i.e.:

$$\mu^\pi(s) = \mathbb{E}\left[(1-\gamma)\sum_{t=1}^\infty \gamma^{t-1}\mathbb{1}\left[s_t = s\right] | \pi, \sigma\right].$$

Note that

$$\sum_{s,a} \psi^\pi(s,a) = \sum_s \mu^\pi(s) = 1.$$

State-action occupancy measure and state occupancy measure satisfy

$$\psi^\pi(s,a) = \mu^\pi(s) \cdot \pi(a|s), \tag{5}$$

which for deterministic $\pi$ is equivalent to

$$\psi^\pi(s,a) = \mathbb{1}\left[\pi(s) = a\right] \cdot \mu^\pi(s). \tag{6}$$

Apart from the standard MDP quantities mentioned above, we also mention quantities introduced in the main text. We denote by $\Theta^\epsilon$ state-action pairs $(s,a)$ for which the margin between $\widehat{\rho}^{\pi_\dagger}$ and $\widehat{\rho}^{\pi_\dagger\{s;a\}}$ is equal to $\epsilon$, i.e.:

$$\Theta^\epsilon = \left\{(s,a) : \widehat{\rho}^{\pi_\dagger\{s;a\}} - \widehat{\rho}^{\pi_\dagger} = -\epsilon\right\}, \tag{7}$$

which can be expressed through reward function $\widehat{R}$ using state-action occupancy measures $\psi$:

$$\Theta^\epsilon = \left\{(s,a) : \left\langle \psi^{\pi_\dagger\{s;a\}} - \psi^{\pi_\dagger}, \widehat{R}\right\rangle = -\epsilon\right\}.$$

Finally, quantity $\Gamma^{\{s;a\}}(\pi)$ measures how well the occupancy measure of $\pi$ is aligned with $\psi^{\pi_\dagger\{s;a\}}$ relative to $\psi^{\pi_\dagger}$:

$$\Gamma^{\{s;a\}}(\pi) = \left\langle \psi^{\pi_\dagger\{s;a\}} - \psi^{\pi_\dagger}, \psi^\pi\right\rangle. \tag{8}$$

### D.3 Occupancy Measures as Linear Constraints

In this subsection, we introduce the Bellman flow linear constraints that characterize $\psi^\pi$ and $\mu^\pi$. In order to characterize $\psi^\pi$, we require the following constraints:

$$\forall s : \sum_a \psi(s,a) = (1-\gamma)\sigma(s) + \sum_{\tilde{s},\tilde{a}} \gamma \cdot P(\tilde{s}, \tilde{a}, s) \cdot \psi(\tilde{s}, \tilde{a}). \tag{9}$$

$$\forall (s,a) : \psi(s,a) \geq 0. \tag{10}$$

The importance of these constraints is reflected in the following lemma.

**Lemma D.3.** *(Theorem 2 in Syed et al. (2008)) Let $\psi$ be a vector that satisfies the Bellman flow constraints (9) and (10). Define policy $\pi$ as*

$$\pi(a|s) = \frac{\psi(s,a)}{\sum_{\tilde{a}} \psi(s,\tilde{a})}. \tag{11}$$

*Then $\psi$ is the state-action occupancy measure of $\pi$, in other words $\psi = \psi^\pi$. Conversely, if $\pi \in \Pi$ is a policy with state-action occupancy measure $\psi$ (i.e, $\psi = \psi^\pi$) then $\psi$ satisfies the Bellman flow constraints (9) and (10), as well as Equation (11).*

As for $\mu^\pi$, it is well-known (e.g., see Rakhsha et al. (2021)) that a vector $\mu$ is the state occupancy measure for policy $\pi$ (i.e., $\mu = \mu^\pi$), if and only if

$$\mu(s) = (1-\gamma)\sigma(s) + \gamma \sum_{\tilde{s},\tilde{a}} \mu(\tilde{s})\pi(\tilde{a}|\tilde{s})P(\tilde{s},\tilde{a},s). \tag{12}$$

# E    Attack Characterization Results

## E.1   Characterization results for the $\ell_2$ attack cost

In this section we provide characterization results for the attack optimization problem (P1) for the $\ell_2$ norm, i.e. $\mathcal{C} = \{c_2\}$. We will later use these results for proving the formal results presented in Section 5.1 and Section 5.2. In addition, these results provide intuition for our results about the more general $\ell_p$ norms, which we will discuss in the next sections. In particular, the main result of this appendix is a set of Karush–Kuhn–Tucker (KKT) conditions that characterize the solution to the optimization problem (P1). As we focus on the cost function $c = c_2$ in this section, we will drop the dependence on $c$ in $\mathcal{A}(c, R, \pi_\dagger, \epsilon_\dagger)$.

To compactly express the KKT characterization results, let us introduce state occupancy difference matrix $\mathbf{\Phi} \in \mathbb{R}^{|S| \cdot (|A|-1) \times |S| \cdot |A|}$ as a matrix with rows consisting of the vectors $\psi^{\pi_\dagger\{s;a\}} - \psi^{\pi_\dagger}$ for all neighboring policies $\pi_\dagger\{s;a\}$. Additionally, for all $s, a \neq \pi_\dagger(s)$, we use $\mathbf{\Phi}(s,a)$ to denote the transpose of the row of $\mathbf{\Phi}$ corresponding to $(s,a)$. Note that $\mathbf{\Phi}(s,a)$ is a column vector. In this notation, given Remark D.2 and Equation (4), the optimization problem (P1) is equivalent to

$$\min_R \frac{1}{2} \|R - R'\|_2^2 \tag{P1"}$$
$$\text{s.t.} \quad \mathbf{\Phi} \cdot R \preccurlyeq -\epsilon_\dagger \cdot \mathbf{1},$$

where $\mathbf{1}$ is a $|S| \cdot (|A|-1)$ vector whose each element equal to 1, and $\preccurlyeq$ specifies that the left hand side is element-wise less than or equal to the right hand side. Given this notation, the following lemma states the KKT conditions for a reward function $R$ (i.e., an $|S| \cdot |A|$ vector) to be an optimal solution to the optimization problem (P1).

**Lemma E.1.** *(KKT characterization) $R$ is a solution to the optimization problem* (P1) *if and only if there exists an $|S| \cdot |A|$ vector $\lambda$ such that*

$$\begin{aligned}
(R - R') + \mathbf{\Phi}^T \cdot \lambda &= \mathbf{0} && \text{stationarity,} \\
\mathbf{\Phi} \cdot R + \epsilon_\dagger \cdot \mathbf{1} &\preccurlyeq \mathbf{0} && \text{primal feasibility,} \\
\lambda &\succcurlyeq \mathbf{0} && \text{dual feasibility,} \\
\forall (s, a \neq \pi_\dagger(s)) : \lambda(s,a) \cdot (\mathbf{\Phi}(s,a)^T \cdot R + \epsilon_\dagger) &= \mathbf{0} && \text{complementary slackness,}
\end{aligned}$$

*where $\mathbf{0}$ denotes an $|S| \cdot |A|$ vector whose each element equal to $0$, and likewise, $\mathbf{1}$ denotes an $|S| \cdot |A|$ vector whose each element equal to $0$.*

*Proof.* Since (P1) is always feasible (Remark D.2) and all of the constrains are linear, strong duality holds. Now, the Lagrangian of the optimization problem is equal to

$$\mathcal{L} = \frac{1}{2} \|R - R'\|_2^2 + \lambda^T (\mathbf{\Phi} \cdot R + \epsilon_\dagger \cdot \mathbf{1}),$$

and taking the gradient with respect to $R$ gives us

$$\nabla_R \mathcal{L} = (R - R') + \mathbf{\Phi}^T \cdot \lambda.$$

The statement then follows by applying the standard KKT conditions. □

*Remark* E.2. (Uniqueness) The solution to the optimization problem (P1) is unique since the objective $\frac{1}{2}\|R - R'\|_2^2$ is strongly convex.

The above lemma, has the following important consequence.

**Lemma E.3.** *Reward function $R$ satisfies $\widehat{R} = \mathcal{A}(R, \pi_\dagger, \epsilon_\dagger)$ if and only if there exists some $\alpha_{s,a} \geq 0$ such that*

$$R = \widehat{R} + \sum_{(s,a) \in \Theta^{\epsilon_\dagger}} \alpha_{s,a} \cdot \left( \psi^{\pi_\dagger \{s;a\}} - \psi^{\pi_\dagger} \right).$$

.

*Proof.* To prove the statement, we use Lemma E.1. The primal feasibility condition in the lemma always holds as $\widehat{R} \in \mathcal{A}(\overline{R}, \pi_\dagger, \epsilon_\dagger)$. Therefore $\widehat{R} \in \mathcal{A}(R, \pi_\dagger, \epsilon_\dagger)$ if and only if there exists $\lambda$ such that the other three conditions hold. Note that the complementary slackness condition is equivalent to

$$\forall (s, a \neq \pi_\dagger(s)) : \lambda(s,a) = 0 \vee \boldsymbol{\Phi}(s,a)^T \cdot R + \epsilon_\dagger = 0 \iff \forall(s,a) \notin \Theta^{\epsilon_\dagger} : \lambda(s,a) = 0.$$

Therefore from dual feasibility, stationarity and complemantary slackness it follows that $\widehat{R} \in \mathcal{A}(R, \pi_\dagger, \epsilon_\dagger)$ if and only if there exists $\lambda$ such that

$$\lambda \succcurlyeq 0,$$
$$R = \widehat{R} + \sum_{(s,a)} \lambda(s,a) \cdot \left( \psi^{\pi_\dagger\{s;a\}} - \psi^{\pi_\dagger} \right),$$
$$\forall(s,a) \notin \Theta^{\epsilon_\dagger} : \lambda(s,a) = 0.$$

The Lemma therefore follows by setting $\alpha_{s,a} = \lambda(s,a)$ since setting $\lambda(s,a) = 0$ for all $(s,a) \notin \Theta^{\epsilon_\dagger}$ is equivalent to not summing over the terms corresponding to $(s,a) \notin \Theta^{\epsilon_\dagger}$ in the stationarity condition. $\square$

A direct consequence of this lemma is the following result.

**Corollary E.4.** *Assume that $\widehat{R} = \mathcal{A}(R, \pi_\dagger, \epsilon_\dagger)$ and $\widehat{R} \neq R$. It follows that*

$$\widehat{\epsilon} = \epsilon_\dagger,$$

*where*

$$\widehat{\epsilon} = \min_{s, a \neq \pi_\dagger(s)} \left[ \widehat{\rho}^{\pi_\dagger} - \widehat{\rho}^{\pi_\dagger\{s;a\}} \right].$$

*Proof.* Assume to the contrary that $\widehat{\epsilon} \neq \epsilon_\dagger$. Given the primal feasibility condition in Lemma E.1, $\widehat{\epsilon} \geq \epsilon_\dagger$. Therefore $\widehat{\epsilon} > \epsilon_\dagger$. It follows that

$$\forall s, a \neq \pi_\dagger(s) : \widehat{\rho}^{\pi_\dagger} - \widehat{\rho}^{\pi_\dagger\{s;a\}} > \epsilon_\dagger \implies \Theta^{\epsilon_\dagger} = \emptyset.$$

Given Lemma E.3, this implies that $R = \widehat{R}$, which contradicts the initial assumption $R \neq \widehat{R}$. $\square$

# F Proofs of Section 5.1

This section of the appendix contains the proofs of the formal results presented in Section 5.

## F.1 Proof of Theorem 5.1

Before proving the theorem we prove some results that we need for the proof of this theorem, as well as for the results in later sections.

**Lemma F.1.** *Consider policy $\pi$ with state-action occupancy measure $\psi^\pi$. Solution $\rho_{\min}^\pi$ to the following optimization problem:*

$$\min_R \rho^\pi \quad s.t. \quad \widehat{R} = \mathcal{A}(R, \pi_\dagger, \epsilon_\dagger), \tag{P4}$$

*satisfies:*

$$\rho_{\min}^\pi = \begin{cases} \widehat{\rho}^\pi & \text{if} \quad \forall s, a \in \Theta^{\epsilon_\dagger} : \Gamma^{\{s;a\}}(\pi) \geq 0 \\ -\infty & \text{otherwise} \end{cases}.$$

*Proof.* We separately analyze the two cases: the case when $\Gamma^{\{s;a\}}(\pi) \geq 0$ for all $(s, a) \in \Theta^{\epsilon_\dagger}$ holds, and the case when it does not.

**Case 1:** If $\Gamma^{\{s;a\}}(\pi) \geq 0$ for all $(s, a) \in \Theta^{\epsilon_\dagger}$, then by using Equation (4) and Lemma E.3 we obtain that

$$\rho^\pi - \widehat{\rho}^\pi = \left\langle \psi^\pi, R - \widehat{R} \right\rangle = \sum_{(s,a) \in \Theta^{\epsilon_\dagger}} \alpha_{s,a} \cdot \left\langle \psi^\pi, \psi^{\pi_\dagger \{s;a\}} - \psi^{\pi_\dagger} \right\rangle \geq 0.$$

Therefore, $\rho^\pi \geq \widehat{\rho}^\pi$. Furthermore, from Lemma E.3, we know that $R = \widehat{R}$ satisfies the constraint in the optimization problem (P4), so the score of the optimal solution to (P4) is $\rho_{\min}^\pi = \widehat{\rho}^\pi$.

**Case 2:** Now, consider the case when $\Gamma^{\{s;a\}}(\pi) < 0$ for a certain state-action pair $(s, a) \in \Theta^{\epsilon_\dagger}$. Let $\alpha_{s,a}$ be an arbitrary positive number. From Lemma E.3, we know that

$$R = \widehat{R} + \alpha_{s,a} \cdot \left\langle \psi^\pi, \psi^{\pi_\dagger \{s;a\}} - \psi^{\pi_\dagger} \right\rangle$$

satisfies the constraint in the optimization problem (P4), and hence is a solution to (P4). Moreover, by using this solution together with Equation (4), we obtain

$$\rho^\pi - \widehat{\rho}^\pi = \left\langle \psi^\pi, R - \widehat{R} \right\rangle = \alpha_{s,a} \cdot \left\langle \psi^\pi, \psi^{\pi_\dagger \{s;a\}} - \psi^{\pi_\dagger} \right\rangle = \alpha_{s,a} \cdot \Gamma^{\{s;a\}}(\pi). \tag{13}$$

Since $\alpha_{s,a}$ can be arbitrarily large and $\Gamma^{\{s;a\}}(\pi) < 0$, while $\widehat{\rho}^\pi$ is fixed, $\rho^\pi$ can be arbitrarily small. Hence, the score of the optimal solution to (P4) is unbounded from below, i.e., $\rho_{\min}^\pi = -\infty$. $\qquad\square$

We can now prove Theorem 5.1, that is the following statement.

**Statement:** *Consider the following optimization problem parameterized by $\epsilon$:*

$$\max_{\psi \in \Psi} \left\langle \psi, \widehat{R} \right\rangle \tag{P3}$$

$$\text{s.t.} \ \left\langle \psi^{\pi_\dagger \{s;a\}} - \psi^{\pi_\dagger}, \psi \right\rangle \geq 0 \quad \forall s, a \in \Theta^\epsilon.$$

*For $\epsilon = \epsilon_\dagger$, this optimization problem is always feasible, and its optimal solution $\psi_{\max}$ specifies an optimal solution to the optimization problem* (P2a) *with*

$$\pi_\mathcal{D}(a|s) = \frac{\psi_{\max}(s, a)}{\sum_{a'} \psi_{\max}(s, a')}. \tag{14}$$

*The score of $\pi_\mathcal{D}(a|s)$ is lower bounded by $\overline{\rho}^{\pi_\mathcal{D}} \geq \widehat{\rho}^{\pi_\mathcal{D}}$.*

*Proof.* The feasibility of the problem follows from Theorem 4.4 [6]. Note that $\psi_{\max}$ always exists since (P3) is maximizing a continuous function over a closed and bounded set. Concretely, the constraints $\left\langle \psi^{\pi_\dagger \{s;a\}} - \psi^{\pi_\dagger}, \psi \right\rangle \geq 0$ and Equations (9) and (10) each define closed sets, and since $||\psi||_1 = 1$, the set $\Psi$ is bounded.

In order to see why $\psi_{\max}$ specifies an optimal solution to (P2a), note that we can rewrite (P2a) as

$$\max_\pi \rho_{\min}^\pi,$$

---

[6]The proof of Theorem 4.4 does not rely on this result .

where $\rho_{\min}^\pi$ is the solution to the optimization problem (P4). Due to Lemma F.1, this could be rewritten as

$$\max_\pi \widehat{\rho}^\pi$$
$$\text{s.t. } \Gamma^{\{s;a\}}(\pi) \geq 0 \quad \forall(s,a) \in \Theta^{\epsilon\dagger}.$$

Namely, maximizing a function $f(x)$ subject to constraint $x \in \mathcal{X}$ (where $\mathcal{X} \neq \emptyset$) is equivalent to maximizing $\tilde{f}(x)$, where

$$\tilde{f}(x) = \begin{cases} f(x) & \text{if} \quad x \in \mathcal{X} \\ -\infty & \text{o.w.} \end{cases} .$$

Due to (4) and (8), the constrained optimization problem above can be rewritten as

$$\max_\pi \left\langle \psi^\pi, \widehat{R} \right\rangle$$
$$\text{s.t. } \left\langle \psi^{\pi_\dagger\{s;a\}} - \psi^{\pi_\dagger}, \psi^\pi \right\rangle \quad \forall(s,a) \in \Theta^{\epsilon\dagger}.$$

Therefore, given Lemma D.3, $\psi_{\max}$ specifies a solution to (P2a) via (2).

Finally, note that the constraints of the optimization problem (P3) ensure that a policy $\pi$ whose occupancy measure is equal to $\psi_{\max}$ will have $\Gamma^{\{s;a\}}(\pi) \geq 0$ — in other words, $\Gamma^{\{s;a\}}(\pi_\mathcal{D})$ is non-negative for all $(s,a) \in \Theta^{\epsilon\dagger}$. Due to Lemma F.1, we know that such policy $\pi$ will have the worst case utility equal to $\widehat{\rho}^\pi$. Therefore, $\overline{\rho}^\pi \geq \widehat{\rho}^\pi$. $\qquad\square$

*Remark* F.2. Given Lemma D.3, the constraint $\psi \in \Psi$ can equivalently be replaced with constraints (9) and (10), making the optimization problem (P3) a linear program.

### F.2    Proof of Theorem 5.2

The proof of the theorem is similar to the proof of Theorem 5.1 and builds on two lemmas which we introduce in this section.

**Lemma F.3.** *Set $\epsilon = \min\{\epsilon_\mathcal{D}, \widehat{\epsilon}\}$, where*

$$\widehat{\epsilon} = \min_{s,a \neq \pi_\dagger(s)} \left[ \widehat{\rho}^{\pi_\dagger} - \widehat{\rho}^{\pi_\dagger\{s;a\}} \right].$$

*Reward function $R$ satisfies $\widehat{R} = \mathcal{A}(R, \pi_\dagger, \tilde{\epsilon})$ for some $\tilde{\epsilon} \in (0, \epsilon_\mathcal{D}]$ if any only if*

$$R = \widehat{R} + \sum_{(s,a) \in \Theta^\epsilon} \alpha_{s,a} \cdot \left( \psi^{\pi_\dagger\{s;a\}} - \psi^{\pi_\dagger} \right),$$

*for some $\alpha_{s,a} \geq 0$.*

*Proof.* We divide the proof into two parts, respectively proving the sufficiency and the necessity of the condition.

*Part 1 (Necessity):* Assume that $\widehat{R} = \mathcal{A}(R, \pi_\dagger, \tilde{\epsilon})$ for some $\tilde{\epsilon} \in (0, \epsilon_\mathcal{D}]$. From the stationariry and dual feasibility conditions in Lemma E.1, we deduce

$$\exists \lambda \succcurlyeq 0 : R = \widehat{R} + \sum_{s,a \neq \pi_\dagger(s)} \lambda(s,a) \cdot (\psi^{\pi_\dagger\{s;a\}} - \psi^{\pi_\dagger}). \tag{15}$$

We claim that $\lambda(s,a) = 0$ for all $(s,a) \notin \Theta^\epsilon$. Note that this would imply the lemma's statement by setting $\alpha_{s,a} = \lambda(s,a)$, since the terms corresponding to $(s,a) \notin \Theta^\epsilon$ could be skipped in the summation of (15).

To see why the claim holds, assume that $\lambda(s,a) \neq 0$ for some $(s,a)$ where $a \neq \pi_\dagger(s)$. From complementary slackness, we know that $\mathbf{\Phi}(s,a)^T \cdot R + \tilde{\epsilon} = 0$, which implies that

$$\widehat{\epsilon} = \min_{\tilde{s},\tilde{a} \neq \pi_\dagger(\tilde{s})} (-\mathbf{\Phi}(\tilde{s},\tilde{a})^T \cdot R) \leq -\mathbf{\Phi}(s,a)^T \cdot R = \tilde{\epsilon}. \tag{16}$$

However, $\tilde{\epsilon} \leq \widehat{\epsilon}$ holds by primal feasibility. Therefore, all the inequalities are equalities, which implies $\tilde{\epsilon} = \widehat{\epsilon}$. Since $\tilde{\epsilon} \leq \epsilon_{\mathcal{D}}$, we conclude that $\tilde{\epsilon} = \min\{\epsilon_{\mathcal{D}}, \widehat{\epsilon}\} = \epsilon$. Since all of the inequalities in (16) are indeed equalities, we conclude

$$-\mathbf{\Phi}(s,a)^T \cdot R = \epsilon \implies (s,a) \in \Theta^\epsilon,$$

which proves the claim.

*Part 2 (Sufficiency):* Assume that

$$R = \widehat{R} + \sum_{(s,a) \in \Theta^\epsilon} \alpha_{s,a} \cdot \left( \psi^{\pi_\dagger\{s;a\}} - \psi^{\pi_\dagger} \right),$$

for some $\alpha_{s,a} \geq 0$. Set $\tilde{\epsilon} = \epsilon$ and note that $\tilde{\epsilon} \leq \epsilon_{\mathcal{D}}$ by definition. Set

$$\lambda(s,a) = \begin{cases} \alpha_{s,a} & \text{if} \quad (s,a) \in \Theta^\epsilon \\ 0 & \text{o.w.} \end{cases} .$$

We now verify all the conditions of Lemma E.1 hold. Stationarity and dual feasibility hold because $R = \widehat{R} + \sum_{s,a} \lambda(s,a) \cdot \mathbf{\Phi}(s,a)$ and $\lambda \succeq 0$. Primal feasibility holds because $\tilde{\epsilon} = \min\{\epsilon_{\mathcal{D}}, \widehat{\epsilon}\} \leq \widehat{\epsilon}$. Finally, complementary slackness holds because

$$\lambda(s,a) \neq 0 \implies (s,a) \in \Theta^\epsilon \implies \mathbf{\Phi}(s,a)^T R + \epsilon = 0.$$

$\square$

**Lemma F.4.** *Let $\rho_{min}^\pi$ be the solution to the following optimization problem*

$$\min_R \rho^\pi \quad s.t. \quad \widehat{R} = \mathcal{A}(R, \pi_\dagger, \tilde{\epsilon}) \wedge 0 < \tilde{\epsilon} \leq \epsilon_{\mathcal{D}}. \tag{P5}$$

*Then*

$$\rho_{min}^\pi = \begin{cases} \widehat{\rho}^\pi & \text{if} \quad \forall (s,a) \in \Theta^\epsilon : \Gamma^{\{s;a\}}(\pi) \geq 0 \\ -\infty & \text{o.w.} \end{cases} ,$$

*where $\epsilon = \min\{\epsilon_{\mathcal{D}}, \widehat{\epsilon}\}$, and*

$$\widehat{\epsilon} = \min_{s,a \neq \pi_\dagger(s)} \left[ \widehat{\rho}^{\pi_\dagger} - \widehat{\rho}^{\pi_\dagger\{s;a\}} \right].$$

*Proof.* The proof is similar to the proof of Lemma F.1. We separately analyze the two cases: the case when $\Gamma^{\{s;a\}}(\pi) \geq 0$ for all $(s,a) \in \Theta^\epsilon$ holds, and the case when it does not.

**Case 1:** If $\Gamma^{\{s;a\}}(\pi) \geq 0$ for all $(s,a) \in \Theta^\epsilon$, then by using Equation (4) and Lemma F.3 we obtain that

$$\rho^\pi - \widehat{\rho}^\pi = \left\langle \psi^\pi, R - \widehat{R} \right\rangle = \sum_{(s,a) \in \Theta^\epsilon} \alpha_{s,a} \cdot \left\langle \psi^\pi, \psi^{\pi_\dagger\{s;a\}} - \psi^{\pi_\dagger} \right\rangle \geq 0.$$

Therefore, $\rho^\pi \geq \widehat{\rho}^\pi$. Furthermore, from Lemma F.3, we know that $R = \widehat{R}$ satisfies the constraint in the optimization problem (P5), so the score of the optimal solution to (P5) is $\rho_{\min}^\pi = \widehat{\rho}^\pi$.

**Case 2:** Now, consider the case when $\Gamma^{\{s;a\}}(\pi) < 0$ for a certain state-action pair $(s,a) \in \Theta^{\epsilon}$. Let $\alpha_{s,a}$ be an arbitrary positive number. From Lemma F.3, we know that

$$R = \widehat{R} + \alpha_{s,a} \cdot \left\langle \psi^{\pi}, \psi^{\pi_{\dagger}\{s;a\}} - \psi^{\pi_{\dagger}} \right\rangle$$

satisfies the constraint in the optimization problem (P5), and hence is a solution to (P5). Moreover, by using this solution together with Equation (4), we obtain

$$\rho^{\pi} - \widehat{\rho}^{\pi} = \left\langle \psi^{\pi}, R - \widehat{R} \right\rangle = \alpha_{s,a} \cdot \left\langle \psi^{\pi}, \psi^{\pi_{\dagger}\{s;a\}} - \psi^{\pi_{\dagger}} \right\rangle = \alpha_{s,a} \cdot \Gamma^{\{s;a\}}.$$

Since $\alpha_{s,a}$ can be arbitrarily large and $\Gamma^{\{s;a\}} < 0$, while $\widehat{\rho}^{\pi}$ is fixed, $\rho^{\pi}$ can be arbitrarily small. Hence, the score of the optimal solution to (P5) is unbounded from below, i.e., $\rho^{\pi}_{\min} = -\infty$. $\qquad\square$

We are now ready to prove Theorem 5.2.

**Statement:** *Assume that $\epsilon_{\mathcal{D}} \geq \epsilon_{\dagger}$, and define $\widehat{\epsilon} = \min_{s,a \neq \pi_{\dagger}(s)} \left[ \widehat{\rho}^{\pi_{\dagger}} - \widehat{\rho}^{\pi_{\dagger}\{s;a\}} \right]$. Then, the optimization problem (P3) with $\epsilon = \min\{\epsilon_{\mathcal{D}}, \widehat{\epsilon}\}$ is feasible and its optimal solution $\psi_{\max}$ identifies an optimal policy $\pi_{\mathcal{D}}$ for the optimization problem (P2b) via Equation (2). This policy $\pi_{\mathcal{D}}$ satisfies $\overline{\rho}^{\pi_{\mathcal{D}}} \geq \widehat{\rho}^{\pi_{\mathcal{D}}}$.*

*Proof.* The proof is divide into two parts, respectively proving the first and the second claim in the theorem statement.

*Part 1 (Solution to* (P2b)*):* We prove that the optimization problem (P3) is feasible, its optimal solution $\psi_{\max}$ identifies an optimal solution to (P2b) via Equation (2), and satisfies $\overline{\rho}^{\pi_{\mathcal{D}}} \geq \widehat{\rho}^{\pi_{\mathcal{D}}}$.

The feasibility of the problem follows from Theorem 4.4. Note that $\psi_{\max}$ always exists since (P3) is maximizing a continuous function over a closed and bounded set. Concretely, the constraints $\left\langle \psi^{\pi_{\dagger}\{s;a\}} - \psi^{\pi_{\dagger}}, \psi \right\rangle \geq 0$ and Equations (9) and (10) each define closed sets and since $\|\psi\|_1 = 1$, the set $\Psi$ is bounded.

In order to see why $\psi_{\max}$ specifies an optimal solution to (P2b), note that we can rewrite (P2b) as

$$\max_{\pi} \rho^{\pi}_{\min},$$

where $\rho^{\pi}_{\min}$ is the solution to the optimization problem (P5). Due to Lemma F.4, this could be rewritten as

$$\max_{\pi} \widehat{\rho}^{\pi}$$
$$\text{s.t. } \Gamma^{\{s;a\}}(\pi) \geq 0 \quad \forall (s,a) \in \Theta^{\epsilon},$$

where $\epsilon = \min\{\epsilon_{\mathcal{D}}, \widehat{\epsilon}\}$. Namely, maximizing a function $f(x)$ subject to constraint $x \in \mathcal{X}$ (where $\mathcal{X} \neq \emptyset$) is equivalent to maximizing $\tilde{f}(x)$, where

$$\tilde{f}(x) = \begin{cases} f(x) & \text{if } x \in \mathcal{X} \\ -\infty & \text{o.w.} \end{cases} .$$

Due to (4) and (8), the constrained optimization problem above can be rewritten as

$$\max_{\pi} \left\langle \psi^{\pi}, \widehat{R} \right\rangle$$
$$\text{s.t. } \left\langle \psi^{\pi_{\dagger}\{s;a\}} - \psi^{\pi_{\dagger}}, \psi^{\pi} \right\rangle \quad \forall (s,a) \in \Theta^{\epsilon}.$$

Therefore, given Lemma D.3, $\psi_{\max}$ specifies a solution to (P2b) via (2). Finally, given Lemma F.4, $\psi^{\pi_{\mathcal{D}}}$ satisfies the constraints of (P5) and therefore $\widehat{\rho}^{\pi_{\mathcal{D}}}$ is a lower bound on $\overline{\rho}^{\pi_{\mathcal{D}}}$.

$\qquad\square$

### F.3 Proof of Theorem 5.3

**Statement:** *If $\epsilon_\dagger > \epsilon_\mathcal{D}$, then $\pi_\dagger$ is the unique solution of the optimization problem* (P2b), *hence $\pi_\mathcal{D} = \pi_\dagger$.*

*Proof.* As in Theorem 5.2, set $\epsilon = \min\{\widehat{\epsilon}, \epsilon_\mathcal{D}\}$ where

$$\widehat{\epsilon} = \min_{s, a \neq \pi_\dagger(s)} \left[\widehat{\rho}^{\pi_\dagger} - \widehat{\rho}^{\pi_\dagger\{s;a\}}\right].$$

From the feasibility of the attack, we have that

$$\forall s, a \neq \pi_\dagger(s) : \widehat{\rho}^{\pi_\dagger} - \widehat{\rho}^{\pi_\dagger\{s;a\}} \geq \epsilon_\dagger > \epsilon_\mathcal{D} \geq \epsilon \implies \Theta^\epsilon = \emptyset.$$

Therefore, given Lemma F.3, the constraint in the optimization problem (P2b) is satisfied only for $R = \widehat{R}$. This reduces the optimization problem (P2b) to $\max_\pi \widehat{\rho}^\pi$, which has a unique optimal solution: $\pi_\dagger$. $\qquad\square$

## G Proofs of Section 4

In this section, we provide the proofs of the results of Section 4[7] as well as additional results that characterize the defense optimization problem. While our results are stated for the defense optimization problem (P2a), all results hold for (P2b) as well with $\epsilon = \min\{\epsilon_\mathcal{D}, \widehat{\epsilon}\}$.

### G.1 Solution to the defense optimization problem for the $\ell_1$ attack cost

In this section, we present a convex optimization framework for solving the defense optimization problem (P2a) for $\mathcal{C} = \{c_1\}$. Our main result is the following theorem, the proof of which is presented in Section G.3.

**Theorem G.1.** *Let $\mathbf{\Phi} \in \mathbb{R}^{|S| \cdot (|A|-1) \times |S| \cdot |A|}$ be a matrix with rows consisting of the vectors $\psi^{\pi_\dagger\{s;a\}} - \psi^{\pi_\dagger}$ as in Section E and let $\mathbf{\Phi}_\theta$ be the sub-matrix of $\mathbf{\Phi}$ consisting of the rows corresponding to $\Theta^{\epsilon_\dagger}$. Define the set $U \subseteq S \times A$ as the set of state action pairs $(\tilde{s}, \tilde{a})$ for which the following optimization problem is feasible.*

$$\forall s, a : \left|\left(\mathbf{\Phi}_\theta^T \lambda\right)(s, a)\right| \leq 1 \text{ and } \left(\mathbf{\Phi}_\theta^T \lambda\right)(\tilde{s}, \tilde{a}) = -1 \text{ and } \lambda \succcurlyeq 0.$$

*where $\succcurlyeq$ denotes coordinate-wise inequality. Consider the following optimization problem:*

$$\max_\pi \widehat{\rho}^{\pi, R}, \quad \text{s.t.} \quad \pi(a|s) = 0 \quad \forall s, a \in U. \tag{17}$$

*The optimization problem* (17) *is always feasible and its solution is a solution to the defense optimization problem* (P2a) *with $\mathcal{C} = \{c_1\}$.*

### G.2 Proof of Theorem 4.4

**Statement:** *Consider any policy $\pi_{\not\dagger}$ s.t. $\pi_{\not\dagger}(\pi_\dagger(s)|s) = 0$. For $\mathcal{C} = \{c_p \text{ s.t. } p \in [1, \infty)\}$, the optimal value of problem* (P2a) *is bounded from below by $\widehat{\rho}^{\pi_{\not\dagger}}$.*

*Proof.* We claim that $\widehat{R}(s, a) \leq R(s, a)$ for all $R$ satisfying $\widehat{R} \in \mathcal{A}(R)$ and all $s, a \neq \pi_\dagger(s)$. To see why, assume that if this not the case for some $\tilde{s}, \tilde{a}$ and define $\widetilde{R}$ as

$$\widetilde{R}(s, a) = \begin{cases} R(s, a) & \text{if} \quad s, a = \tilde{s}, \tilde{a} \\ \widehat{R}(s, a) & \text{o.w.} \end{cases}$$

It is clear that $\left\|\widetilde{R} - R\right\|_p < \left\|\widehat{R} - R\right\|_p$ for all $p \in [1, \infty)$. We claim that $\widetilde{R}$ is feasible for the attack optimization problem (P1) with parameters $c_p, R, \pi_\dagger, \epsilon_\dagger$. This would contradict the assumption, $\widehat{R} \in \mathcal{A}(R)$ as it would mean $\widehat{R}$ is not an optimal solution to (P1), finishing the proof.

---

[7]The results related to computational hardness are discussed separately in Appendix H.

To prove the claim, since (P1) is equivalent to (P1'), we need to show that

$$\rho^{\widetilde{R},\pi_\dagger\{\tilde{s};\tilde{a}\}} - \rho^{\widetilde{R},\pi_\dagger} \le -\epsilon_\dagger,$$

for all $\tilde{s}, \tilde{a} \ne \pi_\dagger(\tilde{s})$. By definition of $\widetilde{R}$ however, $\rho^{\widetilde{R},\pi_\dagger\{\tilde{s};\tilde{a}\}} = \rho^{\widehat{R},\pi_\dagger\{\tilde{s};\tilde{a}\}}$ as

$$\rho^{\widetilde{R},\pi_\dagger\{\tilde{s};\tilde{a}\}} - \rho^{\widehat{R},\pi_\dagger\{\tilde{s};\tilde{a}\}} = \left\langle \widetilde{R} - \widehat{R}, \psi^{\pi_\dagger\{\tilde{s};\tilde{a}\}} \right\rangle = 0, \tag{18}$$

where the first equality follows from Equation (4), and the second equality follows from the fact that $\widetilde{R}$ only differs from $\widehat{R}$ in $(s,a)$, $\psi^{\pi_\dagger\{\tilde{s};\tilde{a}\}}(s,a) = 0$. Simliarly $\rho^{\widetilde{R},\pi_\dagger} \ge \rho^{\widehat{R},\pi_\dagger}$ as

$$\rho^{\widetilde{R},\pi_\dagger} - \rho^{\widehat{R},\pi_\dagger} = \left\langle \widetilde{R} - \widehat{R}, \psi^{\pi_\dagger} \right\rangle \ge 0,$$

where the second inequality follows from the fact that $\widetilde{R}$ is never less than $\widehat{R}$. Therefore,

$$\rho^{\widetilde{R},\pi_\dagger\{\tilde{s};\tilde{a}\}} - \rho^{\widetilde{R},\pi_\dagger} \le \rho^{\widehat{R},\pi_\dagger\{\tilde{s};\tilde{a}\}} - \rho^{\widehat{R},\pi_\dagger} \le -\epsilon_\dagger,$$

where the second inequality follows from the assumption that $\widehat{R} \in \mathcal{A}(R)$. We haver therefore shown (18), finishing the proof. □

### G.3 Characterization of the inner minimization problem in (P2a)

We begin by providing characterisation results for the inner minimization problem in (P2a) for different known cost functions. These results can be seen as extensions of Lemma F.1. Formally, for fixed $p$, consider the following optimization problem

$$\min_R \rho^\pi \quad \text{s.t.} \quad \widehat{R} = \mathcal{A}(c_p, R, \pi_\dagger, \epsilon_\dagger), \tag{P4}$$

The following lemmas characterize the value of the above optimization problem for different values of $p$. In stating these lemmas, we use $\boldsymbol{\Phi} \in \mathbb{R}^{|S|\cdot(|A|-1)\times|S|\cdot|A|}$ be a matrix with rows consisting of the vectors $\psi^{\pi_\dagger\{s;a\}} - \psi^{\pi_\dagger}$ as in Section E and let $\boldsymbol{\Phi}_\theta$ be the sub-matrix of $\boldsymbol{\Phi}$ consisting of the rows corresponding to $\Theta^{\epsilon_\dagger}$.

**Lemma G.2.** *Let $\pi$ be a fixed policy and assume that $1 < p < \infty$ is a fixed number. Define the function $u_p : \mathbb{R} \to \mathbb{R}$ as $u_p(x) = sgn(x) \cdot |x|^{\frac{1}{p-1}}$ where $sgn(x) = \mathbb{1}\,[x > 0] - \mathbb{1}\,[x < 0]$ and let $u_p : \mathbb{R}^n \to \mathbb{R}^n$ be its coordinate-wise extension to $\mathbb{R}^n$, i.e, $u_p(x)_i = u_p(x_i)$.*

*The solution to (P4) equals $\widehat{\rho}^\pi$ if*

$$\left\langle \psi^\pi, u_p(\boldsymbol{\Phi}_\theta^T \lambda) \right\rangle \ge 0 \quad \forall \lambda \succcurlyeq 0,$$

*and equals $-\infty$ otherwise.*

**Lemma G.3.** *Let $\pi$ be a fixed policy. Define the function $u_\infty : \mathbb{R} \to \mathbb{R}$ as*

$$u_\infty(x) = \begin{cases} -1 & if \quad x \le 0 \\ 1 & o.w. \end{cases},$$

*and let $u_\infty : \mathbb{R}^n \to \mathbb{R}^n$ be its coordinate-wise extension to $\mathbb{R}^n$, i.e, $u_\infty(x)_i = u_\infty(x)_i$. The solution to (P4) equals $\widehat{\rho}^\pi$ if*

$$\left\langle \psi^\pi, u_\infty(\boldsymbol{\Phi}_\theta^T \lambda) \right\rangle \ge 0 \quad \forall \lambda \succcurlyeq 0 \quad s.t. \quad \lambda \ne 0.$$

*and equals $-\infty$ otherwise.*

**Lemma G.4.** *Let $\pi$ be a fixed policy and let $U \subseteq S \times A$ be the set of state action pairs $(\tilde{s}, \tilde{a})$ for which the following optimization problem is feasible.*

$$\forall s, a : \left|\left(\boldsymbol{\Phi}_\theta^T \lambda\right)(s, a)\right| \le 1 \ and \ \left(\boldsymbol{\Phi}_\theta^T \lambda\right)(\tilde{s}, \tilde{a}) = -1 \ and \ \lambda \succcurlyeq 0. \tag{19}$$

*Then the solution to (P4) is $-\infty$ if $\pi(\tilde{a}|\tilde{s}) > 0$ for some $\tilde{s}, \tilde{a} \in U$ and is $\widehat{\rho}^\pi$ otherwise.*

When it is clear from context, we will drop the dependence on $p$ in $u_p$. The proof of Lemmas G.2, G.3 and G.4 are provided below. Note that Lemma G.4 immediately implies Theorem G.1 since the feasibility of (17) already follows from Theorem 4.4.

*Proof of Lemma G.2.* Throughout the proof, we will drop the dependence on $c_p$, $\epsilon_\dagger$ and $\pi_\dagger$ in $\mathcal{A}$. The proof follows a similar structure as the results for the $\ell_2$ norm; namely, Lemmas E.1, E.3 and F.1.

We begin by analyzing the constraint $\mathcal{A}(R) = \widehat{R}$ using the KKT conditions. Since $1 < p < \infty$, we can change the objective to $\frac{1}{p} \left\| R - \overline{R} \right\|_p^p$ for convenience. Of the four KKT conditions, primal feasiblity holds if and only if $\mathbf{\Phi}^T R \preccurlyeq -\epsilon$. For the stationarity condition, forming the lagrangian of (P1), we obtain

$$\mathcal{L} = \frac{1}{p} \left\| R - \overline{R} \right\|_p^p + \lambda^T (\mathbf{\Phi} R - \epsilon_\dagger) = \sum \frac{1}{p} (R(s,a) - \overline{R}(s,a))^p + \lambda^T \mathbf{\Phi} R - \epsilon_\dagger \lambda^T \mathbf{1}$$

Taking the gradient,

$$\nabla_R \mathcal{L} = 0 \iff sgn(R(s,a) - \overline{R}(s,a)) \cdot \left| R(s,a) - \overline{R}(s,a) \right|^{p-1} + \left( \mathbf{\Phi}^T \lambda \right)(s,a) = 0 \quad \forall s,a$$

$$\iff \left( \mathbf{\Phi}^T \lambda \right)(s,a) = sgn(\overline{R}(s,a) - R(s,a)) \cdot \left| \overline{R}(s,a) - R(s,a) \right|^{p-1} \quad \forall s,a$$

$$\iff \left| \left( \mathbf{\Phi}^T \lambda \right)(s,a) \right|^{\frac{1}{p-1}} \cdot sgn \left( \left( \mathbf{\Phi}^T \lambda \right)(s,a) \right) = \overline{R}(s,a) - R(s,a) \quad \forall s,a$$

$$\overset{(i)}{\iff} u(\mathbf{\Phi}^T \lambda) = \overline{R} - R,$$

where for $(i)$ we have used the definition of $u$. the complementary slackness condition states that

$$\lambda(s,a) = 0 \quad \forall (s,a) \notin \Theta^{\epsilon_\dagger}.$$

Finally, dual feasible states that $\lambda \succcurlyeq 0$.

Returning to the condition $\mathcal{A}(R) = \widehat{R}$, primal feasibility always holds as $\widehat{R} = \mathcal{A}(\overline{R})$. We therefore conclude that a vector $R$ satisfies $\mathcal{A}(R) = \widehat{R}$ if and only if the following problem is feasible

$$R = \widehat{R} + u(\mathbf{\Phi}^T \lambda)$$

$$\lambda \succcurlyeq 0$$

$$\lambda(s,a) = 0 \quad \forall (s,a) \notin \Theta^{\epsilon_\dagger}$$

By definition of $\mathbf{\Phi}_\theta$, this means that

$$A(R) = \widehat{R} \iff R = \widehat{R} + u(\mathbf{\Phi}_\theta^T \lambda) \quad \text{for some} \quad \lambda \succcurlyeq 0.$$

Returning back to (P4), the problem can be rewritten as

$$\min_{R,\lambda} \quad \langle \psi^\pi, R \rangle$$

$$\text{s.t.} \quad \lambda \succcurlyeq 0$$

$$R = u(\mathbf{\Phi}_\theta^T \lambda) + \widehat{R}.$$

or equivalently,

$$\min_{\lambda \succcurlyeq 0} \quad \left\langle \psi^\pi, u(\mathbf{\Phi}_\theta^T \lambda) \right\rangle + \widehat{\rho}^\pi. \tag{20}$$

Now, assume that $\left\langle \psi^\pi, u(\mathbf{\Phi}_\theta^T \lambda) \right\rangle < 0$ for some $\lambda \succcurlyeq 0$. Observe that $u(c \cdot x) = c^{\frac{1}{1-p}} \cdot x$ for positive constants $c > 0$. Therefore,

$$\left\langle \psi^\pi, u \left( \mathbf{\Phi}_\theta^T (c \cdot \lambda) \right) \right\rangle = c^{\frac{1}{p-1}} \cdot \left\langle \psi^\pi, u \left( \mathbf{\Phi}_\theta^T \lambda \right) \right\rangle$$

Since $1 < p < \infty$, letting $c \to \infty$, the value of $\left\langle \psi^\pi, u \left( \mathbf{\Phi}_\theta^T (c \cdot \lambda) \right) \right\rangle$ can be made arbitrarily low. Therefore, the value of (20) equals $-\infty$. Conversely, assume that $\left\langle \psi^\pi, u(\mathbf{\Phi}_\theta^T \lambda) \right\rangle \geq 0$ for all $\lambda \succcurlyeq 0$. It follows that the value of (20) is bigger than equal to $\widehat{\rho}^\pi$. Since $\lambda = 0$ is feasible for (20), the value is exactly $\widehat{\rho}^\pi$. $\qquad \square$

Before we prove Lemma G.3, we will state and prove the following Lemma which characterizes the subgradient of the $\ell_\infty$ norm.

**Lemma G.5.** *The vectors $w, z_i \in \mathbb{R}^n$ satisfy $w \in \partial \|z\|_\infty$ if and only if **(a)** $z = 0$ and $\|w\|_1 \leq 1$ **or (b)** $\|w\|_1 = 1$ and $z \in \widetilde{u}(w)$ where*

$$\widetilde{u}(w) := \left\{ x \in \mathbb{R}^n : x_i \in \begin{cases} \{c\} & \text{if } w_i > 0 \\ \{-c\} & \text{if } w_i < 0 \\ [-c,c] & \text{if } w_i = 0 \end{cases} \quad \text{for some } c \geq 0 \right\}$$

*Proof.* By Proposition A.22 in Bertsekas (1971),

$$\partial \|z\|_\infty = \text{conv}\{w \text{ s.t } \|w\|_1 \leq 1, z^T w = \|z\|_\infty\} = \{w \text{ s.t } \|w\|_1 \leq 1, z^T w = \|z\|_\infty\}. \tag{21}$$

analyzing the above result, note that

$$z^T w = \sum z_i w_i$$
$$\overset{(a)}{\leq} \sum |w_i| \|z\|_\infty$$
$$= \|w\|_1 \cdot \|z\|_\infty$$
$$\overset{(b)}{\leq} \|z\|_\infty .$$

Since $z^T w = \|z\|_\infty$, equality holds in $(b)$, implying $\|w\|_1 = 1$ or $z = 0$ and in $(a)$, implying $z_i w_i = |w_i| \|z\|_\infty$ for all $i$. This means that if $|z_i| < \|z\|_\infty$, then $w_i = 0$, and if $|z_i| = \|z\|_\infty$, then $w_i \geq 0$ if $z_i \geq 0$ and $w_i \leq 0$ if $z_i \leq 0$.

In other words (this time conditioning on $w$), if $\|w\|_1 < 1$, then $z = 0$. Otherwise, if $w_i > 0$ then $z_i = \|z\|_\infty$, if $w_i < 0$, then $z_i = -\|z\|_\infty$ and if $w_i = 0$ then $z_i$ can be any value in $[-\|z\|_\infty, \|z\|_\infty]$. Therefore, the proof follows by setting $c = \|z\|_\infty$. (the last condition is obviously true but it is important to make explicit as will become clear later). This brings us to the following result. $\square$

We can now prove Lemma G.3.

*Proof of Lemma G.3.* The main ideas of the KKT analysis in the proof of Theorem G.2 still hold and the only condition that changes is the stationarity slackness condition. Formally, the lagrangian equals

$$\left\| R - \overline{R} \right\|_\infty + \lambda^T (\boldsymbol{\Phi} R - \epsilon_\dagger).$$

Therefore, the stationarity condition can be written as

$$0 \in \boldsymbol{\Phi}^T \lambda + \partial \left\| R - \overline{R} \right\|_\infty \iff -\boldsymbol{\Phi}^T \lambda \in \partial \left\| R - \overline{R} \right\|_\infty \tag{22}$$

By Lemma G.5, and given the fact that $\widetilde{u}(-x) = -\widetilde{u}(x)$, the above condition holds if and only if

$$\left( \overline{R} - R \in \widetilde{u}(\boldsymbol{\Phi}^T \lambda) \wedge \left\| \boldsymbol{\Phi}^T \lambda \right\|_1 = 1 \right) \vee \left( \left\| \boldsymbol{\Phi}^T \lambda \right\|_1 \leq 1 \wedge \overline{R} - R = 0 \right)$$

Combining with the complementary slackness and dual feasibility, (P4) can be rewritten as

$$\min_{v,\lambda} \quad \widehat{\rho}^\pi + \langle \psi^\pi, v \rangle$$
$$\text{s.t} \quad \lambda \succcurlyeq 0$$
$$\left( v \in \widetilde{u}(\boldsymbol{\Phi}_\theta^T \lambda) \wedge \left\| \boldsymbol{\Phi}_\theta^T \lambda \right\|_1 = 1 \right) \vee \left( \left\| \boldsymbol{\Phi}_\theta^T \lambda \right\|_1 \leq 1 \wedge v = 0 \right)$$

Now, observe that $\widetilde{u}(c \cdot x) = \widetilde{u}(x)$ for all $c > 0$. Therefore, the $\left\| \boldsymbol{\Phi}_\theta^T \lambda \right\|_1 = 1$ inside the first clause of the last constraint is equivalent to $\boldsymbol{\Phi}_\theta^T \lambda \neq 0$ since if $\boldsymbol{\Phi}_\theta^T \lambda \neq 0$ and $\left\| \boldsymbol{\Phi}_\theta^T \lambda \right\| \neq 1$, then $\left\| \boldsymbol{\Phi}_\theta^T \lambda \right\| = 1$ can be satisfied by

replacing $\lambda$ with $\frac{1}{\left\|\mathbf{\Phi}_\theta^T \lambda\right\|_1} \cdot \lambda$. Similarly, the $\left\|\mathbf{\Phi}_\theta^T \lambda\right\|_1 \leq 1$ inside the second clause is redundant. Therefore, the last constraint can be rewritten as

$$\left(v \in \widetilde{u}(\mathbf{\Phi}_\theta^T \lambda) \wedge \mathbf{\Phi}_\theta^T \lambda \neq 0\right) \vee v = 0$$

Now, observe that $\mathbf{\Phi}_\theta^T \lambda \neq 0$ is equivalent to $\lambda \neq 0$ as the rows of $\mathbf{\Phi}_\theta$ are independent; this is becasue the only row with nonzero $(s, a \neq \pi_\dagger(s))$ is the one corresponding to $(s, a \neq \pi_\dagger(s))$. Therefore, the optimization problem can be rewritten as

$$\min_{v, \lambda} \quad \widehat{\rho}^\pi + \langle \psi^\pi, v \rangle$$
$$\text{s.t} \quad \lambda \succcurlyeq 0$$
$$\left(v \in \widetilde{u}(\mathbf{\Phi}_\theta^T \lambda) \wedge \lambda \neq 0\right) \vee v = 0$$

Assume that there exists $\lambda \neq 0$ satisfying $\lambda \succcurlyeq 0$ such that $\langle u(\mathbf{\Phi}_\theta^T \lambda), \psi^\pi \rangle < 0$. Then for any $c > 0$, $c \cdot u(\mathbf{\Phi}_\theta^T \lambda) \in \widetilde{u}(\mathbf{\Phi}_\theta^T \lambda)$ and therefore, the value of the optimization problem is $-\infty$. Otherwise, the value is lower bounded by $\widehat{\rho}^\pi$ and the bound is attained with $v = 0$, concluding the proof. $\quad\square$

*Proof of Lemma G.4.* We begin by analyzing (P1) as before. Forming the Lagrangian,

$$\mathcal{L} = \left\|R - \overline{R}\right\|_1 + \lambda^T (\mathbf{\Phi}R - \epsilon_\dagger)$$

Therefore, the stationarity condition states that

$$\mathbf{\Phi}^T \lambda \in -\partial \left\|R - \overline{R}\right\|_1$$

As before, the primal feasibility condition holds, dual feasibility states that $\lambda \succcurlyeq 0$ and complementary slackness states that $\lambda(s, a) = 0$ for all $(s, a) \notin \Theta^{\epsilon_\dagger}$. Therefore, $\mathcal{A}(R) = \widehat{R}$ if and only if

$$\mathbf{\Phi}_\theta^T \lambda \in -\partial \left\|\widehat{R} - R\right\|_1 \tag{23}$$

Note however that vectors $z, w$ satisfy $w \in \partial \left\|z\right\|_1$ if and only if

$$w_i \begin{cases} = 1 & \text{if } z_i > 0, \\ = -1 & \text{if } z_i < 0, \\ \in [-1, 1] & \text{if } z_i = 0 \end{cases} \quad\Longleftrightarrow\quad z_i \begin{cases} \geq 0 & \text{if } w_i = 1 \\ \leq 0 & \text{if } w_i = -1 \\ = 0 & \text{if } w_i \in (-1, 1) \end{cases}$$

Now, assume that $\pi(\tilde{a}|\tilde{s}) > 0$ for some $(\tilde{s}, \tilde{a})$ for which (19) is feasible and $\lambda$ be the vector satisfying (19). By (23), we conclude that the vector $R(s, a) = \widehat{R}(s, a) - t \cdot \mathbb{1}\left[(s, a) = (\tilde{s}, \tilde{a})\right]$ satisfies $\mathcal{A}(R) = \widehat{R}$ for any $t > 0$. Therefore, if $\pi(\tilde{a}|\tilde{s}) > 0$, the solution to (P4) is $-\infty$. Conversely, if (19) is not feasible for any such $\tilde{s}, \tilde{a}$, then it follows that $R(s, a) \geq \widehat{R}(s, a)$ for all $R, s, a$ satisfying $\mathcal{A}(R) = \widehat{R}$ and $\pi(a|s) > 0$. Therefore, the value of the optimization problem is lower bounded by $\widehat{\rho}^\pi$. Since $\mathcal{A}(\widehat{R}) = \widehat{R}$, the lower bound is attainble which proves the lemma. $\quad\square$

### G.4 Proof of Proposition 4.2

**Statement:** *Let $c_{const}(R, R')$ be a constant cost function, and assume that $c_{const} \in \mathcal{C}$. Then problem (P2a) is unbounded from below.*

*Proof.* We will show that for any $\pi$, the value of the inner optimization problem of (P2a) equals $-\infty$, thereby proving the result.
Let $\pi$ be an arbitrary policy and consider an arbitrary vector $R$. By definition of $c_{const}$ and the fact that $\widehat{R}$ is feasible for (P1), $\widehat{R} \in \mathcal{A}(c_{const}, R, \pi_\dagger, \epsilon_\dagger)$. Therefore, $R, c_{const}$ are feasible for (P2a). Since $R$ was arbitrary, it can be chosen such that the $\rho^\pi$ is arbitrarily low, proving the claim. $\quad\square$

### G.5 Proof of Theorem 4.3

**Statement:** *There exists MDP $M = (S, A, \widehat{R}, P, \gamma, \sigma)$ for which problem* (P2a) *is unbounded from below when $c_\infty \in \mathcal{C}$.*

*Proof.* Let $\pi$ be an arbitrary policy. We will build an example where the value of the inner minimization problem of (P2a) is $-\infty$.

Consider a single-state MDP with the reward function $\widehat{R} = (1, 0, 0, 0)$ where $\pi_\dagger$ is the first action and $\epsilon_\dagger = 1$. Since $\widehat{R}$ is symmetric with respect to $a_2, a_3, a_4$, we assume without loss of generality that $\pi(a_2) \leq \pi(a_3) \leq \pi(a_4)$. For any $x$, consider the reward function $R' = (1 - x, x, -x, -x)$ with actions $a_1, a_2, a_3, a_4$. We claim that $\widehat{R} \in \mathcal{A}(c_\infty, R', \pi_\dagger, \epsilon_\dagger)$. To see why, consider the attack optimization problem (P1). The cost of the attack optimization problem is at least $x$ because $R'(\pi_\dagger) - R'(a_2) = 1 - 2x$ and any feasible $\widetilde{R}$ needs to satisfy $\widetilde{R}(\pi_\dagger) - \widetilde{R}(a_2) \geq 1$, which implies

$$
\begin{aligned}
\left\| R' - \widetilde{R} \right\|_\infty &\geq \max\{\widetilde{R}(\pi_\dagger) - R'(\pi_\dagger), R'(a_2) - \widetilde{R}(a_2)\} \\
&\geq \frac{1}{2}\left(\widetilde{R}(\pi_\dagger) - R'(\pi_\dagger) + R'(a_2) - \widetilde{R}(a_2)\right) \\
&= \frac{1}{2}(2x - 1 + 1) = x
\end{aligned}
$$

Since $\left\| R' - \widehat{R} \right\|_\infty = x$, we conclude $\widehat{R} \in \mathcal{A}(c_\infty, R', \pi_\dagger, \epsilon_\dagger)$ as claimed.

We now consider the score of the policy $\pi$ under the reward function $R'$ and show that, with the proper choice of $x$, it is unbounded from below. Since $\widehat{R} \in \mathcal{A}(c_\infty, R', \pi_\dagger, \epsilon_\dagger)$, this would show that the value of the inner minimization in (P2a) equals $-\infty$, proving the theorem.

By definition, $\rho^{R', \pi}$ can be written as

$$
\rho^{R', \pi} = \pi(a_1) + x(\pi(a_2) - \pi(a_1) - \pi(a_3) - \pi(a_4)) = \pi(a_1) + x(\pi(a_2) - (1 - \pi(a_2))) = \pi(a_1) + x(2\pi(a_2) - 1).
$$

Since $\pi(a_2) \leq \pi(a_3) \leq \pi(a_4)$ however, we conclude that $\pi(a_2) < 0.5$. Therefore, tending $x$ to infinity finishes the proof. $\qquad\square$

## H Computational hardness results

In this section, we provide computational hardness results for different choices of $\mathcal{C}$, showing that the defense optimization problem (P2a) is NP-hard in different cases. While our results are stated for the defense optimization problem (P2a), all results hold for (P2b) as well with $\epsilon = \min\{\epsilon_\mathcal{D}, \widehat{\epsilon}\}$. Our proofs rely on the results of Appendix G and we therefore refer to this and rely on notation introduced in this Appendix; namely, the notation $\boldsymbol{\Phi}_\theta$ and $u_p$ as introduced in Lemma G.2.

### H.1 Hardness result for $p = \infty$

4.5 We begin by considering the case of $c_\infty$. By reducing the 3SAT problem to the defense optimization problem (P2a), we will show that (P2a) is NP-hard. More formally, we prove the following theorem.

**Theorem H.1.** *For $\mathcal{C} = \{c_\infty\}$, it is NP-hard to determine whether the optimal value of problem* (P2a) *is greater than or equal to $\widehat{\rho}^{\pi_\dagger}$.*

We assume without loss of generality that the clauses do not contain duplicate variables as the 3SAT instance remains NP-hard in this case.

Before proving the theorem, we prove a weaker result which essentially proves hardness assuming we can set the matrix $\boldsymbol{\Phi}_\theta$ and vector $\mu^{\pi_\dagger}$ can be set arbitrarily, without the restriction that they correspond to an MDP.

**Proposition H.2.** *Assume we are given an instance of 3SAT with clauses $c_1, \ldots, c_m$ and variables $(x_1, \ldots x_m)$. Define $k = 2n + 1$ and $\ell = 4n + m + 1$. It is possible to build, in polynomial time, a matrix $\widetilde{M} \in \mathbb{R}^{k \times \ell}$, a vector $\widetilde{\mu} \in \mathbb{R}^\ell$ such that $\widetilde{\mu}$ is strictly positive and*

- *If the 3SAT instance is satisfiable, then there exists a $\lambda \neq 0 \in \mathbb{R}^k$ satisfying $\lambda \succcurlyeq 0$ such that*

$$\left\langle u_\infty(\widetilde{M}^T \lambda), \widetilde{\mu} \right\rangle < -\frac{1}{4}.$$

  *In addition, $\widetilde{M}^T \lambda$ does not contain any zero entries.*
- *If the 3SAT instance is not satisfiable, then for any $\lambda \neq 0 \in \mathbb{R}^k$ satisfying $\lambda \succcurlyeq 0$,*

$$\left\langle u_\infty(\widetilde{M}^T \lambda), \widetilde{\mu} \right\rangle > \frac{1}{4}$$

*Proof.* We begin by an empty matrix and vector and in each step, add a value to $\widetilde{\mu}$ and add a column of size $k$ to $\widetilde{M}$.

We will let $(a_0, a_1, \ldots, a_n, b_1, \ldots, b_n)$ be placeholder names for the coordinates of $\lambda$ Since the optimization problem we are considering involves the matrix product $\widetilde{M}^T \lambda$, for ease of notation, we will specify each column of $\widetilde{M}$ by the linear map forming the corresponding coordinate of $\widetilde{M}^T \lambda$. As an example, $a_0 + 4a_3 + 5b_3$ is a column that has value 1 in $a_0$, 4 in $a_3$, 5 in $b_3$ and 0 everywhere else.

Defining $c := 500k$, we first add the columns below.

$$c \left( a_0 + \sum_{i=1}^{n} a_i + \sum_{i=1}^{n} b_i \right) \tag{24}$$

$$0.9a_0 - (a_i + b_i) \quad \forall i \in [n] \tag{25}$$

$$-1.1a_0 + (a_i + b_i) \quad \forall i \in [n] \tag{26}$$

$$4(0.9a_0 - (a_i - b_i)) \quad \forall i \in [n] \tag{27}$$

$$4(0.9a_0 - (b_i - a_i)) \quad \forall i \in [n] \tag{28}$$

The value of $\widetilde{\mu}$ for (24), (25), (26), (27) and (28) is $(6n^2 + m - \frac{1}{2})$, $3n$, $3n$, 1 and 1 respectively. Note that (24) is the all-ones vector. The constant $c$ in (24) and the constant 4 in (27) and (28) will not matter for the proof, but will be useful for later.

Next for each clause with variables $(x_i, x_j, x_k)$, we will add the following column where $d_i$ denotes $a_i$ if the variable $x_i$ appears as positive in the clause and $b_i$ if the variable appears as negative.

$$0.95 \cdot a_0 - (d_i + d_j + d_k) \tag{29}$$

The value of $\widetilde{\mu}$ in the above is 1.

We now prove that $\widetilde{\mu}, \widetilde{M}$ have the mentioned properties.

If the 3SAT instance is satisfiable, this is easy to show. Set $a_0 = 1$ and set $(a_i, b_i)$ to $(1, 0)$ if $x_i$ is set to true in the satisfiable arrangement and to $(0, 1)$ if $x_i$ is set to 0. It is clear that the coordinates of $u(\widetilde{M}^T \lambda)$ corresponding to (24), (25) and (26) equal 1, $-1$ and $-1$ respectively. Furthermore, since $\{a_i - b_i, b_i - a_i\} = \{1, -1\}$, exactly half of the coordinates of $u(\widetilde{M}^T \lambda)$ corresponding to (27) and (28) equal 1, while the other half equal $-1$. Furthermore, all of the coordinates corresponding to (29) equal -1. Therefore, the inner product $\left\langle \widetilde{\mu}, u(\widetilde{M}^T \lambda) \right\rangle$ equals

$$(6n^2 + m - \frac{1}{2}) - n \cdot 3n - n \cdot 3n + 0 - m = -\frac{1}{2}$$

which proves the claim.

Conversely, assume that the 3SAT instance is not satisfiable. Let $s_1, \ldots s_\ell$ denote the coordinates of $\widetilde{\mu}$ with $s_1$ corresponding to (24) and let $u[s]$ denote $\left(u(\widetilde{M}^T \lambda)\right)(s)$. Let $\lambda \neq 0$ satisfying $\lambda \succcurlyeq 0$ be an arbitrary vector. First note that since $\lambda \neq 0$, $\left(\widetilde{M}^T \lambda\right)(s_1)$ is strictly positive and therefore $u[s_1] = 1$. Since $\widetilde{\mu}(s_1) = 6n^2 + m - \frac{1}{2}$, we need to show that

$$\left\langle u(\widetilde{M}^T \lambda), \widetilde{\mu} \right\rangle > \frac{1}{4} \iff \sum_{s \neq s_1} \widetilde{\mu}(s) \cdot u[s] > -6n^2 - m + \frac{1}{2} + \frac{1}{4} \tag{30}$$

$$\iff \sum_{s \neq s_1} \frac{\widetilde{\mu}(s) \cdot (u[s] + 1)}{2} > \frac{1}{2} \cdot \left(-6n^2 - m + \frac{3}{4} + 2n \cdot 3n + 2n \cdot 1 + m\right) \tag{31}$$

$$\iff \sum_{s \neq s_1} \widetilde{\mu}(s) \cdot \mathbb{1}\left[u[s] = 1\right] > n + \frac{3}{8} \tag{32}$$

Now note that since the value of $\widetilde{\mu}$ in coordinates corresponding to (25) and (26) is $> n + \frac{3}{8}$, if $u[s]$ equals one in any of these coordinates, then the claim is proved.

Therefore, we assume w.l.o.g that for any $i$,

$$0.9a_0 \leq a_i + b_i \leq 1.1a_0.$$

This implies that if any of $\{a_i, b_i\}_{i \geq 1}$ is strictly positive, so is $a_0$. Since at least one of $\{a_0, a_i, b_i\}$ needs to be strictly positive by the assumption $\lambda \neq 0$, we conclude that $a_0$ is strictly positive.

Now note that $u[s]$ is 1 in more than half of the coordinates corresponding to (27) and (28), then (32) will hold. We can therefore assume w.l.o.g that at least half of these coordinates equal $-1$. We note however that if $u[s]$ equals $-1$ for *both* (27) and (28) for some $1 \leq i \leq n$, then

$$0.9a_0 \leq a_i - b_i \wedge 0.9a_0 \leq b_i - a_i \implies a_0 \leq 0 \implies a_0 = 0,$$

which is not possible. We can therefore conclude that for any fixed $i$, $u(\widetilde{M}^T \lambda)$ equals 1 in exactly one of the two coordinates corresponding to (27) and (28). Therefore, either $a_i - b_i \geq 0.9a_0$ or $b_i - a_i \geq 0.9a_0$. Since $a_i + b_i \geq 0.9a_0$, we get that either $a_i \geq 0.9a_0$ or $b_i \geq 0.9a_0$. Since $a_i + b_i \leq 1.1a_0$, the bigger one is $\geq 0.9a_0$ and the smaller one is $\leq 0.2a_0$.

Finally, note that if $u[s]$ equals 1 for any of the coordinates corresponding to (29), then (32) would hold since $u[s]$ was already 1 for half of the coordinates corresponding to (27) and (28). We can therefore assume that $u[s]$ is $-1$ for all the coordinates corresponding to (29). Note however that if $d_i + d_j + d_k \geq 0.95a_0$, then at least one of the $d_i$ must have been $> 0.2$. This means that all of the clauses must hold true (in the 3SAT sense) with $x_i$ set to

$$x_i = \begin{cases} 1, & \text{if } a_i \geq 0.5 \\ -1, & \text{otherwise} \end{cases}.$$

This is not possible however as we assumed that the 3SAT instance was not satisfiable. $\qquad \square$

**Claim H.3.** *Define $\widetilde{M}$ as above and let $s_1$ denote the column of the matrix corresponding to (24). For all rows $\widetilde{v}$ in $\widetilde{M}$, we have*

$$\widetilde{v}(s_1) \geq 50 \sum_{i > 1} |\widetilde{v}(s_i)|,$$

*which further implies $\sum_{i \geq 1} |\widetilde{v}(s_i)| \leq (1 + 1/50)\widetilde{v}(s_1)$.*

*Proof.* The claim follows trivially from the fact that as $\widetilde{v}(s_1) = c$ and all other entries in $\widetilde{v}$ are less than 4. $\qquad \square$

Next, we prove the following lemma which essentially states that for any desired value of $\Theta^{\epsilon\dagger}$, we can find a plausible reward function $\widehat{R}$ such that $\Theta^{\epsilon\dagger}$ equals this value.

**Lemma H.4.** *Let $M$ be an ergodic MDP with unspecified reward function and let $\pi_\dagger$ be a policy in this MDP. For any value of $\epsilon_\dagger > 0$ and any set of state-action pairs $\Theta \subseteq S \times A$ satisfying $\Theta \cap \{(s, \pi(s)) : s \in S\} = \emptyset$, there is a reward function $\widehat{R}$ such that*

    *1. $\widehat{R}$ is feasible for the attack problem* (P1), *i.e, $\mathbf{\Phi}^T \widehat{R} \preccurlyeq -\epsilon_\dagger$.*

    *2. $\Theta^{\epsilon_\dagger} = \Theta$.*

*Proof.* Consider the following reward function,

$$\widehat{R}(s, a) = \begin{cases} 0 & \text{if} \quad a = \pi_\dagger(s) \\ -\frac{\epsilon_\dagger}{\mu^{\pi_\dagger\{s;a\}}(s)} & \text{if} \quad a \in \Theta \\ -\frac{2 \cdot \epsilon_\dagger}{\mu^{\pi_\dagger\{s;a\}}(s)} & \text{o.w.} \end{cases}$$

It is clear that $\rho^{\pi_\dagger} = 0$ and

$$\rho^{\pi_\dagger\{s;a\}} = \begin{cases} -\epsilon_\dagger & \text{if} \quad (s, a) \in \Theta \\ -2\epsilon_\dagger & \text{o.w.} \end{cases}, \quad \text{for all } (s, a \neq \pi_\dagger(s)).$$

which proves the claim. $\qquad\qquad\square$

We now use the above results to construct an MDP, formally proving Theorem H.1. Given an instance of 3SAT, let $\widetilde{M}, \widetilde{\mu}$ denote the values specified in Proposition H.2. Recall that $\widetilde{M} \in \mathbb{R}^{k \times \ell}$ and $\widetilde{\mu} \in \mathbb{R}^\ell$ where $k = 2n + 1$ and $\ell = 4n + m + 1$. Let $\delta > 1/2, \gamma \in [1/2, 1]$ be parameters to be specified later. (see Claims H.11, H.7, and H.5.) Intuitively, we need $\delta$ to be close to zero and $\gamma$ to be close to one.

Given these values, we will build an MDP with reward vector $\widehat{R}$ with $\ell + 3$ states and $k + 1$ actions for which $|\Theta^\epsilon| = k$ and the 3SAT instance is satisfiable if and only if

$$\exists \lambda \neq 0 \in \mathbb{R}^k : \lambda \geq 0 \wedge \left\langle u_\infty(\mathbf{\Phi}_\theta^T \lambda), \psi^{\pi_\dagger} \right\rangle < 0. \tag{33}$$

In our construction, the states $s_{i \geq 1}$ will each correspond to the columns of $\widetilde{M}$ while the states $s_{-2}, s_{-1}, s_0$ will be new. Furthermore, in state $s_0$, each of the actions $a \neq \pi_\dagger(s_0)$ will correspond to a row of $\widetilde{M}$.

Let $M$ denote the submatrix of $\mathbf{\Phi}_\theta$ with only columns corresponding to $(s, \pi_\dagger(s))$. We first observe that (33) can be simplified because $\psi^{\pi_\dagger}(s, a)$ is 0 for all $s, a \neq \pi_\dagger(s)$. Therefore, (33) is equivalent to

$$\exists \lambda \neq 0 \in \mathbb{R}^k : \lambda \geq 0 \wedge \left\langle u_\infty(M^T \lambda), \mu^{\pi_\dagger} \right\rangle < 0. \tag{34}$$

In order to specify this MDP, we first specify the transition probabilities of $\pi_\dagger$. In state $s_0$, following $\pi_\dagger$ leads to $s_i$ for $i \in \{-1, -2\}$ with probability $\delta/2$ and leads $s_i$ for $i \geq 1$ with the probability $(1 - \delta)\widehat{\mu}(s_i)$ where

$$\widehat{\mu} = \frac{\widetilde{\mu}}{\|\widetilde{\mu}\|_1}. \tag{35}$$

In states $s \neq s_0$, following $\pi_\dagger$ will lead back to $s$ with probability 1. The initial distribution $\sigma$ of the MDP is chosen as $\sigma(s_0) = 1$ and $\sigma(s_i) = 0$ for $i \neq 0$.

It is straightforward to see that

$$\mu^{\pi_\dagger}(s_0) = (1 - \gamma), \quad \forall i \in \{-2, -1\} : \mu^{\pi_\dagger}(s_i) = \frac{\gamma \cdot \delta}{2}, \quad \forall i \geq 1 : \mu^{\pi_\dagger}(s_i) = \gamma \cdot (1 - \delta) \cdot \frac{\widetilde{\mu}(s_i)}{\|\widetilde{\mu}\|_1}.$$

Now, for each of the rows in $\widetilde{M}$ like $\widetilde{v}$, we add an action $a$ to state $s_0$ with the following transition probabilities.

$$P(s_0, a, s_j) = \begin{cases} P(s_0, \pi_\dagger(s_0), s_j) & \text{if} \quad j = 0 \\ P(s_0, \pi_\dagger(s_0), s_j) - \frac{\delta}{4\theta} \cdot \left(\sum_i \widetilde{v}(s_i)\right) & \text{if} \quad j \in \{-2, -1\} \\ P(s_0, \pi_\dagger(s_0), s_j) + \frac{\delta}{2\theta} \cdot \widetilde{v}(s_j) & \text{if} \quad j \geq 1 \end{cases}. \tag{36}$$

where $\theta$ is taken to be large enough such that for all $\widetilde{v}$,

$$\theta \cdot \min_i \widehat{\mu}(s_i) \geq \|\widetilde{v}\|_1.$$

Note that the value of $\theta$ is the same for all the rows of matrix; it is set to the maximum of $\frac{1}{\min_i \widehat{\mu}_i} \cdot \|\widetilde{v}\|_1$ across all rows $\widetilde{v}$ of $\widetilde{M}$.

**Claim H.5.** *If $\delta \leq \frac{1}{10}$, the transition probabilities specified in (35) are valid, i.e., they are in the range $[0, 1]$ and $\sum_{s'} P(s_0, a, s') = 1$.*

*Proof.* In order to make sure these transition probabilities are valid, they need to sum to one and they all need to be non-negative. They sum to one by definition. As for being non-negative, it holds trivially for $j = 0$ and holds for $j \neq 0$ by definition of $\delta$. Formally, for $j \leq -1$,

$$P(s_0, \pi_\dagger(s_0), s_0) - \frac{\delta}{4} \cdot \left(\sum_i v(s_i)\right) = \delta \cdot \left(\frac{1}{2} - \frac{1}{4} \cdot \frac{\sum_i \widetilde{v}(s_i)}{\theta}\right)$$

$$\geq \delta \left(\frac{1}{2} - \frac{1}{4} \cdot (\min_i \widehat{\mu}_i) \cdot \frac{\sum \widetilde{v}(s_i)}{\|\widetilde{v}\|_1}\right)$$

$$\geq \delta(\frac{1}{2} - \frac{1}{4})$$

$$> 0$$

and for $j \geq 1$,

$$P(s_0, \pi_\dagger(s_0), s_j) + \frac{\delta}{2} \cdot \frac{\widetilde{v}(s_j)}{\theta} \geq P(s_0, \pi_\dagger(s_0), s_j) - \frac{\delta}{2} \cdot \frac{|\widetilde{v}(s_j)|}{\theta}$$

$$\geq (1 - \delta) \min_i \widehat{\mu}_i - \frac{\delta}{2} \cdot \frac{\widetilde{v}(s_j)}{\|\widetilde{v}\|_1} \cdot \left(\min_i \widehat{\mu}_i\right)$$

$$\geq (\min_i \widehat{\mu}_i) \cdot (1 - \frac{3\delta}{2}) > 0$$

$\square$

Now, using Lemma H.4, set the reward for the MDP such that $\Theta^{\epsilon\dagger}$ consists of the added actions in $s_0$. Note that there are multiple actions in the states $s_{i \neq 0}$ as well; however, given Lemma H.4, their transition probabilities are not important and can be set arbitrarily as long as the MDP remains ergodic.

**Claim H.6.** *For all $a, \neq \pi_\dagger(s_0)$,*

$$(M^T \lambda)(a) = \left(-\frac{\delta}{4\theta} \cdot \gamma \cdot (\sum \widetilde{v}(s_j)), \quad -\frac{\delta}{4\theta} \cdot \gamma \cdot (\sum \widetilde{v}(s_j)), \quad -(1 - \gamma), \quad \gamma \cdot \frac{\delta}{2\theta} \cdot \widetilde{v}\right),$$

*where $\widetilde{v}$ is the row corresponding to $a$ and we have abused notation by using $\gamma \cdot \frac{\delta}{2} \cdot \widetilde{v}$ to denote the last $k$ entries of the vector.*

*Proof.* As before, it is straightforward to see that $\mu^{\pi_\dagger\{s_0;a\}}(s_0) = (1 - \gamma)$ and

$$\forall i \in \{-2, -1\} : \mu^{\pi_\dagger\{s_0;a\}}(s_i) = \gamma \cdot \left(\frac{\delta}{2} - \frac{\delta}{4\theta} \cdot (\sum \widetilde{v}(s_j))\right),$$

and

$$\forall i \geq 1 : \mu^{\pi_\dagger\{s_0;a\}}(s_i) = \gamma \cdot \left((1 - \delta) \cdot \widehat{\mu}(s_i) + \frac{\delta}{2\theta} \cdot \widetilde{v}(s_i)\right).$$

$\square$

**Claim H.7.** *If* $\max\{\delta, 1 - \gamma\} \leq \frac{1}{4\|\widetilde{\mu}\|_1}$, *then*

$$\frac{\|\widetilde{\mu}\|_1}{\gamma(1-\delta)} \left( \sum_{j=-2}^{0} \mu^{\pi_\dagger}(s_j) \right) \leq 1/8$$

*Proof.*

$$\frac{\|\widetilde{\mu}\|_1}{\gamma(1-\delta)} \cdot \left( \sum_{j=-2}^{0} \mu^{\pi_\dagger}(s_j) \right) = \frac{\delta \cdot \gamma + (1 - \gamma)}{\gamma(1-\delta)} \cdot \|\widetilde{\mu}\|_1 \tag{37}$$

$$\leq \frac{1}{4} \cdot \|\widetilde{\mu}\|_1 \cdot (\delta + (1 - \gamma)) \leq \frac{1}{8} \tag{38}$$

$\square$

**Claim H.8.** *The 3SAT instance is satisfiable if and only if* (34) *holds.*

*Proof.* Given Claim H.6, the sub-matrix of $M$ corresponding to columns $s_{j \geq 1}$ equals

$$\frac{\gamma \cdot \delta}{2 \cdot \theta} \cdot \widetilde{M}.$$

Let $u_\infty[s]$ denote $u_\infty\left((M^T\lambda)(s)\right)$. For any $\lambda$,

$$\langle \mu^{\pi_\dagger}, u(M^T\lambda) \rangle = \sum_{j=-2}^{0} \mu^{\pi_\dagger}(s_j)u_\infty[s_j] + \left\langle \frac{\gamma \cdot (1-\delta)}{\|\widetilde{\mu}\|_1} \cdot \widetilde{\mu}, u(\frac{\gamma \cdot \delta}{2 \cdot \theta} \cdot \widetilde{M}^T\lambda) \right\rangle$$

$$= \sum_{j=-2}^{0} \mu^{\pi_\dagger}(s_j)u_\infty[s_j] + \left\langle \frac{\gamma \cdot (1-\delta)}{\|\widetilde{\mu}\|_1} \cdot \widetilde{\mu}, u(\widetilde{M}^T\lambda) \right\rangle$$

Multiplying both sides by $\frac{\|\widetilde{\mu}\|_1}{\gamma(1-\delta)}$, it follows that

$$\left| \frac{\|\widetilde{\mu}\|_1}{\gamma(1-\delta)} \cdot \langle \mu^{\pi_\dagger}, u(M^T\lambda) \rangle - \langle \widetilde{\mu}, u(\widetilde{M}^T\lambda) \rangle \right| \leq \frac{\|\widetilde{\mu}\|_1}{\gamma(1-\delta)} \cdot \left( \sum_{j=-2}^{0} |\mu^{\pi_\dagger}(s)u_\infty[s]| \right) \tag{39}$$

$$\leq \frac{\|\widetilde{\mu}\|_1}{\gamma(1-\delta)} \cdot \left( \sum_{j=-2}^{0} \mu^{\pi_\dagger}(s) \right) \tag{40}$$

$$\leq \frac{1}{8} \tag{41}$$

Therefore, by Lemma H.4, if the 3SAT is satisfiable, then there exists a $\lambda$ such that

$$\frac{\|\widetilde{\mu}\|_1}{\gamma(1-\delta)} \cdot \langle \mu^{\pi_\dagger}, u(M^T\lambda) \rangle \leq -\frac{1}{4} + \frac{1}{8} < 0.$$

If the 3SAT is not satisfiable, then by Lemma H.4, for all feasible $\lambda$,

$$\frac{\|\widetilde{\mu}\|_1}{\gamma(1-\delta)} \cdot \langle \mu^{\pi_\dagger}, u(M^T\lambda) \rangle \geq \frac{1}{4} - \frac{1}{8} > 0$$

which proves the claim. $\square$

## H.2 Proof of Theorem 4.5

**Statement:** *For $\mathcal{C} = \{c_p \text{ s.t. } p \in [1, \infty)\}$, it is NP-hard to determine whether the optimal value of problem (P2b) is greater than or equal to $\widehat{\rho}^{\pi_\dagger}$* We will prove the result using the same MDP as the previous Section.

We need to show the 3SAT instance is satisfiable if and only if

$$\exists p \in [1, \infty), \lambda \in \mathbb{R}^k : \lambda \geq 0 \wedge \left\langle u_p(\mathbf{\Phi}_\theta^T \lambda), \psi^{\pi_\dagger} \right\rangle < 0.$$

As before, let $M$ denote the submatrix of $\mathbf{\Phi}_\theta$ with only columns corresponding to $(s, \pi_\dagger(s))$. As in the previous section we note that because $\psi^{\pi_\dagger}(s, a)$ is 0 for all $s, a \neq \pi_\dagger(s)$, the above condition is equivalent to

$$\exists p \in [1, \infty), \lambda \in \mathbb{R}^k : \lambda \geq 0 \wedge \left\langle u_p(M^T \lambda), \mu^{\pi_\dagger} \right\rangle < 0. \tag{42}$$

We split the proof into two lemmas.

**Lemma H.9.** *If the 3SAT instance is satisfiable, then* (42) *holds.*

*Proof.* In this case, then the same proof as before basically holds. Formally, consider the $\lambda$ used before in the proof and consider the vector $M^T \lambda$. We need to show that there exists $p$ for which $\left\langle \mu^{\pi_\dagger}, u_p(M^T \lambda) \right\rangle$ is negative. Note however that

$$\left\langle \mu^{\pi_\dagger}, u_p(M^T \lambda) \right\rangle = \sum_{i=-2}^{k} \mu^{\pi_\dagger}(s_i) \cdot u_p \left( (M^T \lambda)(s_i) \right)$$

For fixed $x \neq 0$, $u_p(x)$ is a continuous function of $p$ and $\lim_{\infty} u_p(x) = u_\infty(x)$. If we show that all of the coordinates in $M^T \lambda$ are non-zero, this would imply that $\left\langle \mu^{\pi_\dagger}, u_p(M^T \lambda) \right\rangle$ converges to $\left\langle \mu^{\pi_\dagger}, u_\infty(M^T \lambda) \right\rangle < 0$ for large enough $p$ which proves the claim. Therefore, $\left\langle \mu^{\pi_\dagger}, u_p(M^T \lambda) \right\rangle$ is continuous in $p$ and letting $p$ be large enough proves the claim.

It remains to verify that all of the coordinates in $M^T \lambda$ used in the above proof were non-zero. Since all of the coordinates of $\widetilde{M}^T \lambda$ were non-zero by our construction of $\lambda$, we conclude that $M^T \lambda$ is non-zero on $s_{i \geq 1}$. For $i \in \{-2, -1\}$, given claim H.3, we have $\sum_{i \geq 1} \widetilde{v}(s_i) > 0$, for all rows $\widetilde{v}$ of $\widetilde{M}$, which means $M^T \lambda$ is strictly negative on these states. Finally, the entry corresponding to $s_0$ equals $-\mu^{\pi_\dagger}(s_0)$ in all rows of $M$, which implies $\left( M^T \lambda \right)(s_0)$ is strictly negative, proving the claim. $\square$

**Lemma H.10.** *If* (42) *holds, then the 3SAT instance is satisfiable.*

*Proof.* We assume that $\lambda \neq 0$ without loss of generality; if $\lambda = 0$ then $\left\langle \mu^{\pi_\dagger}, u_p(M^T \lambda) \right\rangle = 0$ which is not negative.
Letting $u_p[s]$ denote $u_p \left( (M^T \lambda)(s_i) \right)$ and $d_p[s]$ denote $\mu^{\pi_\dagger}(s) \cdot u_p[s]$, then $\left\langle \mu^{\pi_\dagger}, u_p(M^T \lambda) \right\rangle$ can be rewritten as

$$\left\langle \mu^{\pi_\dagger}, u_p(M^T \lambda) \right\rangle = \sum_{i=-2}^{\ell} \mu^{\pi_\dagger}(s_i) \cdot u_p[s_i] \tag{43}$$

$$= \sum_{i=-2}^{0} d_p[s_i] + \sum_{(24)} d_p[s] + \sum_{(25)} d_p[s] + \sum_{(26)} d_p[s] + \sum_{(27)} d_p[s] + \sum_{(28)} d_p[s] + \sum_{(29)} d_p[s] \tag{44}$$

where in the above, we have broken the sum in different parts, depending on what $s_i$ corresponds to and we have abused the notation by using the *set* (24) denote all states that *correspond* to Equation (24). Note that the sum corresponding to (24) consists of a single state. We will also use $s_1$ to denote this state.

We begin by stating some properties of $u_p$ and $d_p$.

**Claim H.11.** *Let $\gamma$ be close enough to 1 such that*

$$\frac{c\delta}{4\theta} \geq (1 - \gamma).$$

*Then for all rows $\tilde{v}$ from $\widetilde{M}$,*

$$\forall s \neq s_1 : |u_p[s]| \leq u_p[s_1] \tag{45}$$

*Proof.* We need to show that for all rows $v$ in $M$, $|v(s_1)| \geq |v(s)|$ for all $s \neq s_1$. For $i \geq 1$, this follows immediately from Claim H.3 and Claim H.6.

As for $s_0$, we need to show that

$$v(s_1) \geq |v(s_0)| \iff \frac{\gamma \cdot \delta}{2\theta} \widetilde{v}(s_1) \geq (1 - \gamma)$$

Note however that

$$\frac{\gamma \cdot \delta}{2\theta} \widetilde{v}(s_1) \geq \frac{\delta}{4\theta} \widetilde{v}(s_1) \geq (1 - \gamma)$$

by the assumption on $\gamma$. $\qquad\square$

**Claim H.12.** *For any $i \in [n]$, letting $\tilde{s}_i$ and $\tilde{s}'_i$ denote the states corresponding to (27) and (28) respectively.*

$$d_p[\tilde{s}_i] + d_p[\tilde{s}'_i] \geq 0.$$

*Proof.* To prove this, observe that

$$0.9a_0 - (a_i - b_i) + 0.9a_0 - (b_i - a_i) = 1.8a_0 \geq 0$$

There are therefore two possibilities: **(a)** Both $0.9a_0 - (a_i - b_i)$ and $0.9a_0 - (b_i - a_i)$ are non-negative. In this case, both $d_p[\tilde{s}_i]$ and $d_p[\tilde{s}'_i]$ are non-negative and the claim follows. **(b)** $0.9a_0 - (a_i - b_i) \geq 0$ and $0.9a_0 - (b_i - a_i) \leq 0$ or vice versa. Assume w.l.o.g that $0.9a_0 - (a_i - b_i) \geq 0$, i.e, $d_p[\tilde{s}_i] \geq 0$. In this case, $|0.9a_0 - (a_i - b_i)| \geq |0.9a_0 - (b_i - a_i)|$ and therefore, since $\mu^{\pi\dagger}(\tilde{s}_i) = \mu^{\pi\dagger}(\tilde{s}'_i)$, we conclude that $d_p[\tilde{s}_i] \geq |d_p[\tilde{s}'_i]|$ and therefore the claim follows. $\qquad\square$

We split the proof into several cases.
**Case 1:** There exists $s \in (25) \cup (26)$ such that $d_p[s] \geq 0$.
Let $\bar{s}$ be such a state In this case, define $\widetilde{d}_p[s]$ as follows

$$\widetilde{d}_p[s] = \begin{cases} 0 & \text{if} \quad s = \bar{s} \\ 0 & \text{if} \quad s \in (27) \cup (28) \\ d_p[s] & \text{if} \quad s = s_1 \\ -|d_p[s]| & \text{o.w.} \end{cases}$$

We first claim that $\sum d_p[s] \geq \sum \widetilde{d}_p[s]$. To prove this, note that $d_p[s] \geq \widetilde{d}_p[s]$ for all $s \notin (27) \cup (28)$. We therefore need to prove that

$$\sum_{s \in (27) \cup (28)} d_p[s] \geq 0,$$

which follows from Claim H.12. by summing over all $i \in [n]$.

Defining $\widetilde{S} := S \backslash (\{s_1, \overline{s}\} \cup (27) \cup (28))$, we note that

$$\sum_s \widetilde{d}_p[s] = d_p[s_1] - \sum_{s \in \widetilde{S}} |d_p[s_i]|$$

$$= \mu^{\pi\dagger}(s_1) \cdot u_p[s_1] - \sum_{s \in \widetilde{S}} \mu^{\pi\dagger}(s) \cdot |u_p[s_i]|$$

$$\overset{(45)}{\geq} \mu^{\pi\dagger}(s_1) \cdot u_p[s_1] - \sum_{s \in \widetilde{S}} \mu^{\pi\dagger}(s) \cdot |u_p[s_1]|$$

$$= |u_p[s_1]| \left( \mu^{\pi\dagger}(s_1) - \sum_{s \in \widetilde{S}} \mu^{\pi\dagger}(s) \right)$$

Defining $\widetilde{S}_+ := \widetilde{S} \backslash \{s_{-2}, s_{-1}, s_0\}$,

$$\widetilde{\mu}(s_1) - \sum_{s \in \widetilde{S}_+} \widetilde{\mu}(s) = 6n^2 + m - \frac{1}{2} - (2n - 1) \cdot 3n - m$$

$$= 6n^2 + m - \frac{1}{2} - 6n^2 + 3n - m$$

$$= 3n - \frac{1}{2} > \frac{1}{4},$$

Claim H.7 now gives a contradiction as the inner product in (42) becomes positive.
**Case 2:** $d_p[s] < 0$ for all $s \in (25) \cup (26)$ and $|a_j - b_j| \leq 0.9a_0$ for some $j \in [n]$.
In this case, we conclude that

$$a_i + b_i \in [0.9, 1.1]a_0 \quad \forall i \in [1, n]$$

Since either $a_i$ or $b_i$ must be strictly positive for some $i$ (because $\lambda \neq 0$), we conclude that $a_0 > 0$.
Similar to case 1, we introduce a new vector $\widetilde{d}_p$ such that $\sum_s d_p[s] \geq \sum_s \widetilde{d}_p[s]$.

Let $\widetilde{s}_j$ and $\widetilde{s}'_j$ denote the states corresponding to (27) and (28) respectively for $i = j$. Note that by assumption, $u_p[\widetilde{s}_j], u_p[\widetilde{s}'_j] \geq 0$. Assume without loss of generality that $u_p[\widetilde{s}_j] \geq u_p[\widetilde{s}'_j]$. Define the vector $\widetilde{d}_p$ as

$$\widetilde{d}_p[s] = \begin{cases} 0 & \text{if} \quad s \in (27) \cup (28) \backslash \{\widetilde{s}_j\} \\ d_p[s] & \text{if} \quad s \in \{s_1, \widetilde{s}_j\} \\ -|d_p[s]| & \text{o.w.} \end{cases}$$

As before, $\sum d_p[s] \geq \sum \widetilde{d}_p[s]$. More formally, for (27) and (28), we use Claim H.12 and for all the other states, we have $d_p[s] \geq \widetilde{d}_p[s]$.
We now claim that

$$\forall s \in (25) \cup (26) \cup (29) : |u_p[s]| \leq u_p[\widetilde{s}_j]. \tag{46}$$

This is because

$$(M^T \lambda)(\widetilde{s}_j) \geq \frac{(M^T \lambda)(\widetilde{s}_j) + (M^T \lambda)(\widetilde{s}'_j)}{2} = 3.6a_0,$$

while $|(M^T \lambda)(s)| \leq 2.35a_0$ for all $s \in (25) \cup (26) \cup (29)$. This is because $a_i + b_i \in [0.9 \cdot a_0, 1.1 \cdot a_0]$.

Defining $\widetilde{S} := (25) \cup (26) \cup (29)$, we conclude that

$$\sum_s d_p[s] \geq \sum_s \widetilde{d}_p[s]$$

$$= \mu^{\pi\dagger}(s_1) \cdot u_p[s_1] + \mu^{\pi\dagger}(\widetilde{s}_j) \cdot u_p[\widetilde{s}_j] + \sum_{s \in \widetilde{S}} \mu^{\pi\dagger}(s) \cdot -|u_p[s]| + \sum_{i=-2}^{0} \mu^{\pi\dagger}(s_i) \cdot -|u_p[s_i]|$$

$$\geq \mu^{\pi\dagger}(s_1) \cdot u_p[s_1] + \mu^{\pi\dagger}(\widetilde{s}_j) \cdot u_p[\widetilde{s}_j] + \sum_{s \in \widetilde{S}} \mu^{\pi\dagger}(s) \cdot -|u_p[\widetilde{s}_j]| + \sum_{i=-2}^{0} \mu^{\pi\dagger}(s_i) \cdot -|u_p[s_1]|$$

$$= \left( \mu^{\pi\dagger}(s_1) - \sum_{i=-2}^{0} \mu^{\pi\dagger}(s_i) \right) \cdot u_p[s_1] + \mu^{\pi\dagger}(\widetilde{s}_j) \cdot u_p[\widetilde{s}_j] + \sum_{s \in \widetilde{S}} \mu^{\pi\dagger}(s) \cdot -|u_p[\widetilde{s}_j]|$$

$$\geq \left( \mu^{\pi\dagger}(s_1) - \sum_{i=-2}^{0} \mu^{\pi\dagger}(s_i) \right) \cdot u_p[\widetilde{s}_j] + \mu^{\pi\dagger}(\widetilde{s}_j) \cdot u_p[\widetilde{s}_j] + \sum_{s \in \widetilde{S}} \mu^{\pi\dagger}(s) \cdot -|u_p[\widetilde{s}_j]|$$

$$= u_p[\widetilde{s}_j] \cdot \left( \mu^{\pi\dagger}(s_1) + \mu^{\pi\dagger}(\widetilde{s}_j) - \sum_{i=-2}^{0} \mu^{\pi\dagger}(s_i) - \sum_{s \in \widetilde{S}} \mu^{\pi\dagger}(s) \right)$$

Here, for the final inequality we have used the fact that $\mu^{\pi\dagger}(s_1) - \sum_{i=-2}^{0} \mu^{\pi\dagger}(s_i) \geq 0$, which follows from Claim H.7.

As before, note that

$$\widetilde{\mu}(s_1) + \widetilde{\mu}(\widetilde{s}_j) - \sum_{s \in \widetilde{S}} \widetilde{\mu}(s) = 6n^2 + m - \frac{1}{2} + 1 - 2n \cdot 3n - m \geq \frac{1}{2}$$

Claim H.7 now gives a contradiction as the inner product in (42) becomes positive.

**Case 3:** $d_p[s] < 0$ for all $s \in (25) \cup (26)$ and $|a_i - b_i| \geq 0.9a_0$ for all $1 \leq i \leq n$.

Similar as before, define $\widetilde{d}_p$ as

$$\widetilde{d}_p[s] = \begin{cases} 0 & \text{if} \quad s \in (27) \cup (28) \\ d_p[s] & \text{if} \quad s = s_1 \\ \min\{d_p[s], 0\} & \text{if} \quad s \in (29) \\ -|d_p[s]| & \text{o.w.} \end{cases}$$

As before, $\sum_s d_p[s] \geq \sum_s \widetilde{d}_p[s]$. Therefore,

$$\sum_s d_p[s] \geq \sum_s \widetilde{d}_p[s]$$

$$= \mu^{\pi\dagger}(s_1) \cdot u_p[s_1] - \sum_{s \in (25) \cup (26)} \mu^{\pi\dagger}(s) \cdot |u_p[s]| + \sum_{s \in (29)} \mu^{\pi\dagger}(s) \cdot \min\{u_p[s], 0\} - \sum_{i=-2}^{0} \mu^{\pi\dagger}(s_i) \cdot |u_p[s_i]|$$

$$\geq \mu^{\pi\dagger}(s_1) \cdot u_p[s_1] - \sum_{s \in (25) \cup (26)} \mu^{\pi\dagger}(s) \cdot |u_p[s_1]| - \sum_{s \in (29)} \mu^{\pi\dagger}(s) \cdot |u_p[s_1]| \mathbb{1}\left[u_p[s] < 0\right] - \sum_{i=-2}^{0} \mu^{\pi\dagger}(s_i) \cdot |u_p[s_1]|$$

$$= u_p[s_1] \cdot \left( \mu^{\pi\dagger}(s_1) - \sum_{s \in (25) \cup (26)} \mu^{\pi\dagger}(s) - \sum_{s \in (29)} \mu^{\pi\dagger}(s) \mathbb{1}\left[u_p[s] < 0\right] - \sum_{i=-2}^{0} \mu^{\pi\dagger}(s_i) \right)$$

Assume for the sake of contradiction that the 3SAT instance is not satisfiable. There are two possibilities. The first is that for at least one $s \in (29)$, we have $u_p[s] \geq 0$. In that case,

$$\widetilde{\mu}(s_1) - \sum_{s \in (25) \cup (26)} \widetilde{\mu}(s) - \sum_{s \in (29)} \widetilde{\mu}(s) \mathbb{1}\left[u_p[s] < 0\right] \geq \frac{1}{2}$$

As before, Claim H.7 gives a contradiction. Otherwise, $u_p[s] < 0$ for all $s \in (29)$. In that case, we have

$$\left\langle u_\infty(\widetilde{M}^T \lambda), \widetilde{\mu} \right\rangle = 6n^2 + m - 1/2 - 2n \cdot 3n - m < -1/2,$$

which contradicts Proposition H.2 as $\lambda \neq 0$. $\qquad\square$

# I  Proofs of Appendix B

In this section, we provide a more formal treatment of the results in Appendix B, formally stating and proving these results.

**Proposition I.1.** *Assume that condition* (3) *holds. Set $\widehat{\epsilon}$ as*

$$\widehat{\epsilon} = \min_{s, a \neq \pi_\dagger(s)} \left[ \widehat{\rho}^{\pi_\dagger} - \widehat{\rho}^{\pi_\dagger\{s;a\}} \right].$$

*Consider the following policy*

$$\pi_{\mathcal{D}}(a|s) = \frac{\mathbb{1}\left[a \in \Theta_s^\epsilon \cup \{\pi_\dagger(s)\}\right]}{|\Theta_s^\epsilon| + 1}. \tag{47}$$

*Equation* (47) *characterizes the solution to the optimization problems* (P2a) *and* (P2b) *with parameters $\epsilon = \epsilon_\dagger$ and $\epsilon = \min\{\widehat{\epsilon}, \epsilon_{\mathcal{D}}\}$ respectively. Furthermore, in both cases $\overline{\rho}^{\pi_{\mathcal{D}}} = \widehat{\rho}^{\pi_{\mathcal{D}}}$*

*Proof.* Given Theorem 5.1, Theorem 5.2, and Lemma D.3, it suffices to show that if $\epsilon \leq \widehat{\epsilon}$, the solution to the optimization problem

$$\max_{\psi \in \Psi} \left\langle \psi, \widehat{R} \right\rangle \tag{P4}$$

$$\text{s.t.} \ \left\langle \psi^{\pi_\dagger\{s;a\}} - \psi^{\pi_\dagger}, \psi \right\rangle \geq 0 \quad \forall s, a \in \Theta^\epsilon,$$

corresponds to the occupancy measure of policy $\pi_{\mathcal{D}}$ defined by Equation (47). Namely, the optimization problems (P2a) and (P2b) correspond to the optimization problem (P3) with parameters $\epsilon = \epsilon_\dagger$ and $\epsilon = \min\{\widehat{\epsilon}, \epsilon_{\mathcal{D}}\}$ respectively. Since $\widehat{\epsilon} \leq \epsilon_\dagger$, the primal feasibility condition in Lemma E.1 implies that the solution to the above optimization problem characterizes both cases ((P2a) and (P2b)).

Now, due to Lemma D.3, we have

$$\psi \in \Psi \iff \psi \succcurlyeq 0 \wedge \forall s : \sum_a \psi(s, a) = (1 - \gamma)\sigma(s) + \gamma \sum_{\tilde{s}, \tilde{a}} P(\tilde{s}, \tilde{a}, s)\psi(\tilde{s}, \tilde{a}).$$

Since $P(\tilde{s}, \tilde{a}, s)$ is independent of $\tilde{a}$, the second condition is equivalent to

$$\forall s : \sum_a \psi(s, a) = (1 - \gamma)\sigma(s) + \gamma \sum_{\tilde{s}} \left( P(\tilde{s}, \pi_\dagger(\tilde{s}), s)(\sum_{\tilde{a}} \psi(\tilde{s}, \tilde{a})) \right),$$

which, due to (12), is equivalent to

$$\sum_a \psi(s, a) = \mu(s).$$

Furthermore, given the independence of the transition distributions from policies, we have the following

$$(\psi^{\pi_\dagger\{s;a\}} - \psi^{\pi_\dagger})(\tilde{s}, \tilde{a}) = \begin{cases} \mu(s) & \text{if} \quad (\tilde{s}, \tilde{a}) = (s, a) \\ -\mu(s) & \text{if} \quad (\tilde{s}, \tilde{a}) = (s, \pi_\dagger(s)) \\ 0 & \text{o.w} \end{cases}. \tag{48}$$

Therefore, the constraint $\left\langle \psi^{\pi_\dagger\{s;a\}} - \psi^{\pi_\dagger}, \psi \right\rangle \geq 0$ is equivalent to $\psi(s,a) \geq \psi(s, \pi_\dagger(s))$. Furthermore, note that

$$
\begin{aligned}
(s,a) \in \Theta^\epsilon &\iff \widehat{\rho}^{\pi_\dagger\{s;a\}} - \widehat{\rho}^{\pi_\dagger} = -\epsilon_\dagger \\
&\iff \left\langle \psi^{\pi_\dagger\{s;a\}} - \psi^{\pi_\dagger}, \widehat{R} \right\rangle \leq -\epsilon_\dagger \\
&\iff \widehat{R}(s,a) - \widehat{R}(s, \pi_\dagger(s)) = -\frac{\epsilon_\dagger}{\mu(s)} \\
&\iff a \in \Theta_s^\epsilon.
\end{aligned}
$$

Putting it all together, the optimization problem (P3) is equivalent to

$$
\begin{aligned}
\max_{\psi} \ & \left\langle \widehat{R}, \psi \right\rangle \\
\text{s.t.} \quad & \psi(s, \pi_\dagger(s)) \leq \psi(s,a) \quad \forall s, a \in \Theta_s^\epsilon \\
& \sum_a \psi(s,a) = \mu(s) \quad \forall s \in S \\
& \psi(s,a) \geq 0 \quad \forall(s,a).
\end{aligned}
$$

Note that the maximization is now over all vectors $\psi \in \mathbb{R}^{|S| \cdot |A|}$ as the constraint $\psi \in \Psi$ has been made explicit. Furthermore, given Lemma D.3 and Equation (5), any vector $\psi$ satisfying the last two constraints (the bellman constraints) corresponds to a policy $\pi$ through

$$
\pi(a|s) = \frac{\psi(s,a)}{\mu(s)}.
$$

In other words, probability of choosing $a$ in state $s$ is proportional to $\psi(s,a)$.

Now, let us analyze the solution to this optimization problem which we will denote by $\psi_{\max}$. This solution $\psi_{\max}$ exists, since the optimization problem is maximizing a continuous function on a closed and bounded set.

We first claim that if $a \notin \Theta_s^\epsilon \cup \{\pi_\dagger(s)\}$, then $\psi_{\max}(s,a) = 0$. If this is not the case, then $\psi_{\max}$ is not optimal. Concretely, consider the following vector $\psi$

$$
\psi(\tilde{s}, \tilde{a}) = \begin{cases} \psi_{\max}(\tilde{s}, \tilde{a}) + \frac{1}{|\Theta_s^\epsilon|+1} \psi_{\max}(s,a) & \text{if} \quad \tilde{s} = s \wedge \tilde{a} \in \Theta_s^\epsilon \cup \{\pi_\dagger(s)\} \\ 0 & \text{if} \quad \tilde{s} = s \wedge \tilde{a} = a \\ \psi_{\max}(\tilde{s}, \tilde{a}) & \text{o.w.} \end{cases} \quad .
$$

In other words, we uniformly spread the probability of choosing action $a$ in state $s$ over the set $\Theta_s^\epsilon \cup \{\pi_\dagger(s)\}$. The vector $\psi$ still satisfies the constraints: if $\tilde{a} \in \Theta_s^\epsilon$, $\psi(s, \pi_\dagger(s)) - \psi(s, \tilde{a}) = \psi_{\max}(s, \pi_\dagger(s)) - \psi_{\max}(s, \tilde{a})$ and the objective has strictly improved because

$$
\widehat{\rho}^{\pi_\dagger\{s;a\}} - \widehat{\rho}^{\pi_\dagger} \leq -\widehat{\epsilon} \leq -\epsilon \implies \widehat{R}(s,a) \leq \widehat{R}(s, \pi_\dagger(s)) - \frac{\epsilon}{\mu(s)}.
$$

Since $a \notin \Theta_s^\epsilon$, the inequality is strict and therefore

$$
\forall \tilde{a} \in \Theta_s^\epsilon \cup \{\pi_\dagger(s)\} : \widehat{R}(s, \tilde{a}) > \widehat{R}(s,a).
$$

This means that $\psi$ was not optimal, contradicting the initial assumption.

Now note that if $\psi_{\max}(s,a) > \psi_{\max}(s, \pi_\dagger(s))$ for some $a \in \Theta_s^\epsilon$, then again $\psi_{\max}$ isn't optimal as we could replace it with

$$
\psi(\tilde{s}, \tilde{a}) = \begin{cases} \psi_{\max}(\tilde{s}, \tilde{a}) + \dfrac{\psi_{\max}(s,a) - \psi_{\max}(s, \pi_\dagger(s))}{|\Theta_s^\epsilon| + 1} & \text{if} \ \ \tilde{s} = s \wedge \tilde{a} \in \Theta_s^\epsilon \cup \{\pi_\dagger(s)\} \backslash \{a\} \\ \psi_{\max}(\tilde{s}, \tilde{a}) - \dfrac{|\Theta_s^\epsilon|(\psi_{\max}(s,a) - \psi_{\max}(s, \pi_\dagger(s)))}{|\Theta_s^\epsilon| + 1} & \text{if} \ \ \tilde{s} = s \wedge \tilde{a} = a \\ \psi_{\max}(\tilde{s}, \tilde{a}) & \text{o.w.} \end{cases} \quad .
$$

Intuitively, since the action $a$ was being chosen with strictly higher probability than action $\pi_\dagger(s)$, we have uniformly spread this excess probability among the set $\Theta_s^\epsilon \cup \{\pi_\dagger(s)\}$. This vector would still be feasible as $\psi(s,a) = \psi(s, \pi_\dagger(s))$ and would be strictly better in terms of utility as $\widehat{R}(s, \pi_\dagger(s)) > \widehat{R}(s,a)$. This contradicts our initial assumption and therefore $\psi_{\max}(s,a) = \psi_{\max}(s, \pi_\dagger(s))$ for all $a \in \Theta_s^\epsilon$.

Since the occupancy measure $\psi_{\max}$ satisfies $\psi_{\max}(s,a) = 0$ for all $a \notin \Theta_s^\epsilon \cup \{\pi_\dagger(s)\}$ and $\psi_{\max}(s,a) = \psi(s, \pi_\dagger(s))$ for all $a \in \Theta_s^\epsilon$, we conclude that it is the occupancy measure for the policy $\pi_\mathcal{D}$ as defined in Equation (47).

In order to prove $\overline{\rho}^{\pi_\mathcal{D}} = \widehat{\rho}^{\pi_\mathcal{D}}$, first note that for $(s,a) \in \Theta^\epsilon$

$$\left\langle \psi^{\pi_\dagger\{s;a\}} - \psi^{\pi_\dagger}, \psi^{\pi_\mathcal{D}} \right\rangle = \mu(s)(\psi^{\pi_\mathcal{D}}(s,a) - \psi^{\pi_\mathcal{D}}(s, \pi_\dagger(s)))$$
$$= \mu(s)^2(\pi_\mathcal{D}(a|s) - \pi_\mathcal{D}(\pi_\dagger(s)|s)) = 0,$$

where we used Equation (48) and Equation (47). Therefore

$$\overline{\rho}^{\pi_\mathcal{D}} - \widehat{\rho}^{\pi_\mathcal{D}} = \left\langle \overline{R} - \widehat{R}, \psi^{\pi_\mathcal{D}} \right\rangle$$
$$\overset{(i)}{=} \sum_{(s,a) \in \Theta^\epsilon} \alpha_{s,a} \left\langle \psi^{\pi_\dagger\{s;a\}} - \psi^{\pi_\dagger}, \psi^{\pi_\mathcal{D}} \right\rangle$$
$$= \sum_{(s,a) \in \Theta^\epsilon} \alpha_{s,a} \cdot 0$$
$$= 0,$$

where $(i)$ follows from Lemma E.3 in the known parameter case and Lemma F.3 in the unknown parameter case. $\qquad\square$

## J  Additional experiments

In this section, we provide the worst-case score of the policy from the defender's perspective , i.e., the value of the inner minimization in (P2b), for our policy. Given the results in Section 5.2 (Theorems 5.3 and 5.2), by construction of $\pi_\mathcal{D}$, this minimum is achieved for $R = \widehat{R}$ and therefore the worst-case score equals $\widehat{\rho}^{\pi_\mathcal{D}}$. As mentioned in Section 5.2, $\widehat{\rho}^{\pi_\mathcal{D}}$ is *a certificate* in that we are guaranteed $\overline{\rho}^{\pi_\mathcal{D}} \geq \widehat{\rho}^{\pi_\mathcal{D}}$ as long as $\epsilon_\mathcal{D} \geq \epsilon_\dagger$.

The results are shown in Figure 6. As shown in the Figure, the value of $\widehat{\rho}^{\pi_\mathcal{D}}$ is larger when $\epsilon_\mathcal{D} < \epsilon_\dagger$. This is in line with results of Section 5.2 as in this case, $\pi_\mathcal{D} = \pi_\dagger$ and $\pi_\dagger$ is the optimal policy in $\widehat{R}$. In addition, when $\epsilon_\mathcal{D} < \epsilon_\dagger$, the value of $\widehat{\rho}^{\pi_\mathcal{D}}$ increases with $\epsilon_\dagger$. This is because when we increase the attack parameter $\epsilon_\dagger$, the policy $\pi_\dagger$ needs to become optimal in $\widehat{R}$ with a larger margin, which causes its score to increase.

We note that for both environments, when no defense is employed, i.e, when using the policy $\pi_\dagger$, we will obtain the same worst-case value when $\epsilon_\mathcal{D} < \epsilon_\dagger$ as $\pi_\mathcal{D} = \pi_\dagger$, and we will obtain the worst-case score $-\infty$ when $\epsilon_\mathcal{D} \geq \epsilon_\dagger$.

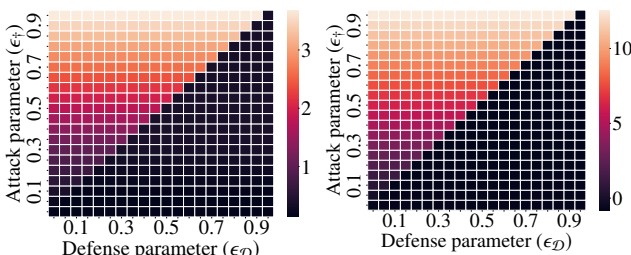

Figure 6: The value of the worst-case score from the defense perspective, i.e., $\widehat{\rho}^{\pi_\mathcal{D}}$ under different values of $\epsilon_\dagger, \epsilon_\mathcal{D}$ for the Navigation (left) and Grid world (right) environments.

