# OpenReview forum: "Defense Against Reward Poisoning Attacks in Reinforcement Learning"
_TMLR — Accepted by TMLR_

### Review · Reviewer_tuvE · 2022-10-12

**Summary Of Contributions:**

This paper presents a theoretical framework for adversarial attack of the reward function, where the attack is made at training time, and modifies the reward function so that the optimal policy becomes a target policy. The goal of the defender, who knows that the attacker chose a reward function that minimizes a cost from a known class of functions, is to find the best policy for the worst original reward function possible.

The paper makes a number of theoretical contributions, including an explicit way of deriving the best response policy in Theorem 5.1 for a specific cost. The are some experiments that validate the theoretical contributions.


**Audience:**

Yes

**Broader Impact Concerns:**

No broader impact concerns, this is a theoretical RL paper.

**Claims And Evidence:**

Yes

**Requested Changes:**

It took me some time to realize what the problem statement was, and I strongly suggest the authors to work on the wording to facilitate the understanding of e.g. what (P2a) is. If I understand correctly, (P2a) can be seen as the following:
  - for all existing (deterministic) policy there are a collection of rewards that make this policy optimal (this is probably not true since it depends on the dynamics of the MDP you are considering, but let's assume it is true),
  - for each target policy, the attacker will pick the reward function that minimizes a cost out of all the compatible reward functions,
  - as a strategy, the defender wants to pick the policy that will maximize the minimum performance in terms of all the rewards that the attacker could have picked.


On the definition of P1: I am bit surprised by the feasibility results mentioned by the authors since if the attack parameter is positive a solution does not necessarily exist. For example, think of an MDP where the dynamics are independent from the actions taken. In this setup, all policies are optimal and have the same value and therefore no single policy can dominate another one with a margin.


I believe that Theorem 4.3. is probably true but the proof seems erroneous at the moment.
$$\rho^{R, \pi} = \pi(a_1) + x(\\pi(a_2) - \pi(a_1) - \pi(a_3) - pi(a_4)) = \pi(a_1) + x(\\pi(a_2) - (1 - \pi(a_2))) = \pi(a_1) + 2x(\\pi(a_2) - 0.5)$$
Therefore, taking any policy with $\pi(a_2)=0.5$ is enough to bound the score.


Theorem 4.4 is central to the paper, as it is the base for the proof of Theorem 5.1. However, I find the proof to be a hand wavy at the moment. Please expand on the current version of the proof. For example, I find it surprising that "the only constraint in (P1’) affected by the value of $\tilde{R}(\tilde{s}, \tilde{a})$ is the constraint in $\tilde{s}, \tilde{a}$. If a reward function is modified in a single state action, the value function (or the score) is modified for more than this single state because of the dynamics in the MDP. Even if this turns out to be true, the justification is not convincing at the moment.


You mention that your method is computationally tractable. Please provide the computational complexity of your method (from a back-of-the-envelope calculation, it seems to be $O(|S|^4|A|^4)$ only to define the constraints.


In the experimental section, make sure that you mention using a solver for solving the convex optimization problem defined in Theorem 5.1 (it is understandable but it could be made clearer), and please provide some discussion around what the policy $\pi^\mathcal{D}$ turns out to be (qualitatively) for the two environments.


Not crucial but nice to have: maybe elaborate on the connection between (P1’) and (P1), which seems to be related to the line of work on policy iteration by using a simplex like method: \
“On the complexity of solving Markov decision problems.” Littman et al. 1995 \
“The simplex and policy-iteration methods are strongly polynomial for the markov decision problem with a fixed discount rate.” Ye 2011


**Strengths And Weaknesses:**

Strengths:
  - The paper proposes a framework for adversarial attacks.
  - There are a number of theoretical results around the proposed framework.
  - Theorem 5.1 reframes the problem in a way that can be solved with convex optimization methods.

Weaknesses:
  - Some of the proofs are unclear (see requested changes).
  - It is hard for the reader to really grasp how the described problem is realistic at the moment (see requested changes).
  - (Not relevant for TMLR guidelines). I have a hard time picturing how this problem formulation can scale to non-trivial environments where pre-computing occupancy measures for $|S| \times |A|$ is computationally unfeasible.

---

> ### Author Response · Authors · 2022-11-23
> **Thank you for your feedback!**
>
> Thank you for your comments and valuable suggestions. We are glad to see that you find this paper to have a number of theoretical contributions. Please find below our response to your comments.
>
> -----
>
> **Comments about the problem formulation**
>
>
> > ... suggest the authors work on the wording to facilitate the understanding of e.g. what (P2a) is. …
>
>
> We thank the reviewer for this suggestion. Our defense consists of the following three steps:
> - We calculate the set of all possible rewards $R$ such that an attack on $R$  *could* lead to the solution $\widehat{R}$, i.e., $\widehat{R} \in \mathcal{A}(c,R, \pi_{\dagger}, \epsilon_{\dagger})$.
> - For each policy, the worst-case score for the policy equals $\min_{R, c} \rho^\pi$ where the minimum is over all reward functions $R$ calculated previously and all cost functions $c\in \mathcal{C}$.
> - We find a policy with maximum worst-case score.
>
> We have modified the explanation after the optimization problem (P2a) to emphasize this point.
>
> Additionally, we have added a new figure (Figure 2 in Section 3.3) which illustrates the problem setting studied in this paper and the above steps. We have also moved one figure from the appendix to the main part of the paper, Figure 3, which illustrates the attack and defense strategies.
>
>
> > On the definition of P1: …
>
> We note that in our model, the reward function depends both on the state *and* the action taken in the state. As such, even if the transition probabilities are independent of the actions, the score of different policies is not necessarily the same; while all policies visit the same states with the same frequencies, they have different rewards depending on the actions they take.
>
>
> -----
>
> **Comments about the formal results**
>
> > … Theorem 4.3. is probably true but the proof seems erroneous at the moment.
>
> We have modified the proof and added clarifications. Essentially, we can assume that $\pi(a_2) < \frac{1}{3}$ without loss of generality since $\widehat{R}$ is symmetric with respect to $a_2, a_3, a_4$. Formally, while it is true that a policy with $\pi(a_2)\ge 0.5$ would have bounded score for the choice of $R’ = (1-x, x, -x, -x)$, since $\hat R$ is symmetric with respect to $a_2, a_3, a_4$, we can similarly consider $R’=(1-x, -x, x, -x)$ and $R’(1-x, -x, -x, x)$. As for any $\pi$ at least one of $\pi(a_2)$, $\pi(a_3)$, and $\pi(a_4)$, is strictly less than $0.5$, the solution to (P2a) is unbounded from below because of the inner minimization over $R$.
>
> > Theorem 4.4 is central to the paper … Please expand on your current version of the proof …
>
> We have expanded on the steps of the proof for clarification. Intuitively, given the optimization problem (P1’),  the attacker would never increase the reward of a non-target state-action pair $s, a\ne \pi_\dagger(s)$, as this would increase the objective it is minimizing, and would never help satisfy the constraints.
>
> -----
>
> **Comment about computational tractability**
>
> > You mention that your method is computationally tractable. Please provide the computational complexity of your method…
>
> We thank the reviewer for this suggestion and have added a discussion for the computational complexity in Section 5.1. The overall complexity of our approach is $O(|S|^4 |A|^4)$. We note that this is the same computational complexity of the attack problem as they both involve solving LPs with the same size (up to a constant factor).
>
> -----
>
> **Comment about experiments**
>
> >In the experimental section, make sure that you mention using a solver for solving the convex optimization problem defined in Theorem 5.1 (it is understandable but it could be made clearer)...
>
> We thank the reviewer for this suggestion. We have added a discussion in the beginning of the experiments section to emphasize this point.
>
> -----
>
> Thank you again for your helpful feedback. Please let us know if you have any additional comments.

---

### Review · Reviewer_uuwj · 2022-11-10

**Summary Of Contributions:**

At its core, the paper provides a characterization of the set of "plausibly" optimal policies under an adversarial perturbation of the reward function, and then uses this result to either:
- derive policies that are robust (i.e. defensive) against the perturbation
- show an impossibility result
depending on which assumptions are placed on the adversarial perturbation.

In particular the author's analysis considers an adversary that tries to induce the agent to adopt a target policy by perturbing the reward function to make this target policy optimal. The agent is aware (to varying degrees of details) of this perturbation, and tries to learn a policy that will provably perform as well as possible under the original (i.e. unperturbed) reward. Since the agent cannot give any such guarantees for fully general perturbations, the authors focus on perturbed rewards that within an epsilon-ball of the real reward according to some l_p norm.

The case where this approach gives the more insightful results are for l_2 and l_1 norms, where the authors show remarkable levels of characterization of the solution set, allowing the agent to recover a optimal-in-class policy (i.e. among all plausible policies induced by the epsilon ball) . This is achieved by characterizing the "plausible" policies in terms of KKT conditions on the occupancy measure of the policies and the perturbed reward, and then taking a maximum over this conservative set.

**Audience:**

Yes

**Broader Impact Concerns:**

Overall the paper covers topic related to safety in RL, which has the potential to be a sensitive topic. However due to the highly theoretical nature of the contribution my concerns of real-world unethical use of these results are minimal.

**Claims And Evidence:**

Yes

**Requested Changes:**

Overall I think the paper is comfortably above the acceptance threshold, but there is still clear room for more polish.

From the two main weaknesses:
- The authors should address which of these results can be extended to large or continuous state-spaces, and how hard such an extension would be.
- The authors should address the problem of the sub-optimality gap of the defensive/robust policy. Proving rigorous results quantifying this gap would be an interesting contribution on its own, but at least a discussion on how much the adversary can exploit the defensive mechanism to induce sub-optimal policies should be discussed.

For the minor weaknesses:
- Clarify/rigorously prove the comment on page 6
- Discuss better how $\epsilon_{\mathcal{D}}$ should be chosen in practice, and highlight the trade-offs of over vs underestimation.

**Strengths And Weaknesses:**

The strong points of the paper are:
- leveraging standard tools from robust statistics to derive a natural formulation for the adversarial perturbation problem, and a natural and pretty complete analysis that includes pretty efficient (i.e. polynomial) formulation to compute robust policies.
- a very clean exposition with a very good flow, and that covers most of the natural question that arise in this setting
- providing both feasible and unfeasible scenarios, and exploring the implications both in terms of the reward as well as computations

The paper has two main weaknesses. The first is that it takes an exclusively finite/discrete space perspective:
- from the theoretical perspective by relying extensively on iterations over finite actions/states, and on discrete KKT conditions
- from the practical perspective by relying on general constrained convex solvers, which do not immediately extend to an infinite number of constraints
The second is that despite providing a provably defensive policy (i.e. one that will achieve at least as much return under $\bar{R}$ than under $\hat{R}$), they provide no guarantee on the gap between the true (non-attacked) optimal policy and the difensive policy, or otherwise any upper bound on the maximum achievable performance by an attacked agent.

There are a few minor weaknesses regarding clarity that lead me to doubt (possibly due to misunderstandings) the soundness of some results:
- On page 6 the authors discuss a $\pi_{\cancel{\pi}_\dagger}$ policy, and claim that for any policy $\bar{\rho}^{\cancel{\pi}_\dagger} \geq \hat{\rho}^{\cancel{\pi}_\dagger}$. This does not seem intuitive to me, for example when $\bar{R} = \hat{R}$ this would clearly not be the case.
- The authors repeatedly state that $\epsilon_{\mathcal{D}}$ should be taken as large as $\epsilon_\dagger$, but while they describe in detail the disadvantages of selecting a too small value, I do not seem to find any discussion of the drawbacks of taking it too large. This would lead to the counterintuitive conclusion that it is always better to just select an extremely large epsilon (e.g. as large as the norm of $\hat{R}$).
- Thm. 5.3 seems to suggest that underestimating $\epsilon_{\mathcal{D}}$ will lead to the only feasible policy being $\pi_\dagger$. This again leads to the counterintuitive conclusion that one can easily estimate $\epsilon_\dagger$ by doing binary search on the epsilon value until a threshold is found where $\pi_\dagger$ is not the only solution in the set anymore.

Another very minor weakness is the $R'$ notation used in P1. While it is more general, AFAIU in the papr $R'$ is always set to $\bar{R}$, and it would be more straightforward to introduce it as such (no need to pay the cognitive price of a more general notation if it is not used).

---

> ### Author Response · Authors · 2022-11-23
> **Thank you for your feedback!**
>
> Thank you for your comments and valuable suggestions. We are glad to see that you find the paper to be comfortably above the acceptance threshold. Please find below our response to your comments.
>
> -----
>
> **Response to main concerns**
>
> > ... which of these results can be extended to large or continuous state-spaces, and how hard such an extension would be …
>
> The computational complexity of our defense is similar to the attack problem as they both involve solving linear programs with similar sizes. As such, we believe that our defense should be computationally tractable as long as the attack problem is. For large scale MDPs however, RL solutions typically use function approximation. As the attack problem we study is based on prior work, in order to extend our results to these settings, a first question is how the attack problem needs to be modified. While studying both the attack and defense problems would likely pose new challenges, we believe that the general formulation we present in our paper, as well as the techniques we use would be useful in this case as well. We have added a further discussion in Section 7 (*Unknown-model and scalability*) that emphasizes these points.
>
> > ... sub-optimality gap of the defensive/robust policy …
>
>
> We thank the reviewer for this suggestion. While we believe that it would be interesting to study the sub-optimality gap, and in particular, compare the sub-optimality gap of our defense to the one of the target policy, the set of corresponding results would be orthogonal to the current set of the results we have in the paper. We have added additional discussion to Section 7 (in paragraph *Beyond the worst-case utility*) that illustrates how these sub-optimality gaps compare through an example. The example is based on Figure 3 (in the latest submission, placed at the beginning of Section 5), which we moved from the appendix to the main part of the paper. Note that one can generally try to optimize performance relative to the target policy. However, as discussed in Section 7 (in paragraph *Beyond the worst-case utility*), this leads to a different optimization problem, and a possibly weaker goal since the target policy can have arbitrarily bad utility under the true reward function – our defense has a provable guarantee on the worst-case return.
>
> -----
>
> **Response to minor concerns**
>
> > On page 6 the authors discuss a $\pi_{\not \pi_{\dagger}}$ policy and claim that for any policy $\hat{\rho}^{\pi_{\not \pi_{\dagger}}} = \bar{\rho}^{\pi_{\not \pi_{\dagger}}}$. This does not seem intuitive to me...
>
> We note that the claim holds for any policy $\pi_{\not \pi_{\dagger}}$ s.t. $\pi_{\not \pi_{\dagger}}(\pi_{\dagger}(s)|s) = 0$ for all $s$. In the case of $\overline{R}=\widehat{R}$, the two scores are equal as $\hat{\rho}^{\pi_{\not \pi_{\dagger}}} = \overline{\rho}^{\pi_{\not \pi_{\dagger}}}$.
>
> > ... describe in detail the disadvantages of selecting a too small value, I do not seem to find any discussion of the drawbacks of taking it too large.
>
> We discuss this after Theorem 5.3 and further discuss this in Section 7 (Selecting $\epsilon_D$ and non-oblivious attack). The main issue with taking a small value of $\epsilon_D$ is that when the attacker did not poison the reward function, i.e., $\overline{R}=\widehat{R}$, the policy $\pi_\dagger$ is optimal, while the defense mechanism would (mistakenly) choose a suboptimal policy. See also our newly added discussion about sub-optimality gaps in paragraph *Beyond the worst-case utility* of Section 7, which indicates that $\epsilon_D$ is important for bounding the sub-optimality gap of the defense policy.
>
> > This again leads to the counterintutive conclusion that one can easily estimate $\epsilon^{\dagger}$ by doing binary search…
>
> This search strategy does not work when the attacker does not modify the reward function, i.e., when  $\widehat{R} = \overline{R}$ already forces the target policy. If the constraints in (P1) are satisfied by a margin larger than $\epsilon^{\dagger}$ when $\widehat{R} = \overline{R}$, the search strategy would overestimate $\epsilon^{\dagger}$ and decrease the agent’s performance.
>
> > Another very minor weakness is the $R’$ notation used in (P1). While it is more general, AFAIU in the paper $R’$ is always set to $\overline R$…
>
> This is a subtle but important notational detail. For example, we use the attack function $\mathcal A$ to define our problem statement: the optimization problems (P2a) and (P2b). In these optimization problems, $\mathcal A$ takes a generic reward function $R$ as its input, i.e., $R’$ is set to $R$.
>
> -----
>
> Thank you again for your helpful feedback. Please let us know if you have any additional comments.

---

### Review · Reviewer_5huZ · 2022-11-10

**Summary Of Contributions:**

The paper propose a defense mechanism to protect against reward perturbation in reinforcement learning. They assume a scenario where the goal of the attacker is to change the reward function in such a way that the target policy of the attacker is optimal under this new reward while staying as close as close possible to the original reward function.
To defend against such attacks the author first propose a min-max formulation where the agent has access to the perturbed reward function but not the true reward function, with this min-max formulation the agent is trying to find the optimal policy to solve the worst case scenario.
They then provide several theoretical results about the guarantees of solving such problems. They show that in the most general setting no guarantees can be obtained. However in specific cases, such as when the cost function of the attacker is the L2-norm, they show that an optimal policy against the attacker can be found by solving a convex optimization problem.
Finally, they show in a Navigation and Grid world experiment how the proposed approach indeed make the agent more resilient to such attacks.

**Audience:**

Yes

**Broader Impact Concerns:**

The author mention in the intro that AI models should be "technically robust and resilient to security threat". This is indeed very important and this paper is a good step in this direction. However, I think that the paper should clearly mention the limit of the proposed approach to avoid giving a false sense of security.

**Claims And Evidence:**

Yes

**Requested Changes:**

- I strongly encourage the author to include a discussion about the necessary assumptions, such as ergodicity, on how and why they are important for the results. As well as discuss, the limitations that such assumptions bring.
- I strongly encourage the author to provide more details about the computation of $\psi^{\pi_\dagger}$ and other occupancy measures used for solving the different problems.
- Clarify whether (P3) is used to solve (P2a) and (P2b).

**Strengths And Weaknesses:**

Strength:
The paper is quite complete it propose a clearly formulated problem and propose a theoretically sound solution. The theoretical results are quite solid and offers an elegant solution to the problem. The experiments also clearly show the robustness of the agent to such attacks.

Weakness:
- The main paper lacks some important details that could make the paper easier to read:
    - Some theoretical results would benefit from a small discussion about the implications of the result and why such result is useful. Right some of this is left to the reader to figure out. For example, it's not clearly explained why they introduce the reformulation (P1'), this is only clear from the appendix. The proof of the equivalence of (P1) and (P1'), is in the appendix, however I think a short comment about why this holds would help with reading the paper.
    - Some other details, for example the fact that P3 is linear in $\psi$ would be worth mentioning in the main text. Or some indications about the structure of the proofs, or some more intuition about the results would be welcome to make the paper easier to read. For example the paper introduces $\Theta^\epsilon$ without any explication, a short explanation on why this is needed could be valuable.
- Although the code is available, the main paper is lacking some crucially important details for reproducing the results. From my understanding of the results of the paper, solving the different problems in the paper requires to compute the state-action occupancy measure for the target policies $\psi^{\pi_\dagger}$. However the details on how such quantities is omitted from the paper and the appendix. I believe such details are crucial because those quantities can be hard to compute for example if we don't have access to the state transition probabilities which is usually the case, and leads to approximation which needs to be mentioned in the paper. Please correct me if those quantities are not needed for solving the different problems, and provide more details on what quantities are required to solve the different problems and how they are computed.
- The authors mention in the appendix that they use a convex programming solver to solve (P1'), (P2a) and (P2b). From my understanding, to solve (P2a) and (P2b), the authors actually solve (P3). However this is never clearly stated, and explanation in appendix C.1 makes it more confusing since it doesn't mention (P3).
- The experiments are quite toyish, and it's not clear if the observation would scale to more complex environments. In particular, the problems seem to be harder to solve for environments with large state and action spaces. In general, the paper doesn't discuss the assumptions required for the results and doesn't mention any potential limitations of the proposed approach.

---

> ### Author Response · Authors · 2022-11-23
> **Thank you for your feedback!**
>
> Thank you for your comments and valuable suggestions. We are glad to see that you find our theoretical results quite solid and that you believe they offer an elegant solution to the studied problem. Please find below our response to your comments.
>
> -----
>
> **Comments about additional discussions and details**
>
> Thank you for your comments and suggestions regarding additional discussions and details that could be added to the paper, i.e.,
>
> > Some theoretical results would benefit from a small discussion…
>
> > Some other details, …, would be worth mentioning in the main text …
>
> In the new version of the paper, we provide additional explanations of our theoretical results. Following your suggestions, we comment on:
> - why (P1’) is introduced and why it is equivalent to (P1): see Section 3.2 and the discussion below (P1’),
> - the linearity of (P3): see Section 5.1 (paragraph below Theorem 5.1) where we now more explicitly mention that this is a linear program,
> - the rationale for introducing $\Theta^{\epsilon}$: see Section 5.1, in particular, the paragraphs above Theorem 5.1 and Figure 3, as well as  the intuition behind our approach in the appendix.
>
> -----
>
> **Comments about implementation details**
>
> Thank you for your comments regarding implementation details. You are correct, our implementation is based on first computing state-action occupancy measures, and then utilizing these occupancy measures when solving the optimization problems. To solve the optimization problems, we can use standard solvers for convex programming (in our experiments, we used CVXPY). Regarding the occupancy measures, they can be computed via the Bellman flow constraints; the new version of the paper specifies these details below Theorem 5.1 (Section 5.1).
>
> Note that the occupancy measures can be pre-computed, and for this we can use the poisoned model $\widehat M$ given to the learning agent, hence, the defense has access to transition probabilities. The new version of the paper more clearly states that the defense relies on the knowledge of $\widehat M$ when computing the occupancy measure. We added this part in Section 5.1 where we discuss the computation of the occupancy measures (below Theorem 5.1), but also at the beginning of Section 3.1 where we now state that the agent derives its policy using the poisoned MDP $\widehat M$.
>
> -----
>
> **Comments about solving optimization problems (P2a)  and (P2b)**
>
> Yes, you are correct that the optimization problem (P3) is the basis for solving the optimization problem (P2a) (and consequently (P2b), since (P2a) is used for solving (P2b)). We have added a clarification below Theorem 5.1 emphasizing the importance of Theorem 5.1 (and hence (P3)) for computing the solutions of (P2a) (and consequently (P2b)). We have added similar clarifications in Appendix C.1 and the experiment section, reiterating this point.
>
> -----
>
> **Comments about limitations**
>
> We have added additional discussions and remarks about the assumptions that we rely on in this work (see our response to the comments about implementation details and requested changes). We have added an additional discussion on the limitations of this work to Section 7. Specific discussion that addresses the reviewers concerns is in the paragraph *Unknown-model and scalability*.
>
> -----
>
> **Additional comments in the requested changes section**
>
> > ... a discussion about the necessary assumptions, such as ergodicity …
>
>
> We thank the reviewer for the suggestion. The ergodicity assumption is based on prior work and we adopt it to make the attack problem feasible. We have added a remark in Section 3.2, i.e., Remark 3.1, which explains this in more detail. Please also note our response to the other comments about our assumptions (see above).
>
> >... more details about the computation of …
>
> We thank the reviewer for this suggestion. We have added an explanation for how $\psi^{\pi}$ are computed, as well as how (P3) can be solved after Theorem 5.1.
>
> > Clarify whether (P3) is used to solve (P2a) and (P2b)...
>
>
> Yes, this is indeed the case as we elaborated in our response to the previous comments. Please note that the statements of Theorem 5.1 and Theorem 5.3 imply this as well. We have added clarifications after Theorem 5.1, at the beginning of the experiment section and in Appendix C.1 to further emphasize this point.
>
> -----
>
> Thank you again for your helpful feedback. Please let us know if you have any additional comments.

---

> > ### Comment · Reviewer_5huZ · 2022-11-29
> > **Thanks for the clariffications**
> >
> > Thanks for clarifying and answering my questions.
> > I have two further minor comments:
> > - In figure 4.c it would be interesting to plot the worst-case reward function that corresponds to $\pi_D$ instead of $\hat R$.
> > - You mention that both the attacker and the defender need to know the transition probabilities in order to solve the problem. This quite a strong assumption and is often not the case in a lot of RL problems. I believe you should more clearly mention this strong limitation (you quickly mention in the conclusion that future work is needed to maybe use function approximation to solve the problem, but I believe you could make it more clear).

---

> > > ### Author Response · Authors · 2022-12-05
> > > **Thank you for your feedback!**
> > >
> > > We thank the reviewer for the additional comments. Please find our response to these comments below.
> > >
> > > > … figure 4.c it would be interesting to plot …
> > >
> > > We have added the worst-case score of our policy from the defender’s perspective in Appendix J (Figure 7).
> > > As we explain in the Appendix, this corresponds to the score under the reward function
> > > $\widehat{R}$ for the defense policy $\pi_D$. In contrast, for the attacker’s policy $\pi_\dagger$, this worst-case value generally equals $-\infty$.
> > >
> > > We further note that Figure 4.c denotes the score of the policies under the *true reward function $\overline{R}$* and not the modified reward function $\widehat{R}$.
> > >
> > > >… more clearly mention this strong limitation …
> > >
> > > We thank the reviewer for this suggestion and have edited the discussion in Section 7 to emphasize this point.

---

> > > > ### Comment · Reviewer_5huZ · 2022-12-05
> > > > **Correction about previous remark**
> > > >
> > > > I made a mistake my first remark was about figure 3.c). My intuition is that the worst-case reward should be the same for $a_1$, $a_4$ and $a_7$, but it would be great to have a confirmation.

---

> > > > > ### Author Response · Authors · 2022-12-06
> > > > > **Response to the comment about Fig. 3c**
> > > > >
> > > > > Thank you for clarifying your remark!
> > > > > While we have a worst case guarantee for the defense policy (in this case, guaranteeing the expected score of $22/3$), we do not necessarily have worst-case guarantees for each of the individual actions. In this case, choosing $a_1$ (or $a_7$) deterministically would give a worst-case score of $7$, while choosing $a_4$ deterministically, and in general choosing the target action deterministically, would give the worst-case score $-\infty$. Our policy is preferable to all of these deterministic policies as it has a worst-case score of $\frac{22}{3} \approx 7.33 > 7$.
> > > > >
> > > > >
> > > > > The key point here is that while, in the worst-case, the reward of $a_4$ can be arbitrarily bad, this can only happen when reward of $a_1$ or $a_7$ is sufficiently good.
> > > > > Formally, given $\widehat{R}$, the defense can infer that the original reward function takes the values $(7 + x, 8 - x - y, 7 + y)$ on the actions $(a_1, a_4, a_7)$, for some $x, y \ge 0$ &mdash; the rewards of the other actions remain the same and the defense policy does not select them.
> > > > > Therefore, as our policy takes the uniform distributions over the $(a_1, a_4, a_7)$, it still has the worst-case guarantee
> > > > > $$
> > > > > \frac{1}{3}(7 + x + 8 - x - y + 7 + y) = \frac{22}{3}
> > > > > $$
> > > > >
> > > > > We emphasize that this is a *worst-case bound* and there are instances where $a_4$ is an optimal action, e.g., if the reward function in Fig. 3c is the true reward function then the optimal choice is to select $a_4$.

---

> > > > > > ### Comment · Reviewer_5huZ · 2022-12-06
> > > > > > **Further clarifications**
> > > > > >
> > > > > > Still want to clarify a little bit:
> > > > > >
> > > > > > In Figure 4 you plot the reward function for each action. In figure a) you plot the original reward function $\bar R$, and in figure b-c) you plot the perturbed reward function $\hat R$.
> > > > > > What I would like to know is if in figure c) you could plot the reward function that is the solution to problem (P2b) instead of $\hat R$ ?
> > > > > > To be a bit more clear, we can rewrite (P2b), as trying to find a Nash-Equilibrium such that:
> > > > > > \begin{aligned}
> > > > > > \pi_D = \arg\max \langle \psi^\pi, R_D \rangle \\\\ R_D = \arg\min \langle \psi^{\pi_D}, R \rangle
> > > > > > \end{aligned}
> > > > > >
> > > > > > I'm simplifying a little bit, let me know if I want to develop.
> > > > > > My question is thus could you plot $R_D$ which corresponds to the reward function associated to $\pi_D$ ?

---

> > > > > > > ### Author Response · Authors · 2022-12-06
> > > > > > > **Additional clarifications**
> > > > > > >
> > > > > > > Interestingly, the reward function $R_D$ actually coincides with $\widehat{R}$ and therefore the score of policy $\pi_D$ under $R_D$ is the same as $\hat{\rho}^{\pi_D}$. We provide more details in Appendix J (added in the last revision). We would like to emphasize that repeating the same procedure with $\pi_\dagger$, i.e., calculating the worst-case for $\pi_\dagger$, would generally give us $-\infty$.
> > > > > > >
> > > > > > > Thank you again for your helpful feedback. Please let us know if you have any further comments or questions.

---

### Decision · Action_Editors · 2022-12-26

**Recommendation:** Accept as is

**Comment:**

The authors are making different sets of assumptions (mainly about the level of awareness of the agents wrt the attacks) and derive a formal description of attacks in the RL setting, using results from the statistical robustness literature. From this, they either provide a robust policy or a result of impossibility from analytical reasoning. This is a nice framework for understanding attacks and whenever a robust solution can be found. It brings some novel perspective on the problem.

The work is mainly limited to the discrete setting with reasonably small state and action spaces. The dynamics of the environment has to be fully known by the agent and information about the attacks as well. This is a quite limited setting but the authors provided enough evidence that this could be extended so that the reviewers are confident.

The main issues were about the clarity of the presentation and of the setting definition. The author addressed these issues in the comments and a revised version.

**Audience:**

This paper addresses an important problem in reinforcement learning that is more widely studied in the supervised learning setting. It is mainly a theoretical paper so the audience may be somewhat restricted. Yet this work will be of interest to a large enough audience IMO.

**Claims And Evidence:**

The work is mainly theoretical and relates to adversarial attacks in the reinforcement learning setting. Especially, the attacks are in the shape of perturbations on the reward for which the policy is not optimal anymore wrt the original reward. The authors provide analytical solutions to attacks. Claims are supported by strong theoretical results. There are some limited experimental results supporting the proofs.